# Constant Regret, Generalized Mixability, and Mirror Descent

**Zakaria Mhammedi**
Research School of Computer Science
Australian National University and DATA61
zak.mhammedi@anu.edu.au

**Robert C. Williamson**
Research School of Computer Science
Australian National University and DATA61
bob.williamson@anu.edu.au

## Abstract

We consider the setting of prediction with expert advice; a learner makes predictions by aggregating those of a group of experts. Under this setting, and for the right choice of loss function and "mixing" algorithm, it is possible for the learner to achieve a constant regret regardless of the number of prediction rounds. For example, a constant regret can be achieved for *mixable* losses using the *aggregating algorithm*. The *Generalized Aggregating Algorithm* (GAA) is a name for a family of algorithms parameterized by convex functions on simplices (entropies), which reduce to the aggregating algorithm when using the *Shannon entropy* S. For a given entropy $\Phi$, losses for which a constant regret is possible using the GAA are called $\Phi$-mixable. Which losses are $\Phi$-mixable was previously left as an open question. We fully characterize $\Phi$-mixability and answer other open questions posed by [6]. We show that the Shannon entropy S is fundamental in nature when it comes to mixability; any $\Phi$-mixable loss is necessarily S-mixable, and the lowest worst-case regret of the GAA is achieved using the Shannon entropy. Finally, by leveraging the connection between the *mirror descent algorithm* and the update step of the GAA, we suggest a new *adaptive* generalized aggregating algorithm and analyze its performance in terms of the regret bound.

## 1 Introduction

Two fundamental problems in learning are how to aggregate information and under what circumstances can one learn fast. In this paper, we consider the problems jointly, extending the understanding and characterization of exponential mixing due to [10], who showed that not only does the "*aggregating algorithm*" learn quickly when the loss is suitably chosen, but that it is in fact a generalization of classical Bayesian updating, to which it reduces when the loss is log-loss [12]. We consider a general class of aggregating schemes, going beyond Vovk's exponential mixing, and provide a complete characterization of the mixing behavior for general losses and general mixing schemes parameterized by an arbitrary entropy function.

In the *game of prediction with expert advice* a *learner* predicts the outcome of a random variable (outcome of the *environment*) by aggregating the predictions of a pool of experts. At the end of each prediction round, the outcome of the environment is announced and the learner and experts suffer losses based on their predictions. We are interested in algorithms that the learner can use to "aggregate" the experts' predictions and minimize the *regret* at the end of the game. In this case, the regret is defined as the difference between the cumulative loss of the learner and that of the best expert in hindsight after $T$ rounds.

The *Aggregating Algorithm* (AA) [10] achieves a constant regret — a precise notion of fast learning — for *mixable* losses; that is, the regret is bounded from above by a constant $R_\ell$ which depends only on the loss function $\ell$ and not on the number of rounds $T$. It is worth mentioning that mixability

is a weaker condition than exp-concavity, and contrary to the latter, mixability is an intrinsic, parametrization-independent notion [4].

Reid et al. [6] introduced the *Generalized Aggregating Algorithm* (GAA), going beyond the AA. The GAA is parameterized by the choice of a convex function $\Phi$ on the simplex (entropy) and reduces to the AA when $\Phi$ is the Shannon entropy. The GAA can achieve a constant regret for losses satisfying a certain condition called $\Phi$-*mixability* (characterizing when losses are $\Phi$-mixable was left as an open problem). This regret depends jointly on the *generalized mixability constant* $\eta_\ell^\Phi$ — essentially the largest $\eta$ such that $\ell$ is $(\frac{1}{\eta}\Phi)$-mixable — and the divergence $D_\Phi(e_\theta, q)$, where $q \in \Delta_k$ is a prior distribution over $k$ experts and $e_\theta$ is the $\theta$th standard basis element of $\mathbb{R}^k$ [6]. At each prediction round, the GAA can be divided into two steps; a *substitution step* where the learner picks a prediction from a set specified by the $\Phi$-mixability condition; and an *update step* where a new distribution $q$ over experts is computed depending on their performance. Interestingly, this update step is exactly the *mirror descent algorithm* [8, 5] which minimizes the weighted loss of experts.

**Contributions.** We introduce the notion of a *support loss*; given a loss $\ell$ defined on any action space, there exists a proper loss $\underline{\ell}$ which shares the same Bayes risk as $\ell$. When a loss is mixable, one can essentially work with a proper (support) loss instead — this will be the first stepping stone towards a characterization of (generalized) mixability.

The notion of $\Phi$-mixable and the GAA were previously restricted to finite losses. We extend these to allow for the use of losses which can take infinite values (such as the log-loss), and we show in this case that under the $\Phi$-mixability condition a constant regret is achievable using the GAA.

For an entropy $\Phi$ and a loss $\ell$, we derive a necessary and sufficient condition (Theorems 13 and 14) for $\ell$ to be $\Phi$-mixable. In particular, if $\ell$ and $\Phi$ satisfy some regularity conditions, then $\ell$ is $\Phi$-mixable if and only if $\eta_\ell \Phi - \mathrm{S}$ is convex on the simplex, where $\mathrm{S}$ is the Shannon entropy and $\eta_\ell$ is essentially the largest $\eta$ such that $\ell$ is $\eta$-mixable [10, 9]. This implies that a loss $\ell$ is $\Phi$-mixable only if it is $\eta$-mixable for some $\eta > 0$. This, combined with the fact that $\eta$-mixability is equivalently $(\frac{1}{\eta}\mathrm{S})$-mixability (Theorem 12), reflects one fundamental aspect of the Shannon entropy.

Then, we derive an explicit expression for the generalized mixability constant $\eta_\ell^\Phi$ (Corollary 17), and thus for the regret bound of the GAA. This allows us to compare the regret bound $R_\ell^\Phi$ of any entropy $\Phi$ with that of the Shannon entropy $\mathrm{S}$. In this case, we show (Theorem 18) that $R_\ell^\mathrm{S} \leq R_\ell^\Phi$; that is, the GAA achieves the lowest worst-case regret when using the Shannon entropy — another result which reflects the fundamental nature of the Shannon entropy.

Finally, by leveraging the connection between the GAA and the mirror descent algorithm, we present a new algorithm — the *Adaptive Generalized Aggregating Algorithm* (AGAA). This algorithm consists of changing the entropy function at each prediction round similar to the *adaptive mirror descent algorithm* [8]. We analyze the performance of this algorithm in terms of its regret bound.

**Layout.** In §2, we give some background on loss functions and present new results (Theorem 4 and 5) based on the new notion of a *proper support loss*; we show that, as far as mixability is concerned, one can always work with a proper (support) loss instead of the original loss (which can be defined on an arbitrary action space). In §3, we introduce the notions of classical and generalized mixability and derive a characterization of $\Phi$-mixability (Theorems 13 and 14). We then introduce our new algorithm — the AGAA — and analyze its performance. We conclude the paper by a general discussion and direction for future work. All proofs, except for that of Theorem 16, are deferred to Appendix C.

**Notation.** Let $m \in \mathbb{N}$. We denote $[m] \coloneqq \{1, \ldots, m\}$ and $\tilde{m} \coloneqq m - 1$. We write $\langle \cdot, \cdot \rangle$ for the standard inner product in Euclidean space. Let $\Delta_m \coloneqq \{p \in [0, +\infty[^m \colon \langle p, \mathbf{1}_m \rangle = 1\}$ be the *probability simplex* in $\mathbb{R}^m$, and let $\tilde{\Delta}_m \coloneqq \{\tilde{p} \in [0, +\infty[^{\tilde{m}} \colon \langle \tilde{p}, \mathbf{1}_{\tilde{m}} \rangle \leq 1\}$. We will extensively make use of the affine map $\mathrm{II}_m \colon \mathbb{R}^{\tilde{m}} \to \mathbb{R}^m$ defined by

$$\mathrm{II}_m(u) \coloneqq [u_1, \ldots, u_{\tilde{m}}, 1 - \langle u, \mathbf{1}_{\tilde{m}} \rangle]^\mathsf{T}. \tag{1}$$

We denote $\operatorname{int} \mathcal{C}$, $\operatorname{ri} \mathcal{C}$, and $\operatorname{rbd} \mathcal{C}$ the *interior*, *relative interior*, and *relative boundary* of a set $\mathcal{C} \in \mathbb{R}^m$, respectively [2]. The *sub-differential* of a function $f \colon \mathbb{R}^m \to \mathbb{R} \cup \{+\infty\}$ at $u \in \mathbb{R}^m$ such that $f(u) < +\infty$ is defined by ([2])

$$\partial f(u) \coloneqq \{s^* \in \mathbb{R}^m \colon f(v) \geq f(u) + \langle s^*, v - u \rangle, \forall v \in \mathbb{R}^m\}. \tag{2}$$

Table 1 on page 9 provides a list of the main symbols used in this paper.

## 2 Loss Functions

In general, a loss function is a map $\ell\colon \mathcal{X} \times \mathcal{A} \to [0, +\infty]$ where $\mathcal{X}$ is an outcome set and $\mathcal{A}$ is an action set. In this paper, we only consider the case $\mathcal{X} = [n]$, i.e. finite outcome space. Overloading notation slightly, we define the mapping $\ell\colon \mathcal{A} \to [0, +\infty]^n$ by $[\ell(\boldsymbol{a})]_x = \ell(x, \boldsymbol{a}), \forall x \in [n]$ and denote $\ell_x(\cdot) \coloneqq [\ell(\cdot)]_x$. We further extend the new definition of $\ell$ to the set $\bigcup_{k \geq 1} \mathcal{A}^k$ such that for $x \in [n]$ and $A \coloneqq [\boldsymbol{a}_\theta]_{1 \leq \theta \leq k}^\mathsf{T} \in \mathcal{A}^k$, $\ell_x(A) \coloneqq [\ell_x(\boldsymbol{a}_\theta)]_{1 \leq \theta \leq k}^\mathsf{T} \in [0, +\infty]^k$. We define the *effective domain* of $\ell$ by $\operatorname{dom}\ell \coloneqq \{\boldsymbol{a} \in \mathcal{A}\colon \ell(\boldsymbol{a}) \in [0, +\infty[^n\}$, and the *loss surface* by $\mathcal{S}_\ell \coloneqq \{\ell(\boldsymbol{a})\colon \boldsymbol{a} \in \operatorname{dom}\ell\}$. We say that $\ell$ is *closed* if $\mathcal{S}_\ell$ is closed in $\mathbb{R}^n$. The *superprediction* set of $\ell$ is defined by $\mathscr{S}_\ell^\infty \coloneqq \{\ell(\boldsymbol{a}) + \boldsymbol{d}\colon (\boldsymbol{a}, \boldsymbol{d}) \in \mathcal{A} \times [0, +\infty]^n\}$. Let $\mathscr{S}_\ell \coloneqq \mathscr{S}_\ell^\infty \cap [0, +\infty[^n$ be its *finite* part.

Let $\boldsymbol{a}_0, \boldsymbol{a}_1 \in \mathcal{A}$. The prediction $\boldsymbol{a}_0$ is said to be *better* than $\boldsymbol{a}_1$ if the component-wise inequality $\ell(\boldsymbol{a}_0) \leq \ell(\boldsymbol{a}_1)$ holds and there exists some $x \in [n]$ such that $\ell_x(\boldsymbol{a}_0) < \ell_x(\boldsymbol{a}_1)$ [14]. A loss $\ell$ is *admissible* if for any $\boldsymbol{a} \in \mathcal{A}$ there are no better predictions.

For the rest of this paper (except for Theorem 4), we make the following assumption on losses;

**Assumption 1.** *$\ell$ is a closed, admissible loss such that $\operatorname{dom}\ell \neq \varnothing$.*

It is clear that there is no loss of generality in considering only admissible losses. The condition that $\ell$ is closed is a weaker version of the more common assumption that $\mathcal{A}$ is compact and that $\boldsymbol{a} \mapsto \ell(x, \boldsymbol{a})$ is continuous with respect to the extended topology of $[0, +\infty]$ for all $x \in [n]$ [3, 1]. In fact, we do not make any explicit topological assumptions on the set $\mathcal{A}$ ($\mathcal{A}$ is allowed to be open in our case). Our condition simply says that if a sequence of points on the loss surface converges in $[0, +\infty[^n$, then there exists an action in $\mathcal{A}$ whose image through the loss is equal to the limit. For example the 0-1 loss $\ell_{0\text{-}1}$ is closed, yet the map $\boldsymbol{p} \mapsto \ell_{0\text{-}1}(x, \boldsymbol{p})$ is not continuous on $\Delta_2$, for $x \in \{0, 1\}$.

In this paragraph let $\mathcal{A}$ be the *$n$-simplex*, i.e. $\mathcal{A} = \Delta_n$. We define the *conditional risk* $L_\ell\colon \Delta_n \times \Delta_n \to \mathbb{R}$ by $L_\ell(\boldsymbol{p}, \boldsymbol{q}) = \mathbb{E}_{x \sim \boldsymbol{p}}[\ell_x(\boldsymbol{q})] = \langle \boldsymbol{p}, \ell(\boldsymbol{q}) \rangle$ and the *Bayes risk* by $\underline{L}_\ell(\boldsymbol{p}) \coloneqq \inf_{\boldsymbol{q} \in \Delta_n} L_\ell(\boldsymbol{p}, \boldsymbol{q})$. In this case, the loss $\ell$ is *proper* if $\underline{L}_\ell(\boldsymbol{p}) = \langle \boldsymbol{p}, \ell(\boldsymbol{p}) \rangle \leq \langle \boldsymbol{p}, \ell(\boldsymbol{q}) \rangle$ for all $\boldsymbol{p} \neq \boldsymbol{q}$ in $\Delta_n$ (and *strictly proper* if the inequality is strict). For example, the *log*-loss $\ell_{\log}\colon \Delta_n \to [0, +\infty]^n$ is defined by $\ell_{\log}(\boldsymbol{p}) = -\log \boldsymbol{p}$, where the 'log' of a vector applies component-wise. One can easily check that $\ell_{\log}$ is strictly proper. We denote $\underline{L}_{\log}$ its Bayes risk.

The above definition of the Bayes risk is restricted to losses defined on the simplex. For a general loss $\ell\colon \mathcal{A} \to [0, +\infty]^n$, we use the following definition;

**Definition 2** (Bayes Risk). *Let $\ell\colon \mathcal{A} \to [0, +\infty]^n$ be a loss such that $\operatorname{dom}\ell \neq \varnothing$. The* Bayes risk *$\underline{L}_\ell\colon \mathbb{R}^n \to \mathbb{R} \cup \{-\infty\}$ is defined by*

$$\forall \boldsymbol{u} \in \mathbb{R}^n, \quad \underline{L}_\ell(\boldsymbol{u}) \coloneqq \inf_{\boldsymbol{z} \in \mathscr{S}_\ell} \langle \boldsymbol{u}, \boldsymbol{z} \rangle. \tag{3}$$

The support function of a set $\mathcal{C} \subseteq \mathbb{R}^n$ is defined by $\sigma_\mathcal{C}(\boldsymbol{u}) \coloneqq \sup_{\boldsymbol{z} \in \mathcal{C}} \langle \boldsymbol{u}, \boldsymbol{z} \rangle$, $\boldsymbol{u} \in \mathbb{R}^n$, and thus it is easy to see that one can express the Bayes risk as $\underline{L}_\ell(\boldsymbol{u}) = -\sigma_{\mathscr{S}_\ell}(-\boldsymbol{u})$. Our definition of the Bayes risk is slightly different from previous ones ([3, 9, 1]) in two ways; 1) the Bayes risk is defined on all $\mathbb{R}^n$ instead of $[0, +\infty[^n$; and 2) the infimum is taken over the finite part of the superprediction set $\mathscr{S}_\ell^\infty$. The first point is a mere mathematical convenience and makes no practical difference since $\underline{L}_\ell(\boldsymbol{p}) = -\infty$ for all $\boldsymbol{p} \notin [0, +\infty[^n$. For the second point, swapping $\mathscr{S}_\ell$ for $\mathscr{S}_\ell^\infty$ in (3) does not change the value of $\underline{L}_\ell$ for mixable losses (see Appendix D). However, we chose to work with $\mathscr{S}_\ell$ — a subset of $\mathbb{R}^n$ — as it allows us to directly apply techniques from convex analysis.

**Definition 3** (Support Loss). *We call a map $\underline{\ell}\colon \Delta_n \to [0, +\infty]^n$ a* support loss *of $\ell$ if*

$$\forall \boldsymbol{p} \in \operatorname{ri}\Delta_n, \ \underline{\ell}(\boldsymbol{p}) \in \partial\sigma_{\mathscr{S}_\ell}(-\boldsymbol{p});$$

$$\forall \boldsymbol{p} \in \operatorname{rbd}\Delta_n, \exists (\boldsymbol{p}_m) \subset \operatorname{ri}\Delta_n, \ \boldsymbol{p}_m \overset{m \to \infty}{\to} \boldsymbol{p} \ \text{and} \ \underline{\ell}(\boldsymbol{p}_m) \overset{m \to \infty}{\to} \underline{\ell}(\boldsymbol{p}) \ \text{component-wise},$$

*where $\partial\sigma_{\mathscr{S}_\ell}$ (see (2)) is the sub-differential of the support function — $\sigma_{\mathscr{S}_\ell}$ — of the set $\mathscr{S}_\ell$.*

**Theorem 4.** *Any loss $\ell\colon \mathcal{A} \to [0, +\infty]^n$ such that $\operatorname{dom}\ell \neq \varnothing$, has a proper support loss $\underline{\ell}$ with the same Bayes risk, $\underline{L}_\ell$, as $\ell$.*

Theorem 4 shows that regardless of the action space on which the loss is defined, there always exists a proper loss whose Bayes risk coincides with that of the original loss. This fact is useful in situations where the Bayes risk contains all the information one needs — such is the case for mixability. The next Theorem shows a stronger relationship between a loss and its corresponding support loss.

**Theorem 5.** *Let $\ell\colon \mathcal{A} \to [0, +\infty]^n$ be a loss and $\underline{\ell}$ be a proper support loss of $\ell$. If the Bayes risk $\underline{L}_\ell$ is differentiable on $]0, +\infty[^n$, then $\underline{\ell}$ is uniquely defined on $\mathrm{ri}\,\Delta_n$ and*

$$\forall \boldsymbol{p} \in \mathrm{dom}\,\underline{\ell}, \quad \exists \boldsymbol{a}_* \in \mathrm{dom}\,\ell, \qquad \ell(\boldsymbol{a}_*) = \underline{\ell}(\boldsymbol{p}),$$
$$\forall \boldsymbol{a} \in \mathrm{dom}\,\ell, \quad \exists (\boldsymbol{p}_m) \subset \mathrm{ri}\,\Delta_n, \quad \underline{\ell}(\boldsymbol{p}_m) \overset{m\to\infty}{\to} \ell(\boldsymbol{a}) \quad \textit{component-wise.}$$

Theorem 5 shows that when the Bayes risk is differentiable (a necessary condition for mixability — Theorem 12), the support loss is almost a reparametrization of the original loss, and in practice, it is enough to work with support losses instead. This will be crucial for characterizing $\Phi$-mixability.

## 3   Mixability in the Game of Prediction with Expert Advice

We consider the setting of prediction with expert advice [10]; there a is pool of $k$ experts, parameterized by $\theta \in [k]$, which make predictions $\boldsymbol{a}_\theta^t \in \mathcal{A}$ at each round $t$. In the same round, the learner predicts $\boldsymbol{a}_{\mathfrak{M}}^t \coloneqq \mathfrak{M}\left(\boldsymbol{a}_{1:k}^t, (x^s, \boldsymbol{a}_{1:k}^s)_{1\le s<t}\right) \in \mathcal{A}$, where $\boldsymbol{a}_{1:k}^\cdot \coloneqq [\boldsymbol{a}_\theta^\cdot]_{1\le\theta\le k}$, $(x^s) \subset [n]$ are outcomes of the environment, and $\mathfrak{M}\colon \mathcal{A}^k \times ([n] \times \mathcal{A}^k)^* \to \mathcal{A}$ is a *merging strategy* [9]. At the end of round $t$, $x^t$ is announced and each expert $\theta$ [resp. learner] suffers a loss $\ell_{x^t}(\boldsymbol{a}_\theta)$ [resp. $\ell_{x^t}(\boldsymbol{a}_{\mathfrak{M}}^t)$], where $\ell\colon \mathcal{A} \to [0, +\infty]^n$. After $T > 0$ rounds, the cumulative loss of each expert $\theta$ [resp. learner] is given by $\mathrm{Loss}_\theta^\ell(T) \coloneqq \sum_{t=1}^T \ell_{x^t}(\boldsymbol{a}_\theta^t)$ [resp. $\mathrm{Loss}_{\mathfrak{M}}^\ell(T) \coloneqq \sum_{t=1}^T \ell_{x^t}(\boldsymbol{a}_{\mathfrak{M}}^t)$]. We say that $\mathfrak{M}$ achieves a *constant regret* if $\exists R > 0, \forall T > 0, \forall \theta \in [k], \mathrm{Loss}_{\mathfrak{M}}^\ell(T) \le \mathrm{Loss}_\theta^\ell(T) + R$. In what follows, this game setting will be referred to by $\mathfrak{G}_\ell^n(\mathcal{A}, k)$ and we only consider the case where $k \ge 2$.

### 3.1   The Aggregating Algorithm and $\eta$-mixability

**Definition 6** ($\eta$-mixability). *For $\eta > 0$, a loss $\ell\colon \mathcal{A} \to [0, +\infty]^n$ is said to be $\eta$-mixable, if $\forall \boldsymbol{q} \in \Delta_k$,*

$$\forall \boldsymbol{a}_{1:k} \in \mathcal{A}^k, \exists \boldsymbol{a}_* \in \mathcal{A}, \forall x \in [n], \quad \ell_x(\boldsymbol{a}_*) \le -\eta^{-1} \log \langle \boldsymbol{q}, \exp(-\eta\ell_x(\boldsymbol{a}_{1:k})) \rangle, \qquad (4)$$

*where the* exp *applies component-wise. Letting $\mathfrak{H}_\ell \coloneqq \{\eta > 0\colon \ell \text{ is } \eta\text{-mixable}\}$, we define the* mixability constant *of $\ell$ by $\underline{\eta}_\ell \coloneqq \sup \mathfrak{H}_\ell$ if $\mathfrak{H}_\ell \ne \varnothing$; and $0$ otherwise. $\ell$ is said to be* mixable *if $\eta_\ell > 0$.*

If a loss $\ell$ is $\eta$-mixable for $\eta > 0$, the AA (Algorithm 1) achieves a constant regret in the $\mathfrak{G}_\ell^n(\mathcal{A}, k)$ game[10]. In Algorithm 1, the map $\mathfrak{S}_\ell\colon \mathscr{S}_\ell^\infty \to \mathcal{A}$ is a *substitution function* of the loss $\ell$ [10, 4]; that is, $\mathfrak{S}_\ell$ satisfies the component-wise inequality $\ell(\mathfrak{S}_\ell(\boldsymbol{s})) \le \boldsymbol{s}$, for all $\boldsymbol{s} \in \mathscr{S}_\ell^\infty$.

It was shown by Chernov et al. [1] that the $\eta$-mixability condition (4) is equivalent to the convexity of the $\eta$-*exponentiated* superprediction set of $\ell$ defined by $\exp(-\eta\mathscr{S}_\ell^\infty) \coloneqq \{\exp(-\eta\boldsymbol{s})\colon \boldsymbol{s} \in \mathscr{S}_\ell^\infty\}$. Using this fact, van Erven et al. [9] showed that the mixability constant $\eta_\ell$ of a strictly proper loss $\ell\colon \Delta_n \to [0, +\infty[^n$, whose Bayes risk is twice continuously differentiable on $]0, +\infty[^n$, is equal to

$$\underline{\eta}_\ell \coloneqq \inf_{\tilde{\boldsymbol{p}} \in \mathrm{int}\,\tilde{\Delta}_n} (\lambda_{\max}([\mathsf{H}\tilde{\underline{L}}_{\log}(\tilde{\boldsymbol{p}})]^{-1}\mathsf{H}\tilde{\underline{L}}_\ell(\tilde{\boldsymbol{p}})))^{-1}, \qquad (5)$$

where $\mathsf{H}$ is the Hessian operator and $\tilde{\underline{L}}_\cdot \coloneqq \underline{L}_\cdot \circ \mathrm{II}_n$ ($\mathrm{II}_n$ was defined in (1)). The next theorem extends this result by showing that the mixability constant $\eta_\ell$ of any loss $\ell$ is lower bounded by $\underline{\eta}_\ell$ in (5), as long as $\ell$ satisfies Assumption 1 and its Bayes risk is twice differentiable.

**Theorem 7.** *Let $\eta > 0$ and $\ell\colon \mathcal{A} \to [0, +\infty]^n$ be a loss. Suppose that $\mathrm{dom}\,\ell = \mathcal{A}$ and that $\underline{L}_\ell$ is twice differentiable on $]0, +\infty[^n$. If $\underline{\eta}_\ell > 0$ then $\ell$ is $\underline{\eta}_\ell$-mixable. In particular, $\eta_\ell \ge \underline{\eta}_\ell$.*

We later show that, under the same conditions as Theorem 7, we actually have $\eta_\ell = \underline{\eta}_\ell$ (Theorem 16) which indicates that the Bayes risk contains all the information necessary to characterize mixability.

**Remark 8.** *In practice, the requirement '$\mathrm{dom}\,\ell = \mathcal{A}$' is not necessarily a strict restriction to finite losses; it is often the case that a loss $\bar{\ell}: \bar{\mathcal{A}} \to [0, +\infty]^n$ only takes infinite values on the relative boundary of $\bar{\mathcal{A}}$ (such is the case for the $\log$-loss defined on the simplex), and thus the restriction $\ell \coloneqq \bar{\ell}|_{\mathcal{A}}$, where $\mathcal{A} = \mathrm{ri}\,\bar{\mathcal{A}}$, satisfies $\mathrm{dom}\,\ell = \mathcal{A}$. It follows trivially from the definition of mixability (4) that if $\ell$ is $\eta$-mixable and $\bar{\ell}$ is continuous with respect to the extended topology of $[0, +\infty]^n$ — a condition often satisfied — then $\bar{\ell}$ is also $\eta$-mixable.*

## 3.2 The Generalized Aggregating Algorithm and $(\eta, \Phi)-$mixability

A function $\Phi\colon \mathbb{R}^k \to \mathbb{R} \cup \{+\infty\}$ is an *entropy* if it is convex, its epigraph $\operatorname{epi}\Phi := \{(\boldsymbol{u}, h)\colon \Phi(\boldsymbol{u}) \leq h\}$ is closed in $\mathbb{R}^k \times \mathbb{R}$, and $\Delta_k \subseteq \operatorname{dom}\Phi := \{\boldsymbol{u} \in \mathbb{R}^k\colon \Phi(\boldsymbol{u}) < +\infty\}$. For example, the *Shannon entropy* is defined by $S(\boldsymbol{q}) = +\infty$ if $\boldsymbol{q} \notin [0, +\infty[^k$, and

$$\forall \boldsymbol{q} \in [0, +\infty[^k, \quad S(\boldsymbol{q}) = \sum_{i \in [k]\colon q_i \neq 0} q_i \log q_i, \tag{6}$$

The *divergence* generated by an entropy $\Phi$ is the map $D_\Phi\colon \mathbb{R}^n \times \operatorname{dom}\Phi \to [0, +\infty]$ defined by

$$D_\Phi(\boldsymbol{v}, \boldsymbol{u}) := \begin{cases} \Phi(\boldsymbol{v}) - \Phi(\boldsymbol{u}) - \Phi'(\boldsymbol{u}; \boldsymbol{v} - \boldsymbol{u}), & \text{if } \boldsymbol{v} \in \operatorname{dom}\Phi; \\ +\infty, & \text{otherwise.} \end{cases} \tag{7}$$

where $\Phi'(\boldsymbol{u}; \boldsymbol{v} - \boldsymbol{u}) := \lim_{\lambda\downarrow 0}[\Phi(\boldsymbol{u} + \lambda(\boldsymbol{v} - \boldsymbol{u})) - \Phi(\boldsymbol{u})]/\lambda$ (the limit exists since $\Phi$ is convex [7]).

**Definition 9** ($\Phi$-mixability). *Let* $\Phi\colon \mathbb{R}^k \to \mathbb{R} \cup \{+\infty\}$ *be an entropy. A loss* $\ell\colon \mathcal{A} \to [0, +\infty]^n$ *is* $(\eta, \Phi)$*-mixable for* $\eta > 0$ *if* $\forall \boldsymbol{q} \in \Delta_k$, $\forall \boldsymbol{a}_{1:k} \in \mathcal{A}^k$, $\exists \boldsymbol{a}_* \in \mathcal{A}$, *such that*

$$\forall x \in [n], \ \ell_x(\boldsymbol{a}_*) \leq \operatorname{Mix}_\Phi^\eta(\ell_x(\boldsymbol{a}_{1:k}), \boldsymbol{q}) := \inf_{\hat{\boldsymbol{q}} \in \Delta_k} \langle \hat{\boldsymbol{q}}, \ell_x(\boldsymbol{a}_{1:k}) \rangle + \eta^{-1} D_\Phi(\hat{\boldsymbol{q}}, \boldsymbol{q}). \tag{8}$$

*When* $\eta = 1$, *we simply say that* $\ell$ *is* $\Phi$*-mixable and we denote* $\operatorname{Mix}_\Phi := \operatorname{Mix}_\Phi^1$. *Letting* $\mathfrak{H}_\ell^\Phi := \{\eta > 0\colon \ell$ *is* $(\eta, \Phi)$*-mixable*\}, *we define the* generalized mixability constant *of* $(\ell, \Phi)$ *by* $\eta_\ell^\Phi := \sup \mathfrak{H}_\ell^\Phi$, *if* $\mathfrak{H}_\ell^\Phi \neq \varnothing$; *and* 0 *otherwise.*

Reid et al. [6] introduced the GAA (see Algorithm 2) which uses an entropy function $\Phi\colon \mathbb{R}^k \to \mathbb{R} \cup \{+\infty\}$ and a substitution function $\mathfrak{S}_\ell$ (see previous section) to specify the learner's merging strategy $\mathfrak{M}$. It was shown that the GAA reduces to the AA when $\Phi$ is the Shannon entropy S. It was also shown that under some regularity conditions on $\Phi$, the GAA achieves a constant regret in the $\mathfrak{G}_\ell^n(\mathcal{A}, k)$ game for any finite, $(\eta, \Phi)$-mixable loss.

Our definition of $\Phi$-mixability differs slightly from that of Reid et al. [6] — we use directional derivatives to define the divergence $D_\Phi$. This distinction makes it possible to extend the GAA to losses which can take infinite values (such as the log-loss defined on the simplex). We show, in this case, that a constant regret is still achievable under the $(\eta, \Phi)$-mixability condition. Before presenting this result, we define the notion of $\Delta$-*differentiability*; for $\mathfrak{l} \subseteq [k]$, let $\Delta_\mathfrak{l} := \{\boldsymbol{q} \in \Delta_k\colon q_\theta = 0, \forall \theta \notin \mathfrak{l}\}$. We say that an entropy $\Phi$ is $\Delta$-differentiable if $\forall \mathfrak{l} \subseteq [k]$, $\forall \boldsymbol{u}, \boldsymbol{u}_0 \in \operatorname{ri}\Delta_\mathfrak{l}$, the map $\boldsymbol{z} \mapsto \Phi'(\boldsymbol{u}; \boldsymbol{z})$ is linear on $\mathcal{L}_\mathfrak{l}^0 := \{\lambda(\boldsymbol{v} - \boldsymbol{u}_0)\colon (\lambda, \boldsymbol{v}) \in \mathbb{R} \times \Delta_\mathfrak{l}\}$.

**Theorem 10.** *Let* $\Phi\colon \mathbb{R}^k \to \mathbb{R} \cup \{+\infty\}$ *be a* $\Delta$*-differentiable entropy. Let* $\ell\colon \mathcal{A} \to [0, +\infty]^n$ *be a loss (not necessarily finite) such that* $\underline{L}_\ell$ *is twice differentiable on* $]0, +\infty[^n$. *If* $\ell$ *is* $(\eta, \Phi)$*-mixable then the* GAA *achieves a constant regret in the* $\mathfrak{G}_\ell^n(\mathcal{A}, k)$ *game; for any sequence* $(x^t, \boldsymbol{a}_{1:k}^t)_{t=1}^T$,

$$\operatorname{Loss}_{\text{GAA}}^\ell(T) - \min_{\theta \in [k]} \operatorname{Loss}_\theta^\ell(T) \leq R_\ell^\Phi := \inf_{\boldsymbol{q} \in \Delta_k} \max_{\theta \in [k]} D_\Phi(\boldsymbol{e}_\theta, \boldsymbol{q})/\eta_\ell^\Phi, \tag{9}$$

*for initial distribution over experts* $\boldsymbol{q}^0 = \operatorname{argmin}_{\boldsymbol{q} \in \Delta_k} \max_{\theta \in [k]} D_\Phi(\boldsymbol{e}_\theta, \boldsymbol{q})$, *where* $\boldsymbol{e}_\theta$ *is the* $\theta$*th basis element of* $\mathbb{R}^k$, *and any substitution function* $\mathfrak{S}_\ell$.

Looking at Algorithm 2, it is clear that the GAA is divided into two steps; 1) a *substitution step* which consists of finding a prediction $\boldsymbol{a}_* \in \mathcal{A}$ satisfying the mixability condition (8) using a substitution function $\mathfrak{S}_\ell$; and 2) an *update step* where a new distribution over experts is computed. Except for the case of the AA with the log-loss (which reduces to Bayesian updating [12]), there is not a unique choice of substitution function in general. An example of substitution function $\mathfrak{S}_\ell$ is the *inverse loss* [13]. Kamalaruban et al. [4] discuss other alternatives depending on the curvature of the Bayes risk. Although the choice of $\mathfrak{S}_\ell$ can affect the performance of the algorithm to some extent [4], the regret bound in (9) remains unchanged regardless of $\mathfrak{S}_\ell$. On the other hand, the update step is well defined and corresponds to a *mirror descent step* [6] (we later use this fact to suggest a new algorithm).

| **Algorithm 1:** Aggregating Algorithm | **Algorithm 2:** Generalized Aggregating Algorithm |
|---|---|
| **input** : $q^0 \in \Delta_k$; $\eta > 0$; A $\eta$-mixable loss $\ell \colon \mathcal{A} \to [0, +\infty]^n$; A substitution function $\mathfrak{S}_\ell$. | **input** : $q^0 \in \Delta_k$; A $\Delta$-differentiable entropy $\Phi \colon \mathbb{R}^k \to \mathbb{R} \cup \{+\infty\}$; $\eta > 0$; A $(\eta, \Phi)$-mixable loss $\ell \colon \mathcal{A} \to [0, +\infty]^n$; A substitution function $\mathfrak{S}_\ell$. |
| **output** : Learner's predictions $(a_*^t)$ | **output** : Learner's predictions $(a_*^t)$ |
| **for** $t = 1$ **to** $T$ **do** | **for** $t = 1$ **to** $T$ **do** |
| $\quad$ Observe $A^t = a_{1:k}^t \in \mathcal{A}^k$; | $\quad$ Observe $A^t = a_{1:k}^t \in \mathcal{A}^k$; |
| $\quad a_*^t \leftarrow \mathfrak{S}_\ell \left( -\frac{1}{\eta} \log \sum_{\theta \in [k]} q_\theta^{t-1} e^{-\eta \ell(a_\theta^t)} \right)$; | $\quad a_*^t \leftarrow \mathfrak{S}_\ell \left( \left[ \mathrm{Mix}_\Phi^\eta(\ell_x(A^t), q^{t-1}) \right]_{1 \le x \le n}^\mathsf{T} \right)$; |
| $\quad$ Observe outcome $x^t \in [n]$; | $\quad$ Observe outcome $x^t \in [n]$; |
| $\quad q_\theta^t \leftarrow \dfrac{q_\theta^{t-1} \exp(-\eta \ell_{x^t}(a_\theta^t))}{\langle q^{t-1}, \exp(-\eta \ell_{x^t}(A^t)) \rangle}, \forall \theta \in [k]$; | $\quad q^t \leftarrow \underset{\mu \in \Delta_k}{\mathrm{argmin}} \langle \mu, \ell_{x^t}(A^t) \rangle + \frac{1}{\eta} D_\Phi(\mu, q^{t-1})$; |
| **end** | **end** |

We conclude this subsection with two new and important results which will lead to a characterization of $\Phi$-mixability. The first result shows that $(\eta, \mathrm{S})$-mixability is equivalent to $\eta$-mixability, and the second rules out losses and entropies for which $\Phi$-mixability is not possible.

**Theorem 11.** *Let $\eta > 0$. A loss $\ell \colon \mathcal{A} \to [0, +\infty]^n$ is $\eta$-mixable if and only if $\ell$ is $(\eta, \mathrm{S})$-mixable.*

**Proposition 12.** *Let $\Phi \colon \mathbb{R}^k \to \mathbb{R} \cup \{+\infty\}$ be an entropy and $\ell \colon \mathcal{A} \to [0, +\infty]^n$. If $\ell$ is $\Phi$-mixable, then the Bayes risk satisfies $\underline{L}_\ell \in C^1(]0, +\infty[^n)$. If, additionally, $\underline{L}_\ell$ is twice differentiable on $]0, +\infty[^n$, then $\Phi$ must be strictly convex on $\Delta_k$.*

It should be noted that since the Bayes risk of a loss $\ell$ must be differentiable for it to be $\Phi$-mixable for some entropy $\Phi$, Theorem 5 says that we can essentially work with a proper support loss $\underline{\ell}$ of $\ell$. This will be crucial in the proof of the sufficient condition of $\Phi$-mixability (Theorem 14).

### 3.3 A Characterization of $\Phi$-Mixability

In this subsection, we first show that given an entropy $\Phi \colon \mathbb{R}^k \to \mathbb{R} \cup \{+\infty\}$ and a loss $\ell \colon \mathcal{A} \to [0, +\infty]^n$ satisfying certain regularity conditions, $\ell$ is $\Phi$-mixable if and only if

$$\boxed{\eta_\ell \Phi - \mathrm{S} \text{ is convex on } \Delta_k.} \tag{10}$$

**Theorem 13.** *Let $\eta > 0$, $\ell \colon \mathcal{A} \to [0, +\infty]^n$ a $\eta$-mixable loss, and $\Phi \colon \mathbb{R}^k \to \mathbb{R} \cup \{+\infty\}$ an entropy. If $\eta \Phi - \mathrm{S}$ is convex on $\Delta_k$, then $\ell$ is $\Phi$-mixable.*

The converse of Theorem 13 also holds under additional smoothness conditions on $\Phi$ and $\ell$;

**Theorem 14.** *Let $\ell \colon \mathcal{A} \to [0, +\infty]^n$ be a loss such that $\underline{L}_\ell$ is twice differentiable on $]0, +\infty[^n$, and $\Phi \colon \mathbb{R}^k \to \mathbb{R} \cup \{+\infty\}$ an entropy such that $\tilde{\Phi} := \Phi \circ \mathrm{II}_k$ is twice differentiable on $\mathrm{int} \, \tilde{\Delta}_k$. Then $\ell$ is $\Phi$-mixable only if $\underline{\eta_\ell} \Phi - \mathrm{S}$ is convex on $\Delta_k$.*

As consequence of Theorem 14, if a loss $\ell$ is not classically mixable, i.e. $\eta_\ell = 0$, it cannot be $\Phi$-mixable for any entropy $\Phi$. This is because $\eta_\ell \Phi - \mathrm{S} \overset{*}{=} \underline{\eta_\ell} \Phi - \mathrm{S} = -\mathrm{S}$ is not convex (where equality '*' is due to Theorem 7).

We need one more result before arriving at (10); Recall that the mixability constant $\eta_\ell$ is defined as the supremum of the set $\mathfrak{H}_\ell := \{\eta \ge 0 \colon \ell \text{ is } \eta\text{-mixable}\}$. The next lemma essentially gives a sufficient condition for this supremum to be attained when $\mathfrak{H}_\ell$ is non-empty — in this case, $\ell$ is $\eta_\ell$-mixable.

**Lemma 15.** *Let $\ell \colon \mathcal{A} \to [0, +\infty]^n$ be a loss. If $\mathrm{dom} \, \ell = \mathcal{A}$, then either $\mathfrak{H}_\ell = \varnothing$ or $\eta_\ell \in \mathfrak{H}_\ell$.*

**Theorem 16.** *Let $\ell$ and $\Phi$ be as in Theorem 14 with $\mathrm{dom} \, \ell = \mathcal{A}$. Then $\eta_\ell = \underline{\eta_\ell}$. Furthermore, $\ell$ is $\Phi$-mixable if and only if $\underline{\eta_\ell} \Phi - \mathrm{S}$ is convex on $\Delta_k$.*

*Proof.* Suppose now that $\ell$ is mixable. By Lemma 15, it follows that $\ell$ is $\eta_\ell$-mixable, and from Theorem 11, $\ell$ is $(\eta_\ell^{-1} \mathrm{S})$-mixable. Substituting $\Phi$ for $\eta_\ell^{-1} \mathrm{S}$ in Theorem 14 implies that $(\underline{\eta_\ell}/\eta_\ell - 1) \mathrm{S}$ is convex on $\mathrm{ri} \, \Delta_k$. Thus, $\eta_\ell \le \underline{\eta_\ell}$, and since from Theorem 7 $\underline{\eta_\ell} \le \eta_\ell$, we conclude that $\eta_\ell = \underline{\eta_\ell}$.

From Theorem 14, if $\ell$ is $\Phi$-mixable then $\underline{\eta_\ell}\Phi - \mathrm{S}$ is convex on $\Delta_k$. Now suppose that $\underline{\eta_\ell}\Phi - \mathrm{S}$ is convex on $\Delta_k$. This implies that $\underline{\eta_\ell} > 0$, and thus from Theorem 7, $\ell$ is $\underline{\eta_\ell}$-mixable. Now since $\ell$ is $\underline{\eta_\ell}$-mixable and $\underline{\eta_\ell}\Phi - \mathrm{S}$ is convex on $\Delta_k$, Theorem 13 implies that $\ell$ is $\overline{\Phi}$-mixable. $\qquad\square$

Note that the condition '$\mathrm{dom}\,\ell = \mathcal{A}$' is in practice not a restriction to finite losses — see Remark 8. Theorem 16 implies that under the regularity conditions of Theorem 14, the Bayes risk $\underline{L_\ell}$ [resp. $(\underline{L_\ell}, \Phi)$] contains all necessary information to characterize classical [resp. generalized] mixability.

**Corollary 17** (The Generalized Mixability Constant). *Let $\ell$ and $\Phi$ be as in Theorem 16. Then the generalized mixability constant (see Definition 9) is given by*

$$\eta_\ell^\Phi = \underline{\eta_\ell} \inf_{\tilde{\boldsymbol{q}} \in \mathrm{int}\,\tilde{\Delta}_k} \lambda_{\min}(\mathsf{H}\tilde{\Phi}(\tilde{\boldsymbol{q}})(\mathsf{H}\tilde{\mathrm{S}}(\tilde{\boldsymbol{q}}))^{-1}), \tag{11}$$

*where $\tilde{\Phi} := \Phi \circ \amalg_k, \tilde{\mathrm{S}} = \mathrm{S} \circ \amalg_k$, and $\amalg_k$ is defined in* (1).

Observe that when $\Phi = \mathrm{S}$, (11) reduces to $\eta_\ell^\mathrm{S} = \underline{\eta_\ell}$ as expected from Theorem 11 and Theorem 16.

### 3.4 The (In)dependence Between $\ell$ and $\Phi$ and the Fundamental Nature of $\mathrm{S}$

So far, we showed that the $\Phi$-mixability of losses satisfying Assumption 1 is characterized by the convexity of $\eta\Phi - \mathrm{S}$, where $\eta \in\, ]0, \underline{\eta_\ell}]$ (see Theorems 13 and 14). As a result, and contrary to what was conjectured previously [6], the generalized mixability condition does not induce a correspondence between losses and entropies; for a given loss $\ell$, there is no particular entropy $\Phi^\ell$ — specific to the choice of $\ell$ — which minimizes the regret of the GAA. Rather, the Shannon entropy $\mathrm{S}$ minimizes the regret regardless of the choice of $\ell$ (see Theorem 18 below). This reflects one fundamental aspect of the Shannon entropy.

Nevertheless, given a loss $\ell$ and entropy $\Phi$, the curvature of the loss surface $\mathcal{S}_\ell$ determines the maximum 'learning rate' $\eta_\ell^\Phi$ of the GAA; the curvature of $\mathcal{S}_\ell$ is linked to $\underline{\eta_\ell}$ through the Hessian of the Bayes risk (see Theorem 30 in Appendix H.2), which is in turn linked to $\eta_\ell^\Phi$ through (11).

Given a loss $\ell$, we now use the expression of $\eta_\ell^\Phi$ in (11) to explicitly compare the regret bounds $R_\ell^\Phi$ and $R_\ell^\mathrm{S}$ achieved with the GAA (see (9)) using entropy $\Phi$ and the Shannon entropy $\mathrm{S}$, respectively.

**Theorem 18.** *Let $\mathrm{S}, \Phi \colon \mathbb{R}^k \to \mathbb{R} \cup \{+\infty\}$, where $\mathrm{S}$ is the Shannon entropy and $\Phi$ is an entropy such that $\tilde{\Phi} := \Phi \circ \amalg_k$ is twice differentiable on $\mathrm{int}\,\tilde{\Delta}_k$. A loss $\ell \colon \mathcal{A} \to [0, +\infty[^n$ with $\underline{L_\ell}$ twice differentiable on $]0, +\infty[^n$, is $\Phi$-mixable only if $R_\ell^\mathrm{S} \le R_\ell^\Phi$.*

Theorem 18 is consistent with Vovk's result [10, §5] which essentially states that the regret bound $R_\ell^\mathrm{S} = \eta_\ell^{-1} \log k$ is in general tight for $\eta$-mixable losses.

## 4 Adaptive Generalized Aggregating Algorithm

In this section, we take advantage of the similarity between the GAA's update step and the mirror descent algorithm (see Appendix E) to devise a modification to the GAA leading to improved regret bounds in certain cases. The GAA can be modified in (at least) two immediate ways; 1) changing the learning rate at each time step to speed-up convergence; and 2) changing the entropy, i.e. the regularizer $\Phi$, at each time step — similar to the *adaptive* mirror descent algorithm [8, 5]. In the former case, one can use Corollary 17 to calculate the maximum 'learning rate' under the $\Phi$-mixability constraint. Here, we focus on the second method; changing the entropy at each round. Algorithm 3 displays the modified GAA — which we call the *Adaptive Generalized Aggregating Algorithm* (AGAA) — in its most general form. In Algorithm 3, $\Phi^\star(\boldsymbol{z}) := \sup_{\boldsymbol{q} \in \Delta_k} \langle \boldsymbol{q}, \boldsymbol{z} \rangle - \Phi(\boldsymbol{q})$ is the *entropic dual* of $\Phi$.

Given a $(\eta, \Phi)$-mixable loss $\ell$, we verify that Algorithm 3 is well defined; for simplicity, assume that $\mathrm{dom}\,\ell = \mathcal{A}$ and $\underline{L_\ell}$ is twice differentiable on $]0, +\infty[^n$. From the definition of an entropy, $|\Phi| < +\infty$ on $\Delta_k$, and thus the entropic dual $\Phi_t^\star$ is defined and finite on all $\mathbb{R}^k$ (in particular at $\boldsymbol{\theta}^t$). On the other hand, from Proposition 12, $\Phi$ is strictly convex on $\Delta_k$ which implies that $\Phi^\star$ (and thus $\Phi_t^\star$) is differentiable on $\mathbb{R}^k$ (see e.g. [2, Thm. E.4.1.1]). It remains to check that $\ell$ is $(\eta, \Phi_t)$-mixable. Since for $\eta > 0$, $(\eta, \Phi_t)$-mixability is equivalent to $(\frac{1}{\eta}\Phi_t)$-mixability (by definition), Theorem 16 implies

**Algorithm 3:** Adaptive Generalized Aggregating Algorithm (AGAA)

---

**input** : $\boldsymbol{\theta}^1 = \boldsymbol{0} \in \mathbb{R}^k$; A $\Delta$-differentiable entropy $\Phi\colon \mathbb{R}^k \to \mathbb{R} \cup \{+\infty\}$; $\eta > 0$; A $(\eta, \Phi)$-mixable
           loss $\ell\colon \mathcal{A} \to [0, +\infty[^n$; A substitution function $\mathfrak{S}_\ell$; A protocol of choosing $\boldsymbol{\beta}^t$ at round $t$.
**output** : Learner's predictions $(\boldsymbol{a}_*^t)$

**for** $t = 1$ **to** $T$ **do**
     Let $\Phi_t(\boldsymbol{w}) := \Phi(\boldsymbol{w}) - \langle \boldsymbol{w}, \boldsymbol{\beta}^t - \boldsymbol{\theta}^t \rangle$;                // New entropy
     Observe $A^t := \boldsymbol{a}_{1:k}^t \in \mathcal{A}^k$ ;                    // Experts' predictions
     $\boldsymbol{a}_*^t \leftarrow \mathfrak{S}_\ell \left( \left[ \mathrm{Mix}_{\Phi_t}^\eta (\ell_x(A^t), \nabla \Phi_t^\star(\boldsymbol{\theta}^t)) \right]_{1 \leq x \leq n}^\mathsf{T} \right)$ ;       // Learner's prediction
     Observe $x^t \in [n]$ and pick some $\boldsymbol{v}^t \in \mathbb{R}^k$;
     $\boldsymbol{\theta}^{t+1} \leftarrow \boldsymbol{\theta}^t - \eta \ell_{x^t}(A^t)$;
**end**

---

that $\ell$ is $(\eta, \Phi_t)$-mixable if and only if $\underline{\eta_\ell} \eta^{-1} \Phi_t - \mathrm{S}$ is convex on $\Delta_k$. This is in fact the case since $\Phi_t$ is an affine transformation of $\Phi$, and we have assumed that $\ell$ is $(\eta, \Phi)$-mixable.

In what follows, we focus on a particular instantiation of Algorithm 3 where we choose $\boldsymbol{\beta}^t := -\eta \sum_{s=1}^{t-1} (\ell_{x^s}(A^s) + \boldsymbol{v}^s)$, for some (arbitrary for now) $(\boldsymbol{v}^s) \subset \mathbb{R}^k$. The $(\boldsymbol{v}^t)$ vectors act as correction terms in the update step of the AGAA. Using standard duality properties (see Appendix A), it is easy to show that the AGAA reduces to the GAA except for the update step where the new distribution over experts at round $t \in [T]$ is now given by

$$\boldsymbol{q}^t = \nabla \Phi^\star (\nabla \Phi(\boldsymbol{q}^{t-1}) - \eta \ell_{x^t}(A^t) - \eta \boldsymbol{v}^t).$$

**Theorem 19.** *Let $\Phi\colon \mathbb{R}^k \to \mathbb{R} \cup \{+\infty\}$ be a $\Delta$-differentiable entropy. Let $\ell\colon \mathcal{A} \to [0, +\infty]^n$ be a loss such that $\underline{L}_\ell$ is twice differentiable on $]0, +\infty[^n$. Let $\boldsymbol{\beta}^t = -\eta \sum_{s=1}^{t-1} (\ell_{x^s}(A^s) + \boldsymbol{v}^s)$, where $\boldsymbol{v}^s \in \mathbb{R}^k$ and $A^s := \boldsymbol{a}_{1:k}^s \in \mathcal{A}^k$. If $\ell$ is $(\eta, \Phi)$-mixable then for initial distribution $\boldsymbol{q}^0 = \operatorname{argmin}_{\boldsymbol{q} \in \Delta_k} \max_{\theta \in [k]} D_\Phi(\boldsymbol{e}_\theta, \boldsymbol{q})$ and any sequence $(x^t, \boldsymbol{a}_{1:k}^t)_{t=1}^T$, the AGAA achieves the regret*

$$\forall \theta \in [k], \quad \mathrm{Loss}_{\mathrm{AGAA}}^\ell(T) - \mathrm{Loss}_\theta^\ell(T) \leq R_\ell^\Phi + \Delta R_\theta(T), \tag{12}$$

*where $\Delta R_\theta(T) := \sum_{t=1}^{T-1} (v_\theta^t - \langle \boldsymbol{v}^t, \boldsymbol{q}^t \rangle)$.*

Theorem 19 implies that if the sequence $(\boldsymbol{v}^t)$ is chosen such that $\Delta R_{\theta^*}(T)$ is negative for the best expert $\theta^*$ (in hindsight), then the regret bound '$R_\ell^\Phi + \Delta R_{\theta^*}(T)$' of the AGAA is lower than that of the GAA (see (9)), and ultimately that of the AA (when $\Phi = \mathrm{S}$). Unfortunately, due to Vovk's result [10, §5] there is no "universal" choice of $(\boldsymbol{v}^t)$ which guarantees that $\Delta R_{\theta^*}(T)$ is always negative. However, there are cases where this term is expected to be negative.

Consider a dataset where it is typical for the best experts (i.e., the $\theta^*$'s) to perform poorly at some point during the game, as measured by their average loss, for example. Under such an assumption, choosing the correction vectors $\boldsymbol{v}^t$ to be negatively proportional to the average losses of experts, i.e. $\boldsymbol{v}^t := -\frac{\alpha}{t} \sum_{s=1}^t \ell_{x^s}(A^s)$ (for small enough $\alpha > 0$), would be consistent with the idea of making $\Delta R_{\theta^*}(T)$ negative. To see this, suppose expert $\theta^*$ is performing poorly during the game (say at $t < T$), as measured by its instantaneous and average loss. At that point the distribution $\boldsymbol{q}^t$ would put more weight on experts performing better than $\theta^*$, i.e. having a lower average loss. And since $v_\theta^t$ is negatively proportional to the average loss of expert $\theta$, the quantity $v_{\theta^*}^t - \langle \boldsymbol{v}^t, \boldsymbol{q}^t \rangle$ would be negative — consistent with making $\Delta R_{\theta^*}(T) < 0$. On the other hand, if expert $\theta^*$ performs well during the game (say close to the best) then $v_{\theta^*}^t - \langle \boldsymbol{v}^t, \boldsymbol{q}^t \rangle \simeq 0$, since $\boldsymbol{q}^t$ would put comparable weights between $\theta^*$ and other experts (if any) with similar performance.

**Example 1.** *(A Negative Regret).* One can construct an example that illustrates the idea above. Consider the Brier game $\mathfrak{S}_{\ell_{\mathrm{Brier}}}^2(\Delta_2, 2)$; a probability game with 2 experts $\{\theta_1, \theta_2\}$, 2 outcomes $\{0, 1\}$, and where the loss $\ell_{\mathrm{Brier}}$ is the *Brier* loss [11] (which is 1-mixable). Assume that; expert $\theta_1$ consistently predicts $\Pr(x = 0) = 1/2$; expert $\theta_2$ predicts $\Pr(x = 0) = 1/4$ during the first 50 rounds, then switches to predicting $\Pr(x = 0) = 3/4$ thereafter; the outcome is always $x = 0$. A straightforward simulation using the AGAA with the Shannon entropy, Vovk's substitution function for the Brier loss [11], $\boldsymbol{\beta}^t$ as in Theorem 19 with $\boldsymbol{v}^t := -\frac{1}{8t} \sum_{s=1}^t \ell_{\mathrm{Brier}}(x^s, A^s)$, yields $R_{\ell_{\mathrm{Brier}}}^\Phi + \Delta R_{\theta^*}(T) \simeq -5$,

$\forall T \geq 150$, where in this case $\theta^* = \theta_2$ is the best expert for $T \geq 150$. The learner then does *better* than the best expert. If we use the AA instead, the learner does worse than $\theta_2$ by $\simeq R^{\mathrm{S}}_{\ell_{\mathrm{Brier}}} = \log 2$. $\quad\square$

In real data, the situation described above — where the best expert does not necessarily perform optimally during the game — is typical, especially when the number of rounds $T$ is large. We have tested the aggregating algorithms on real data as studied by Vovk [11]. We compared the performance of the AA with the AGAA, and found that the AGAA outperforms the AA, and in fact achieved a negative regret on two data sets. Details of the experiments are in Appendix J.

As pointed out earlier, there are situations where $\Delta R_{\theta^*}(T) \geq 0$ even for the choice of $(\boldsymbol{v}^t)$ in Example 1, and this could potentially lead to a large positive regret for the AGAA. There is an easy way to remove this risk at a small price; the outputs of the AGAA and the AA can themselves be considered as expert predictions. These predictions can in turn be passed to a new instance of the AA to yield a *meta prediction*. The resulting worst case regret is guaranteed not to exceed that of the original AA instance by more than $\eta^{-1} \log 2$ for an $\eta$-mixable loss. We test this idea in Appendix J.

## 5   Discussion and Future Work

In this work, we derived a characterization of $\Phi$-mixability, which enables a better understanding of when a constant regret is achievable in the game of prediction with expert advice. Then, borrowing techniques from mirror descent, we proposed a new "adaptive" version of the generalized aggregating algorithm. We derived a regret bound for a specific instantiation of this algorithm and discussed certain situations where the algorithm is expected to perform well. We empirically demonstrated the performance of this algorithm on football game predictions (see Appendix J).

Vovk [10, §5] essentially showed that given an $\eta$-mixable loss there is no algorithm that can achieve a lower regret bound than $\eta^{-1} \log k$ on all sequences of outcomes. There is no contradiction in trying to design algorithms which perform well in expectation (maybe better than the AA) on "typical" data while keeping the worst case regret close to $\eta^{-1} \log k$. This was the motivation behind the AGAA. In future work, we will explore other choices for the correction vector $\boldsymbol{v}^t$ with the goal of lowering the (expected) bound in (12). In the present work, we did not study the possibility of varying the learning rate $\eta$. One might obtain better regret bounds using an adaptive learning rate as is the case with the mirror descent algorithm. Our Corollary 17 is useful in that it gives an upper bound on the maximal learning rate under the $\Phi$-mixability constraint. Finally, although our Theorem 18 states that worst-case regret of the GAA is minimized when using the Shannon entropy, it would be interesting to study the dynamics of the AGAA with other entropies.

Table 1: A short list of the main symbols used in the paper

| Symbol | Description |
| --- | --- |
| $\ell$ | A loss function defined on a set $\mathcal{A}$ and taking values in $[0, +\infty]^n$ (see Sec. 2) |
| $\mathscr{S}_\ell$ | The finite part of the superprediction set of a loss $\ell$ (see Sec. 2) |
| $\underline{\ell}$ | The support loss of a loss $\ell$ (see Def. 3) |
| $\underline{L}_\ell$ | The Bayes risk corresponding to a loss $\ell$ (see Definition 2) |
| $\underline{\tilde{L}}_\ell$ | The composition of the Bayes risk with an affine function; $\underline{\tilde{L}}_\ell := \underline{L}_\ell \circ \mathrm{II}_n$ (see (1)) |
| $\mathrm{S}$ | The Shannon Entropy (see (6)) |
| $\eta_\ell$ | The mixability constant of $\ell$ (see Def. 6) ; essentially the largest $\eta$ s.t. $\ell$ is $\eta$-mixable. |
| $\underline{\eta}_\ell$ | Essentially the largest $\eta$ such that $\eta \underline{L}_\ell - \underline{L}_{\log}$ is convex (see (5) and [9]) |
| $\eta_\ell^\Phi$ | The generalized mixability constant (see Def. 9); the largest $\eta$ s.t. $\ell$ is $(\eta, \Phi)$-mixable. |
| $\mathfrak{S}_\ell$ | A substitution function of a loss $\ell$ (see Sec. 3.1) |
| $R_\ell^\Phi$ | The regret achieved by the GAA using entropy $\Phi$ (see (9) and Algorithm 2) |

**Acknowledgments**

This work was supported by the Australian Research Council and DATA61.

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
