[Supplementary Material]

# Supplementary Material for 'Constant Regret, Generalized Mixability, and Mirror Descent'

**Zakaria Mhammedi**
Research School of Computer Science
Australian National University and DATA61
zak.mhammedi@anu.edu.au

**Robert C. Williamson**
Research School of Computer Science
Australian National University and DATA61
bob.williamson@anu.edu.au

## Contents

## A   Notation and Preliminaries

For $n \in \mathbb{N}$, we define $\tilde{n} = n - 1$. We denote $[n] := \{1, \ldots, n\}$ the set of integers between $1$ and $n$. Let $\langle \cdot, \cdot \rangle$ denote the standard inner product in $\mathbb{R}^n$ and $\|\cdot\|$ the corresponding norm. Let $I_n$ and $\mathbf{1}_n$ denote the $n \times n$ identity matrix and the vector of all ones in $\mathbb{R}^n$. Let $\boldsymbol{e}_1, \ldots, \boldsymbol{e}_n$ denote the *standard basis* for $\mathbb{R}^n$. For a set $\mathfrak{l} \subsetneq \mathbb{N}$ and $\boldsymbol{r}_1, \ldots, \boldsymbol{r}_n \in \mathbb{R}^k$, we denote $[\boldsymbol{r}_i]_{i \in \mathfrak{l}} := [\boldsymbol{r}_{i_1}, \ldots, \boldsymbol{r}_{i_k}] \in \mathbb{R}^{n \times k}$, where $\mathfrak{l} = \{i_1, \ldots, i_k\}$ and $i_1 < \cdots < i_k$. We denote its transpose by $[\boldsymbol{r}_i]_{i \in \mathfrak{l}}^{\mathsf{T}} \in \mathbb{R}^{k \times n}$. For two vectors $\boldsymbol{p}, \boldsymbol{q} \in \mathbb{R}^n$, we write $\boldsymbol{p} \leq \boldsymbol{q}$ [resp. $\boldsymbol{p} < \boldsymbol{q}$], if $\forall i \in [n], p_i \leq q_i$ [resp. $p_i < q_i$]. We also denote $\boldsymbol{p} \odot \boldsymbol{q} = [p_i q_i]_{1 \leq i \leq n}^{\mathsf{T}} \in \mathbb{R}^n$ the *Hadamard product* of $\boldsymbol{p}$ and $\boldsymbol{q}$. If $(\boldsymbol{c}_k)$ is a sequence of vectors in $\mathcal{C} \subseteq \mathbb{R}^n$, we simply write $(\boldsymbol{c}_k) \subset \mathcal{C}$. For a sequence $(\boldsymbol{v}_m) \subset \mathbb{R}^n$, we write $\boldsymbol{v}_m \overset{m \to \infty}{\to} \boldsymbol{v}$ or $\lim_{m \to \infty} \boldsymbol{v}_m = \boldsymbol{v}$, if $\forall i \in [n], \lim_{m \to \infty} [\boldsymbol{v}_m]_i = v_i$. For a square matrix $A \in \mathbb{R}^{n \times n}$, $\lambda_{\min}(A)$ [resp. $\lambda_{\max}(A)$] denotes its minimum [resp. maximum] eigenvalue. For $k \geq 1$, $\boldsymbol{u} \in [0, +\infty[^k$ and $\boldsymbol{w} \in \mathbb{R}^k$, we define $\log \boldsymbol{u} := [\log u_i]_{1 \leq i \leq k}^{\mathsf{T}} \in \mathbb{R}^k$ and $\exp \boldsymbol{w} := [\exp w_i]_{1 \leq i \leq k}^{\mathsf{T}} \in \mathbb{R}^k$.

Let $\Delta_n := \{\boldsymbol{p} \in [0, 1]^n : \langle \boldsymbol{p}, \mathbf{1}_n \rangle = 1\}$ be the *probability simplex* in $\mathbb{R}^n$. We also define $\tilde{\Delta}_n := \{\tilde{\boldsymbol{p}} \in [0, +\infty[^{\tilde{n}} : \langle \tilde{\boldsymbol{p}}, \mathbf{1}_{\tilde{n}} \rangle \leq 1\}$. We will use the notations $\Delta_n^k := (\Delta_n)^k$ and $\tilde{\Delta}_n^k := (\tilde{\Delta}_n)^k$. For $\mathfrak{l} \subseteq [n]$, the set $\Delta_{\mathfrak{l}} = \{\boldsymbol{q} \in \Delta_n : q_i = 0, \forall i \in [n] \setminus \mathfrak{l}\}$ is a $|\mathfrak{l}|$-*face* of $\Delta_n$. We denote $\Pi_{\mathfrak{l}}^n : \mathbb{R}^n \to \mathbb{R}^{|\mathfrak{l}|}$ the linear projection operator satisfying $\Pi_{\mathfrak{l}}^n \boldsymbol{u} = [u_i]_{i \in \mathfrak{l}}^{\mathsf{T}}$. If there is no ambiguity from the context, we may simply write $\Pi_{\mathfrak{l}}$ instead of $\Pi_{\mathfrak{l}}^n$. It is easy to verify that $\Pi_{\mathfrak{l}} \Pi_{\mathfrak{l}}^{\mathsf{T}} = I_{|\mathfrak{l}|}$ and that $\boldsymbol{q} \mapsto \Pi_{\mathfrak{l}} \boldsymbol{q}$ is a bijection from $\Delta_{\mathfrak{l}} \subseteq \Delta_n$ to $\Delta_{|\mathfrak{l}|}$. In the special case where $\mathfrak{l} = [\tilde{n}]$, we write $\Pi_n := \Pi_{[\tilde{n}]}^n$ and we define the affine operator $\amalg_n : \mathbb{R}^{\tilde{n}} \to \mathbb{R}^n$ by $\amalg_n(\boldsymbol{u}) := [u_1, \ldots, u_{\tilde{n}}, 1 - \langle \boldsymbol{u}, \mathbf{1}_{\tilde{n}} \rangle]^{\mathsf{T}} = J_n \boldsymbol{u} + \boldsymbol{e}_n$, where $J_n := \begin{bmatrix} I_{\tilde{n}} \\ -\mathbf{1}_{\tilde{n}}^{\mathsf{T}} \end{bmatrix} \in \mathbb{R}^{n \times \tilde{n}}$.

For $\boldsymbol{u} \in \mathbb{R}^n$ and $c \in \mathbb{R}$, we denote $\mathcal{H}_{\boldsymbol{u}, c} := \{\boldsymbol{y} \in \mathbb{R}^n : \langle \boldsymbol{y}, \boldsymbol{u} \rangle \leq c\}$ and $\mathcal{B}(\boldsymbol{u}, c) := \{\boldsymbol{v} \in \mathbb{R}^n : \|\boldsymbol{u} - \boldsymbol{v}\| \leq c\}$. $\mathcal{H}_{\boldsymbol{u}, c}$ is a closed half space and $\mathcal{B}(\boldsymbol{u}, c)$ is the *c-ball* in $\mathbb{R}^n$ centered at $\boldsymbol{u}$. Let $\mathcal{C} \subseteq \mathbb{R}^n$ be a non-empty set. We denote $\operatorname{int} \mathcal{C}$, $\operatorname{ri} \mathcal{C}$, $\operatorname{bd} \mathcal{C}$, and $\operatorname{rbd} \mathcal{C}$ the *interior*, *relative interior*, *boundary*, and *relative boundary* of a set $\mathcal{C} \in \mathbb{R}^n$, respectively [7]. We denote the *indicator function* of $\mathcal{C}$ by $\iota_{\mathcal{C}}$, where for $\boldsymbol{u} \in \mathcal{C}$, $\iota_{\mathcal{C}}(\boldsymbol{u}) = 0$, otherwise $\iota_{\mathcal{C}}(\boldsymbol{u}) = +\infty$. The *support function* of $\mathcal{C}$ is defined by

$$\sigma_{\mathcal{C}}(\boldsymbol{u}) := \sup_{\boldsymbol{s} \in \mathcal{C}} \langle \boldsymbol{u}, \boldsymbol{s} \rangle, \ \boldsymbol{u} \in \mathbb{R}^n.$$

Let $f \colon \mathbb{R}^n \to \mathbb{R} \cup \{+\infty\}$. We denote $\operatorname{dom} f := \{\boldsymbol{u} \in \mathbb{R}^n : f(\boldsymbol{u}) < +\infty\}$ the *effective domain* of $f$. The function $f$ is *proper* if $\operatorname{dom} f \neq \varnothing$. The function $f$ is *convex* if $\forall (\boldsymbol{u}, \boldsymbol{v}) \in \mathbb{R}^n$ and $\lambda \in ]0, 1[, f(\lambda \boldsymbol{u} + (1 - \lambda) \boldsymbol{v}) \leq \lambda f(\boldsymbol{u}) + (1 - \lambda) f(\boldsymbol{v})$. When the latter inequality is strict for all $\boldsymbol{u} \neq \boldsymbol{v}$, $f$ is *strictly convex*. When $f$ is convex, it is *closed* if it is *lower semi-continuous*; that is, for all $\boldsymbol{u} \in \mathbb{R}^n$, $\liminf_{\boldsymbol{v} \to \boldsymbol{u}} f(\boldsymbol{v}) \geq f(\boldsymbol{u})$. The function $f$ is said to be 1-*homogeneous* if $\forall (\boldsymbol{u}, \alpha) \in \mathbb{R}^n \times ]0, +\infty[, f(\alpha \boldsymbol{u}) = \alpha f(\boldsymbol{u})$, and it is said to be 1-*coercive* if $\frac{f(\boldsymbol{u})}{\|\boldsymbol{u}\|} \to +\infty$ as $\|\boldsymbol{u}\| \to \infty$. Let $f$ be proper. The *sub-differential* of $f$ is defined by

$$\partial f(\boldsymbol{u}) := \{\boldsymbol{s}^* \in \mathbb{R}^n : f(\boldsymbol{v}) \geq f(\boldsymbol{u}) + \langle \boldsymbol{s}^*, \boldsymbol{v} - \boldsymbol{u} \rangle, \forall \boldsymbol{v} \in \mathbb{R}^n\}.$$

Any element $\boldsymbol{s} \in \partial f(\boldsymbol{u})$ is called a *sub-gradient* of $f$ at $\boldsymbol{u}$. We say that $f$ is *directionally differentiable* if for all $(\boldsymbol{u}, \boldsymbol{v}) \in \operatorname{dom} f \times \mathbb{R}^n$ the limit $\lim_{t \downarrow 0} \frac{f(\boldsymbol{u} + t\boldsymbol{v}) - f(\boldsymbol{u})}{t}$ exists in $[-\infty, +\infty]$. In this case, we denote the limit by $f'(\boldsymbol{u}; \boldsymbol{v})$. When $f$ is convex, it is directionally differentiable [11]. Let $f$ be proper and directionally differentiable. The *divergence* generated by $f$ is the map $D_f \colon \mathbb{R}^n \times \operatorname{dom} f \to [0, +\infty]$ defined by

$$D_f(\boldsymbol{v}, \boldsymbol{u}) := \begin{cases} f(\boldsymbol{v}) - f(\boldsymbol{u}) - f'(\boldsymbol{u}; \boldsymbol{v} - \boldsymbol{u}), & \text{if } \boldsymbol{v} \in \operatorname{dom} f; \\ +\infty, & \text{otherwise.} \end{cases}$$

For $\mathfrak{l} \subset [n]$ and $f_{\mathfrak{l}} := f \circ \Pi_{\mathfrak{l}}^{\mathsf{T}}$, it is easy to verify that $f_{\mathfrak{l}}'(\Pi_{\mathfrak{l}} \boldsymbol{p}; \Pi_{\mathfrak{l}} \boldsymbol{q} - \Pi_{\mathfrak{l}} \boldsymbol{p}) = f'(\boldsymbol{p}; \boldsymbol{q} - \boldsymbol{p}), \forall (\boldsymbol{p}, \boldsymbol{q}) \in \Delta_{\mathfrak{l}}$. In this case, it holds that $D_f(\boldsymbol{q}, \boldsymbol{p}) = D_{f_{\mathfrak{l}}}(\Pi_{\mathfrak{l}} \boldsymbol{q}, \Pi_{\mathfrak{l}} \boldsymbol{p})$. If $f$ is differentiable [resp. twice differentiable] at $\boldsymbol{u} \in \operatorname{int} \operatorname{dom} f$, we denote $\nabla f(\boldsymbol{u}) \in \mathbb{R}^n$ [resp. $\mathsf{H} f(\boldsymbol{u}) \in \mathbb{R}^{n \times n}$] its *gradient* vector [resp. *Hessian* matrix] at $\boldsymbol{u}$. A vector-valued function $g \colon \mathbb{R}^n \to \mathbb{R}^m$ is differentiable at $\boldsymbol{u}$ if for all $i \in [m]$, $g_i$ is differentiable at $\boldsymbol{u}$. In this case, the *differential* of $g$ at $\boldsymbol{u}$ is the linear operator $\mathsf{D} g(\boldsymbol{u}) : \mathbb{R}^n \to \mathbb{R}^m$ defined by $\mathsf{D} g(\boldsymbol{u}) := [\nabla g_i(\boldsymbol{u})]_{1 \leq i \leq m}^{\mathsf{T}}$. If $f$ has $k$ continuous derivatives on a set $\Omega \subset \mathbb{R}^k$, we write $f \in C^k(\Omega)$.

We define $\tilde{f} : \mathbb{R}^{\tilde{n}} \to \mathbb{R} \cup \{+\infty\}$ by $\tilde{f} := f \circ \mathrm{II}_n + \iota_{\tilde{\Delta}_n}$. That is,

$$\tilde{f}(\tilde{\boldsymbol{u}}) := \begin{cases} f(J_n \tilde{\boldsymbol{u}} + \boldsymbol{e}_n), & \text{for } \tilde{\boldsymbol{u}} \in \tilde{\Delta}_n; \\ +\infty, & \text{for } \tilde{\boldsymbol{u}} \in \mathbb{R}^{n-1} \setminus \tilde{\Delta}_n. \end{cases} \tag{1}$$

If $\tilde{f}$ is directionally differentiable, then $f'(\boldsymbol{p}, \boldsymbol{q} - \boldsymbol{p}) = \tilde{f}'(\tilde{\boldsymbol{p}}, \tilde{\boldsymbol{q}} - \tilde{\boldsymbol{p}})$, for $\boldsymbol{p}, \boldsymbol{q} \in \Delta_n$. If $\tilde{f}$ is differentiable at $\tilde{\boldsymbol{p}} = \Pi_n(\boldsymbol{p})$, then $\tilde{f}'(\tilde{\boldsymbol{p}}, \tilde{\boldsymbol{q}} - \tilde{\boldsymbol{p}}) = \langle \nabla \tilde{f}(\tilde{\boldsymbol{p}}), \tilde{\boldsymbol{q}} - \tilde{\boldsymbol{p}} \rangle$. If, additionally, $f$ is differentiable at $\boldsymbol{p} \in \mathrm{ri}\,\Delta_k$, the chain rule yields $\nabla \tilde{f}(\tilde{\boldsymbol{p}}) = J_n^{\mathsf{T}} \nabla f(\boldsymbol{p})$. Since $J_n(\tilde{\boldsymbol{p}} - \tilde{\boldsymbol{q}}) = \mathrm{II}_n(\tilde{\boldsymbol{p}} - \tilde{\boldsymbol{q}}) = \boldsymbol{p} - \boldsymbol{q}$, it also follows that $\langle \tilde{\boldsymbol{p}} - \tilde{\boldsymbol{q}}, \nabla \tilde{f}(\tilde{\boldsymbol{p}}) \rangle = \langle \boldsymbol{p} - \boldsymbol{q}, \nabla f(\boldsymbol{p}) \rangle$.

The *Fenchel dual* of a (proper) function $f$ is defined by $f^*(\boldsymbol{v}) := \sup_{\boldsymbol{u} \in \mathrm{dom}\, f} \langle \boldsymbol{u}, \boldsymbol{v} \rangle - f(\boldsymbol{u})$, and it is a closed, convex function on $\mathbb{R}^n$ [7]. The following proposition gives some useful properties of the Fenchel dual which will be used in several proofs.

**Proposition 1** ([7]). *Let $f, h : \mathbb{R}^n \to \mathbb{R} \cup \{+\infty\}$. If $f$ and $h$ are proper and there are affine functions minorizing them on $\mathbb{R}^n$, then for all $\boldsymbol{v}_0 \in \mathbb{R}^n$*

$$\begin{array}{llll}
(i) & g(\boldsymbol{u}) = f(\boldsymbol{u}) + r, \; \forall \boldsymbol{u} & \implies & g^*(\boldsymbol{v}) = f^*(\boldsymbol{v}) - r, \; \forall \boldsymbol{v} \\
(ii) & g(\boldsymbol{u}) = f(\boldsymbol{u}) + \langle \boldsymbol{v}_0, \boldsymbol{u} \rangle, \; \forall \boldsymbol{u} & \implies & g^*(\boldsymbol{v}) = f^*(\boldsymbol{v} - \boldsymbol{v}_0), \; \forall \boldsymbol{v} \\
(iii) & f \leq h & \implies & f^* \geq h^*, \\
(iv) & \boldsymbol{s} \in \partial f^*(\boldsymbol{v}), \boldsymbol{v} \in \mathbb{R}^n & \implies & f^*(\boldsymbol{v}) = \langle \boldsymbol{v}, \boldsymbol{s} \rangle - f(\boldsymbol{s}), \\
(v) & g(\boldsymbol{u}) = f(t\boldsymbol{u}), \; t > 0, \forall \boldsymbol{u} & \implies & g^*(\boldsymbol{v}) = f^*(\boldsymbol{v}/t),
\end{array}$$

A function $\Phi : \mathbb{R}^k \to \mathbb{R} \cup \{+\infty\}$ is an *entropy* if it is closed, convex, and $\Delta_k \subseteq \mathrm{dom}\,\Phi$. Its *entropic dual* $\Phi^\star : \mathbb{R}^k \to \mathbb{R} \cup \{+\infty\}$ is defined by $\Phi^\star(\boldsymbol{z}) := \sup_{\boldsymbol{q} \in \Delta_k} \langle \boldsymbol{q}, \boldsymbol{z} \rangle - \Phi(\boldsymbol{q}), \boldsymbol{z} \in \mathbb{R}^k$. For the remainder of this paper, we consider entropies defined on $\mathbb{R}^k$, where $k \geq 2$.

Let $\Phi : \mathbb{R}^k \to \mathbb{R} \cup \{+\infty\}$ be an entropy and $\Phi_\Delta := \Phi + \iota_{\Delta_k}$. In this case, $\Phi^\star = \Phi_\Delta^*$. It is clear that $\Phi_\Delta$ is 1-coercive, and therefore, $\mathrm{dom}\,\Phi^\star = \mathrm{dom}\,\Phi_\Delta^* = \mathbb{R}^k$ [7, Prop. E.1.3.8]. The entropic dual of $\Phi$ can also be expressed using the Fenchel dual of $\tilde{\Phi} : \mathbb{R}^{k-1} \to \mathbb{R} \cup \{+\infty\}$ defined by (1) after substituting $f$ by $\Phi$ and $n$ by $k$. In fact,

$$\begin{aligned}
\Phi^\star(\boldsymbol{z}) &= \sup_{\tilde{\boldsymbol{q}} \in \tilde{\Delta}_k} \langle J_k \tilde{\boldsymbol{q}} + \boldsymbol{e}_k, \boldsymbol{z} \rangle - \Phi(J_k \tilde{\boldsymbol{q}} + \boldsymbol{e}_k), \\
&= \langle \boldsymbol{e}_k, \boldsymbol{z} \rangle + \sup_{\tilde{\boldsymbol{q}} \in \tilde{\Delta}_k} \langle \tilde{\boldsymbol{q}}, J_k^{\mathsf{T}} \boldsymbol{z} \rangle - \tilde{\Phi}(\tilde{\boldsymbol{q}}), \\
&= \langle \boldsymbol{e}_k, \boldsymbol{z} \rangle + \tilde{\Phi}^*(J_k^{\mathsf{T}} \boldsymbol{z}),
\end{aligned} \tag{2}$$

where (2) follows from the fact that $\mathrm{dom}\,\tilde{\Phi} = \tilde{\Delta}_k$. Note that when $\Phi$ is an entropy, $\tilde{\Phi}$ is a closed convex function on $\mathbb{R}^{k-1}$. Hence, it holds that $\tilde{\Phi}^{**} = \tilde{\Phi}$ [11].

The *Shannon entropy* by $\mathrm{S}(\boldsymbol{q}) := \sum_{i \in [k] : q_i \neq 0} q_i \log q_i,$[1] if $\boldsymbol{q} \in [0, +\infty[^k$; and $+\infty$ otherwise.

We will also make use of the following lemma.

**Lemma 2** ([3]). $\forall m \geq 1, \forall A, B \in \mathbb{R}^{m \times m}$, $\lambda_{\max}(AB) = \lambda_{\max}(BA)$ *and* $\lambda_{\min}(AB) = \lambda_{\min}(BA)$.

## B  Technical Lemmas

This appendix presents technical lemmas which will be needed in various proofs of results from the main body of the paper.

For an open convex set $\Omega$ in $\mathbb{R}^n$ and $\alpha > 0$, a function $\phi : \Omega \to \mathbb{R}$ is said to be $\alpha$-*strongly convex* if $\boldsymbol{u} \mapsto \phi(\boldsymbol{u}) - \alpha \|\boldsymbol{u}\|^2$ is convex on $\Omega$ [8]. The next lemma is a characterization of a generalization of $\alpha$-strong convexity, where $\boldsymbol{u} \mapsto \|\boldsymbol{u}\|^2$ is replaced by any strictly convex function.

**Lemma 3.** *Let $\Omega \subseteq \mathbb{R}^n$ be an open convex set. Let $\phi, \psi : \Omega \to \mathbb{R}$ be twice differentiable.*

*If $\psi$ is strictly convex, then $\forall \boldsymbol{u} \in \Omega$, $\mathsf{H}\psi(\boldsymbol{u})$ is invertible, and for any $\alpha > 0$*

$$\forall \boldsymbol{u} \in \Omega, \; \lambda_{\min}(\mathsf{H}\phi(\boldsymbol{u})(\mathsf{H}\psi(\boldsymbol{u}))^{-1}) \geq \alpha \iff \phi - \alpha\psi \text{ is convex}, \tag{3}$$

*Furthermore, if $\alpha > 1$, then the left hand side of (3) implies that $\phi - \psi$ is strictly convex.*

*Proof.* Suppose that $\inf_{\boldsymbol{u}\in\Omega}\lambda_{\min}(\mathsf{H}\phi(\boldsymbol{u})(\mathsf{H}\psi(\boldsymbol{u}))^{-1}) \geq \alpha$. Since $g$ is strictly convex and twice differentiable on $\Omega$, $\mathsf{H}\psi(\boldsymbol{u})$ is symmetric positive definite, and thus invertible. Therefore, there exists a symmetric positive definite matrix $G \in \mathbb{R}^{n\times n}$ such that $GG = \mathsf{H}\psi(\boldsymbol{u})$. Lemma 2 implies

$$
\begin{aligned}
& \inf_{\boldsymbol{u}\in\Omega}\lambda_{\min}(\mathsf{H}\phi(\boldsymbol{u})(\mathsf{H}\psi(\boldsymbol{u}))^{-1}) && \geq \alpha, \\
\Longleftrightarrow\quad & \inf_{\boldsymbol{u}\in\Omega}\lambda_{\min}(G^{-1}\mathsf{H}\phi(\boldsymbol{u})G^{-1}) && \geq \alpha, \\
\Longleftrightarrow\quad & \forall\boldsymbol{u}\in\Omega, \forall\boldsymbol{v}\in\mathbb{R}^n\setminus\{\boldsymbol{0}\}, \frac{\boldsymbol{v}^\mathsf{T} G^{-1}(\mathsf{H}\phi(\boldsymbol{u}))G^{-1}\boldsymbol{v}}{\boldsymbol{v}^\mathsf{T}\boldsymbol{v}} && \geq \alpha, \\
\Longleftrightarrow\quad & \forall\boldsymbol{u}\in\Omega, \forall\boldsymbol{w}\in\mathbb{R}^n\setminus\{\boldsymbol{0}\}, \boldsymbol{w}^\mathsf{T}(\mathsf{H}\phi(\boldsymbol{u}))\boldsymbol{w} && \geq \alpha\boldsymbol{w}^\mathsf{T} GG\boldsymbol{w} = \boldsymbol{w}^\mathsf{T}(\alpha\mathsf{H}\psi(\boldsymbol{u}))\boldsymbol{w}, \\
\Longleftrightarrow\quad & \forall\boldsymbol{u}\in\Omega, \mathsf{H}\phi(\boldsymbol{u}) && \succeq \alpha\mathsf{H}\psi(\boldsymbol{u}), \\
\Longleftrightarrow\quad & \forall\boldsymbol{u}\in\Omega, \mathsf{H}(\phi-\alpha\psi)(\boldsymbol{u}) && \succeq 0,
\end{aligned}
$$

where in the third and fifth lines we used the definition of minimum eigenvalue and performed the change of variable $\boldsymbol{w} = G^{-1}\boldsymbol{v}$, respectively. To conclude the proof of (3), note that the positive semi-definiteness of $\mathsf{H}(\phi-\alpha\psi)$ is equivalent to the convexity of $\phi-\alpha\psi$ [7, Thm B.4.3.1].

Finally, note that the equivalences established above still hold if we replace $\alpha$, "$\geq$", and "$\succeq$" by 1, "$>$", and "$\succ$", respectively. The strict convexity of $\phi-\psi$ then follows from the positive definiteness of $\mathsf{H}(\phi-\psi)$ (ibid.). $\qquad\square$

The following result due to [5] will be crucial to prove the convexity of the superprediction set (Theorem 29).

**Lemma 4** ([5]). *Let $\Delta(\Omega)$ be the set of distributions over some set $\Omega \subseteq \mathbb{R}$. Let a function $Q : \Delta(\Omega) \times \Omega \to \mathbb{R}$ be such that $Q(\cdot, \omega)$ is continuous for all $\omega \in \Omega$. If for all $\boldsymbol{\pi} \in \Delta(\Omega)$ it holds that $\mathbb{E}_{\omega\sim\boldsymbol{\pi}}Q(\boldsymbol{\pi},\omega) \leq r$, where $r \in \mathbb{R}$ is some constant, then*

$$\exists\boldsymbol{\pi}\in\Delta(\Omega), \forall\omega\in\Omega,\ Q(\boldsymbol{\pi},\omega)\leq r.$$

Note that when $\Omega$ in the lemma above is $[n]$, $\Delta([n]) \equiv \Delta_n$.

The next crucial lemma is a slight modification of a result due to [5].

**Lemma 5.** *Let $f\colon \operatorname{ri}\Delta_n \times [n] \to \mathbb{R}$ be a continuous function in the first argument and such that $\forall(\boldsymbol{q},x)\in\operatorname{ri}\Delta_n\times[n], -\infty < f(\boldsymbol{q},x)$. Suppose that $\forall\boldsymbol{p}\in\operatorname{ri}\Delta_n, \mathbb{E}_{x\sim\boldsymbol{p}}[f(\boldsymbol{p},x)] \leq 0$, then*

$$\forall\epsilon>0, \exists\boldsymbol{p}_\epsilon\in\operatorname{ri}\Delta_n, \forall x\in[n], f(\boldsymbol{p}_\epsilon,x)\leq\epsilon.$$

*Proof.* Pick any $\delta > 0$ such that $\delta(n-1) < 1$, and $c_0 < 0$ such that $\forall(\boldsymbol{q},x)\in\operatorname{ri}\Delta_n\times[n], c_0 \leq f(\boldsymbol{q},x)$. We define $\Delta_n^\delta := \{\boldsymbol{p}\in\Delta_n : \forall x\in[n], p_x \geq \delta\}$ and $g(\boldsymbol{q},\boldsymbol{p}) := \mathbb{E}_{x\sim\boldsymbol{q}}[f(\boldsymbol{p},x)]$. For a fixed $\boldsymbol{q}$, $\boldsymbol{p}\mapsto g(\boldsymbol{q},\boldsymbol{p})$ is continuous, since $f$ is continuous in the first argument. For a fixed $\boldsymbol{p}$, $\boldsymbol{q}\mapsto g(\boldsymbol{q},\boldsymbol{p})$ is linear, and thus concave. Since $\Delta_n^\delta$ is convex and compact, $g$ satisfies Ky Fan's minimax Theorem [1, Thm. 11.4], and therefore, there exists $\boldsymbol{p}^\delta \in \Delta_n^\delta$ such that

$$\forall\boldsymbol{q}\in\Delta_n^\delta,\ \mathbb{E}_{x\sim\boldsymbol{q}}[f(\boldsymbol{p}^\delta,x)] = g(\boldsymbol{q},\boldsymbol{p}^\delta) \leq \sup_{\boldsymbol{\mu}\in\Delta_n^\delta} g(\boldsymbol{\mu},\boldsymbol{\mu}) = \sup_{\boldsymbol{\mu}\in\Delta_n^\delta} \mathbb{E}_{x\sim\boldsymbol{\mu}}[f(\boldsymbol{\mu},x)] \leq 0. \quad (4)$$

For $x_0 \in [n]$, let $\hat{\boldsymbol{q}} \in \Delta_n^\delta$ be such that $\hat{q}_{x_0} = 1 - \delta(n-1)$ and $\hat{q}_x = \delta$ for $x \neq x_0$ (this is a legitimate distribution since $\delta(n-1) < 1$ by construction). Substituting $\hat{\boldsymbol{q}}$ for $\boldsymbol{q}$ in (4) gives

$$
\begin{aligned}
& (1-\delta(n-1))f(\boldsymbol{p}^\delta,x_0) + \delta\textstyle\sum_{x\neq x_0}f(\boldsymbol{p}^\delta,x) && \leq 0, \\
\Longrightarrow\quad & (1-\delta(n-1))f(\boldsymbol{p}^\delta,x_0) && \leq -c_0\delta(n-1), \\
\Longrightarrow\quad & f(\boldsymbol{p}^\delta,x_0) && \leq [-c_0\delta(n-1)]/[1-\delta(n-1)].
\end{aligned}
$$

Choosing $\delta^* := \epsilon/[(-c_0+\epsilon)(n-1)]$, and $\boldsymbol{p}_\epsilon := \boldsymbol{p}^{\delta^*}$ gives the desired result.

$\qquad\square$

**Lemma 6.** *Let $f,g\colon I \to \mathbb{R}^n$, where $I \subseteq \mathbb{R}$ is an open interval containing $0$. Suppose $g$ [resp. $f$] is continuous [resp. differentiable] at $0$. Then $t \mapsto \langle f(t), g(t)\rangle$ is differentiable at $0$ if and only if $t \mapsto \langle f(0), g(t)\rangle$ is differentiable at $0$, and we have*

$$\frac{d}{dt}\langle f(t),g(t)\rangle\bigg|_{t=0} = \left\langle \frac{d}{dt}f(t)\bigg|_{t=0}, g(0)\right\rangle + \frac{d}{dt}\langle f(0),g(t)\rangle\bigg|_{t=0}.$$

*Proof.* We have

$$\frac{\langle f(t), g(t)\rangle - \langle f(0), g(0)\rangle}{t} = \frac{\langle f(t), g(t)\rangle - \langle f(0), g(t)\rangle}{t} + \frac{\langle f(0), g(t)\rangle - \langle f(0), g(0)\rangle}{t},$$

$$= \left\langle \frac{f(t) - f(0)}{t}, g(t)\right\rangle + \frac{\langle f(0), g(t)\rangle - \langle f(0), g(0)\rangle}{t}.$$

But since $g$ [resp. $f$] is continuous [resp. differentiable] at 0, the first term on the right hand side of the above equation converges to $\langle \frac{d}{dt} f(t)\big|_{t=0}, g(0)\rangle$ as $t \to 0$. Therefore, $\frac{1}{t}(\langle f(0), g(t)\rangle - \langle f(0), g(0)\rangle)$ admits a limit when $t \to 0$ if and only if $\frac{1}{t}(\langle f(t), g(t)\rangle - \langle f(0), g(0)\rangle)$ admits a limit when $t \to 0$. This shows that $t \mapsto \langle f(0), g(t)\rangle$ is differentiable at 0 if an only if $t \mapsto \langle f(t), g(t)\rangle$ is differentiable at 0, and in this case the above equation yields

$$\frac{d}{dt} \langle f(t), g(t)\rangle \bigg|_{t=0} = \lim_{t \to 0} \frac{\langle f(t), g(t)\rangle - \langle f(0), g(0)\rangle}{t},$$

$$= \lim_{t \to 0} \left( \left\langle \frac{f(t) - f(0)}{t}, g(t)\right\rangle + \frac{\langle f(0), g(t)\rangle - \langle f(0), g(0)\rangle}{t}\right),$$

$$= \left\langle \frac{d}{dt} f(t)\bigg|_{t=0}, g(0)\right\rangle + \frac{d}{dt} \langle f(0), g(t)\rangle \bigg|_{t=0}.$$

$\square$

Note that the differentiability of $t \mapsto \langle f(0), g(t)\rangle$ at 0 does not necessarily imply the differentiability of $g$ at 0. Take for example $n = 3$, $f(t) = \mathbf{1}/3$ for $t \in ]-1, 1[$, and

$$g(t) = \begin{cases} -t\boldsymbol{e}_1 + t\frac{\mathbf{1}}{3}, & \text{if } t \in ]-1, 0[; \\ -t\frac{\mathbf{1}}{3} + t\boldsymbol{e}_2, & \text{if } t \in [0, 1[. \end{cases}$$

Thus, the function $t \mapsto \langle f(0), g(t)\rangle = 0$ is differentiable at 0 but $g$ is not. The preceding Lemma will be particularly useful in settings where it is desired to compute the derivative $\frac{d}{dt}\langle f(0), g(t)\rangle|_{t=0}$ without any explicit assumptions on the differentiability of $g(t)$ at 0. For example, this will come up when computing $\frac{d}{dt}\langle \boldsymbol{p}, D\tilde{\ell}(\tilde{\boldsymbol{\alpha}}^t)\boldsymbol{v}\rangle|_{t=0}$, where $\boldsymbol{v} \in \mathbb{R}^{n-1}$ and $t \mapsto \tilde{\boldsymbol{\alpha}}^t$ is smooth curve on int $\tilde{\Delta}_n$, with the only assumption that $\tilde{L}_\ell$ is twice differentiable at $\tilde{\boldsymbol{\alpha}}^0 \in$ int $\tilde{\Delta}_n$.

**Lemma 7.** *Let $\ell\colon \Delta_n \to [0, +\infty]^n$ be a proper loss. For any $\boldsymbol{p} \in$ ri $\Delta_n$, it holds that*

$$\ell \text{ is continuous at } \boldsymbol{p} \overset{(i)}{\Longleftrightarrow} \underline{L}_\ell \text{ is differentiable at } \boldsymbol{p} \overset{(ii)}{\Longleftrightarrow} \partial[-\underline{L}_\ell](\boldsymbol{p}) = \{\nabla \underline{L}_\ell(\boldsymbol{p})\} = \{\ell(\boldsymbol{p})\}.$$

$\overset{(i)}{\Longleftrightarrow}$ . This equivalence has been shown before by [16].

$[\overset{(ii)}{\Longleftrightarrow}]$ Since $\underline{L}_\ell(\boldsymbol{p}) = -\sigma_{\mathscr{S}_\ell}(-\boldsymbol{p})$, for all $\boldsymbol{p} \in$ ri $\Delta_n$, it follows that $\underline{L}_\ell$ is differentiable at $\boldsymbol{p}$ if and only if $\partial[-\underline{L}_\ell](\boldsymbol{p}) = \partial \sigma_{\mathscr{S}_\ell}(-\boldsymbol{p}) = \{-\nabla \sigma_{\mathscr{S}_\ell}(-\boldsymbol{p})\} = \{\nabla \underline{L}_\ell(\boldsymbol{p})\}$ [7, Cor. D.2.1.4]. It remains to show that $\nabla \tilde{L}_\ell(\boldsymbol{r}) = \ell(\boldsymbol{r})$ when $\underline{L}_\ell$ is differentiable at $\boldsymbol{r} \in$ ri $\Delta_n$. Let $\boldsymbol{\alpha}_x^t = \boldsymbol{r} + t\boldsymbol{e}_x$ and $\tilde{\boldsymbol{\alpha}}_x^t = \Pi_n(\boldsymbol{\alpha}_x^t)$, where $(\boldsymbol{e}_x)_{x \in [n]}$ is the standard basis of $\mathbb{R}^n$. For $x \in [n]$, the functions $f_x(t) := \boldsymbol{\alpha}_x^t$ and $g_x(t) := \tilde{\ell}(\tilde{\boldsymbol{\alpha}}_x^t)$ satisfy the conditions of Lemma 6. Therefore, $h_x(t) := \langle f_x(0), g_x(t)\rangle = \langle \boldsymbol{r}, \tilde{\ell}(\tilde{\boldsymbol{\alpha}}_x^t)\rangle$ is differentiable at 0 and

$$\nabla \tilde{L}(\boldsymbol{r})\boldsymbol{e}_x = \frac{d}{dt} \tilde{L}(\boldsymbol{\alpha}_x^t)\bigg|_{t=0} = \frac{d}{dt} \langle f_x(t), g_x(t)\rangle \bigg|_{t=0},$$

$$= \left\langle \boldsymbol{e}_x, \tilde{\ell}(\tilde{\boldsymbol{r}})\right\rangle + \frac{d}{dt} h_x(t)\bigg|_{t=0},$$

$$= \tilde{\ell}_x(\tilde{\boldsymbol{r}}),$$

where the last equality holds because $h_x$ attains a minimum at 0 due to the properness of $\ell$. The result being true for all $x \in [n]$ implies that $\nabla \tilde{L}(\tilde{\boldsymbol{r}}) = \tilde{\ell}(\tilde{\boldsymbol{r}}) = \ell(\boldsymbol{r})$. $\square$

The next Lemma is a restatement of earlier results due to [14]. Our proof is more concise due to our definition of the Bayes risk in terms of the support function of the superprediction set.

**Lemma 8** ([14])**.** *Let* $\ell\colon \Delta_n \to [0,+\infty]^n$ *be a proper loss whose Bayes risk is twice differentiable on* $]0,+\infty[^n$ *and let* $X_{\boldsymbol{p}} = I_{\tilde{n}} - \mathbf{1}_{\tilde{n}}\tilde{\boldsymbol{p}}^{\mathsf{T}}$*. The following holds*

    *(i)* $\forall \boldsymbol{p} \in \operatorname{ri}\Delta_n, \langle \boldsymbol{p}, \mathsf{D}\tilde{\ell}(\tilde{\boldsymbol{p}})\rangle = \mathbf{0}_{\tilde{n}}^{\mathsf{T}}$.

    *(ii)* $\forall \tilde{\boldsymbol{p}} \in \operatorname{int}\tilde{\Delta}_n, \mathsf{D}\tilde{\ell}(\tilde{\boldsymbol{p}}) = \left[\begin{smallmatrix} X_{\boldsymbol{p}} \\ -\tilde{\boldsymbol{p}}^{\mathsf{T}} \end{smallmatrix}\right]\mathsf{H}\underline{\tilde{L}}_\ell(\tilde{\boldsymbol{p}})$.

    *(iii)* $\forall \tilde{\boldsymbol{p}} \in \operatorname{int}\tilde{\Delta}_n, \mathsf{H}\underline{\tilde{L}}_{\log}(\tilde{\boldsymbol{p}}) = -(X_{\boldsymbol{p}})^{-1}(\operatorname{diag}(\tilde{\boldsymbol{p}}))^{-1}$.

*We show (i) and (ii).* Let $\boldsymbol{p} \in \operatorname{ri}\Delta_n$ and $f(\tilde{\boldsymbol{q}}) := \langle \boldsymbol{p}, \tilde{\ell}(\tilde{\boldsymbol{q}})\rangle = \langle \boldsymbol{p}, \nabla\underline{L}_\ell(\boldsymbol{q})\rangle$, where the equality is due to Lemma 7. Since $\underline{L}_\ell$ is twice differentiable $]0,+\infty[^n$, $f$ is differentiable on $\operatorname{int}\tilde{\Delta}_n$ and we have $\mathsf{D}f(\tilde{\boldsymbol{q}}) = \langle \boldsymbol{p}, \mathsf{D}\tilde{\ell}(\tilde{\boldsymbol{p}})\rangle$. Since $\ell$ is proper, $f$ reaches a minimum at $\tilde{\boldsymbol{p}} \in \operatorname{int}\Delta_n$, and thus $\langle \boldsymbol{p}, \mathsf{D}\tilde{\ell}(\tilde{\boldsymbol{p}})\rangle = \mathbf{0}_{\tilde{n}}^{\mathsf{T}}$ (this shows (i)). On the other hand, we have $\nabla\underline{\tilde{L}}_\ell(\tilde{\boldsymbol{p}}) = J_n^{\mathsf{T}}\nabla\underline{L}_\ell(\boldsymbol{p}) = J_n^{\mathsf{T}}\tilde{\ell}(\tilde{\boldsymbol{p}})$. By differentiating and using the chain the rule, we get $\mathsf{H}\underline{\tilde{L}}_\ell(\tilde{\boldsymbol{p}}) = [\mathsf{D}\tilde{\ell}(\tilde{\boldsymbol{p}})]^{\mathsf{T}}J_n$. This means that $\forall i \in [\tilde{n}]$, $[\mathsf{H}\underline{\tilde{L}}_\ell(\tilde{\boldsymbol{p}})]_{\cdot,i} = \nabla\tilde{\ell}_i(\tilde{\boldsymbol{p}}) - \nabla\tilde{\ell}_n(\tilde{\boldsymbol{p}})$, and thus $\sum_{i=1}^{\tilde{n}} p_i[\mathsf{H}\underline{\tilde{L}}_\ell(\tilde{\boldsymbol{p}})]_{\cdot,i} = \sum_{i=1}^{\tilde{n}} p_i\nabla\tilde{\ell}_i(\tilde{\boldsymbol{p}}) - (1-p_n)\nabla\tilde{\ell}_n(\tilde{\boldsymbol{p}})$. On the other hand, it follows from point (i) of the lemma that $\sum_{i=1}^{n} p_i\nabla\tilde{\ell}_i(\tilde{\boldsymbol{p}}) = \mathbf{0}_{\tilde{n}}$. Therefore, $[\mathsf{H}\underline{\tilde{L}}_\ell(\tilde{\boldsymbol{p}})]\tilde{\boldsymbol{p}} = -\nabla\tilde{\ell}_n(\tilde{\boldsymbol{p}})$ and, as a result, $\forall i \in [\tilde{n}]$, $[\mathsf{H}\underline{\tilde{L}}_\ell(\tilde{\boldsymbol{p}})]_{\cdot,i} - [\mathsf{H}\underline{\tilde{L}}_\ell(\tilde{\boldsymbol{p}})]\tilde{\boldsymbol{p}} = \nabla\tilde{\ell}_i(\tilde{\boldsymbol{p}})$. The last two equations can be combined as $\mathsf{D}\tilde{\ell}(\tilde{\boldsymbol{p}}) = \left[\begin{smallmatrix} X_{\boldsymbol{p}} \\ -\tilde{\boldsymbol{p}}^{\mathsf{T}} \end{smallmatrix}\right]\mathsf{H}\underline{\tilde{L}}_\ell(\tilde{\boldsymbol{p}})$.

[We show (iii)] It follows from $(ii)$, since $\forall i \in [\tilde{n}], \nabla[\tilde{\ell}_{\log}]_i(\tilde{\boldsymbol{p}}) = \frac{1}{p_i}\boldsymbol{e}_i$, for $\tilde{\boldsymbol{p}} \in \operatorname{int}\tilde{\Delta}_n$.

$\square$

In the next lemma we state a new result for proper losses which will be crucial to prove a necessary condition for $\Phi$-mixability (Theorem 14) — one of the main results of the paper.

**Lemma 9.** *Let* $\ell\colon \Delta_n \to [0,+\infty]^n$ *be a proper loss whose Bayes risk is twice differentiable on* $]0,+\infty[^n$*. For* $\boldsymbol{v} \in \mathbb{R}^{n-1}$ *and* $\tilde{\boldsymbol{p}} \in \operatorname{int}\tilde{\Delta}_n$,

$$\left\langle \boldsymbol{p}, (\mathsf{D}\tilde{\ell}(\tilde{\boldsymbol{p}})\boldsymbol{v}) \odot (\mathsf{D}\tilde{\ell}(\tilde{\boldsymbol{p}})\boldsymbol{v})\right\rangle = -\boldsymbol{v}^{\mathsf{T}}\mathsf{H}\underline{\tilde{L}}_\ell(\tilde{\boldsymbol{p}})[\mathsf{H}\underline{\tilde{L}}_{\log}(\tilde{\boldsymbol{p}})]^{-1}\mathsf{H}\underline{\tilde{L}}_\ell(\tilde{\boldsymbol{p}})\boldsymbol{v}, \tag{5}$$

*where* $\boldsymbol{p} = \mathrm{II}_n(\tilde{\boldsymbol{p}})$ *and* $\underline{L}_{\log}$ *is the Bayes risk of the* log *loss.*

*Furthermore, if* $t \mapsto \tilde{\boldsymbol{\alpha}}^t$ *is a smooth curve in* $\operatorname{int}\tilde{\Delta}_n$ *and satisfies* $\tilde{\boldsymbol{\alpha}}^0 = \tilde{\boldsymbol{p}}$ *and* $\frac{d}{dt}\tilde{\boldsymbol{\alpha}}^t\big|_{t=0} = \boldsymbol{v}$*, then* $t \mapsto \langle \boldsymbol{p}, \mathsf{D}\tilde{\ell}(\tilde{\boldsymbol{\alpha}}^t)\boldsymbol{v}\rangle$ *is differentiable at 0 and we have*

$$\frac{d}{dt}\left\langle \boldsymbol{p}, \mathsf{D}\tilde{\ell}(\tilde{\boldsymbol{\alpha}}^t)\boldsymbol{v}\right\rangle\bigg|_{t=0} = -\boldsymbol{v}^{\mathsf{T}}\mathsf{H}\underline{\tilde{L}}_\ell(\tilde{\boldsymbol{p}})\boldsymbol{v}. \tag{6}$$

*Proof.* We know from Lemma 8 that for $\tilde{\boldsymbol{p}} \in \operatorname{int}\tilde{\Delta}_n$, we have $\mathsf{D}\tilde{\ell}(\tilde{\boldsymbol{p}}) = \left[\begin{smallmatrix} X_{\boldsymbol{p}} \\ -\tilde{\boldsymbol{p}}^{\mathsf{T}} \end{smallmatrix}\right]\mathsf{H}\underline{\tilde{L}}_\ell(\tilde{\boldsymbol{p}})$, where $X_{\boldsymbol{p}} = I_{n-1} - \mathbf{1}_{n-1}\tilde{\boldsymbol{p}}^{\mathsf{T}}$. Thus, we can write

$$\left\langle \boldsymbol{p}, \mathsf{D}\tilde{\ell}(\tilde{\boldsymbol{p}})\boldsymbol{v} \odot \mathsf{D}\tilde{\ell}(\tilde{\boldsymbol{p}})\boldsymbol{v}\right\rangle = \boldsymbol{v}^{\mathsf{T}}(\mathsf{D}\tilde{\ell}(\tilde{\boldsymbol{p}}))^{\mathsf{T}}\operatorname{diag}(\boldsymbol{p})\,\mathsf{D}\tilde{\ell}(\tilde{\boldsymbol{p}})\boldsymbol{v},$$
$$= \boldsymbol{v}^{\mathsf{T}}(\mathsf{H}\underline{\tilde{L}}_\ell(\tilde{\boldsymbol{p}}))^{\mathsf{T}}[X_{\boldsymbol{p}}^{\mathsf{T}}, -\tilde{\boldsymbol{p}}]\operatorname{diag}(\boldsymbol{p})\left[\begin{smallmatrix} X_{\boldsymbol{p}} \\ -\tilde{\boldsymbol{p}}^{\mathsf{T}} \end{smallmatrix}\right]\mathsf{H}\underline{\tilde{L}}_\ell(\tilde{\boldsymbol{p}})\boldsymbol{v}. \tag{7}$$

Observe that $[X_{\boldsymbol{p}}^{\mathsf{T}}, -\tilde{\boldsymbol{p}}]\operatorname{diag}(\boldsymbol{p}) = [I_{n-1} - \tilde{\boldsymbol{p}}\mathbf{1}_{n-1}^{\mathsf{T}}, -\tilde{\boldsymbol{p}}]\operatorname{diag}(\boldsymbol{p}) = [\operatorname{diag}(\tilde{\boldsymbol{p}}) - \tilde{\boldsymbol{p}}\tilde{\boldsymbol{p}}^{\mathsf{T}}, -\tilde{\boldsymbol{p}}p_n]$. Thus,

$$[X_{\boldsymbol{p}}^{\mathsf{T}}, -\tilde{\boldsymbol{p}}]\operatorname{diag}(\boldsymbol{p})\left[\begin{smallmatrix} X_{\boldsymbol{p}} \\ -\tilde{\boldsymbol{p}}^{\mathsf{T}} \end{smallmatrix}\right] = [\operatorname{diag}(\tilde{\boldsymbol{p}}) - \tilde{\boldsymbol{p}}\tilde{\boldsymbol{p}}^{\mathsf{T}}, -\tilde{\boldsymbol{p}}p_n]\left[\begin{smallmatrix} I_{n-1} - \mathbf{1}_{n-1}\tilde{\boldsymbol{p}}^{\mathsf{T}} \\ -\tilde{\boldsymbol{p}}^{\mathsf{T}} \end{smallmatrix}\right],$$
$$= \operatorname{diag}(\tilde{\boldsymbol{p}}) - \tilde{\boldsymbol{p}}\tilde{\boldsymbol{p}}^{\mathsf{T}} - \tilde{\boldsymbol{p}}\tilde{\boldsymbol{p}}^{\mathsf{T}} + \tilde{\boldsymbol{p}}\tilde{\boldsymbol{p}}^{\mathsf{T}}(1-p_n) + p_n\tilde{\boldsymbol{p}}\tilde{\boldsymbol{p}}^{\mathsf{T}},$$
$$= \operatorname{diag}(\tilde{\boldsymbol{p}}) - \tilde{\boldsymbol{p}}\tilde{\boldsymbol{p}}^{\mathsf{T}},$$
$$= \operatorname{diag}(\tilde{\boldsymbol{p}})\,X_{\boldsymbol{p}},$$
$$= -(\mathsf{H}\underline{\tilde{L}}_{\log}(\tilde{\boldsymbol{p}}))^{-1}, \tag{8}$$

where the last equality is due to Lemma 8. The desired result follows by combining (7) and (8).

[**We show** (6)] Let $\tilde{\boldsymbol{p}} \in \text{int}\,\tilde{\Delta}_n$, we define $\tilde{\boldsymbol{\alpha}}^t := \tilde{\boldsymbol{p}} + t\boldsymbol{v}$, $\boldsymbol{\alpha}^t := \mathrm{II}_n(\tilde{\boldsymbol{\alpha}}^t) = \boldsymbol{p} + tJ_n\boldsymbol{v}$, and $r(t) := \boldsymbol{\alpha}^t/\|\boldsymbol{\alpha}^t\|$, where $t \in \{s : \tilde{\boldsymbol{p}} + s\boldsymbol{v} \in \text{int}\,\tilde{\Delta}_n\}$. Since $t \mapsto r(t)$ is differentiable at 0 and $t \mapsto \mathsf{D}\tilde{\ell}(\tilde{\boldsymbol{\alpha}}^t)\boldsymbol{v}$ is continuous at 0, it follows from Lemma 3 that

$$\frac{d}{dt}\left\langle r(0), \mathsf{D}\tilde{\ell}(\tilde{\boldsymbol{\alpha}}^t)\boldsymbol{v}\right\rangle\bigg|_{t=0} = \frac{d}{dt}\left\langle r(t), \mathsf{D}\tilde{\ell}(\tilde{\boldsymbol{\alpha}}^t)\boldsymbol{v}\right\rangle\bigg|_{t=0} - \left\langle \dot{r}(0), \mathsf{D}\tilde{\ell}(\tilde{\boldsymbol{p}})\boldsymbol{v}\right\rangle,$$
$$= -\left\langle \dot{r}(0), \mathsf{D}\tilde{\ell}(\tilde{\boldsymbol{p}})\boldsymbol{v}\right\rangle,$$

where the second equality holds since, according to Lemma 8, we have $\langle\boldsymbol{\alpha}^t, \mathsf{D}\tilde{\ell}(\tilde{\boldsymbol{\alpha}}^t)\boldsymbol{v}\rangle = 0$. Since $r(0) = \boldsymbol{p}/\|\boldsymbol{p}\|$, $\dot{r}(0) = \|\boldsymbol{p}\|^{-1}(I_n - r(0)[r(0)]^\mathsf{T})J_n\boldsymbol{v}$, and $J_n = \begin{bmatrix} I_{n-1} \\ -\mathbf{1}_{n-1}^\mathsf{T} \end{bmatrix}$, we get

$$\|\tilde{\boldsymbol{p}}\|\,\frac{d}{dt}\left\langle r(0), \mathsf{D}\tilde{\ell}(\tilde{\boldsymbol{\alpha}}^t)\boldsymbol{v}\right\rangle\bigg|_{t=0} = -\left\langle \left(I_n - r(0)[r(0)]^\mathsf{T}\right)J_n\boldsymbol{v}, \mathsf{D}\tilde{\ell}(\tilde{\boldsymbol{p}})\boldsymbol{v}\right\rangle,$$
$$= -\left\langle J_n\boldsymbol{v}, \mathsf{D}\tilde{\ell}(\tilde{\boldsymbol{p}})\boldsymbol{v}\right\rangle, \tag{9}$$
$$= -\left\langle J_n\boldsymbol{v}, \begin{bmatrix} X_{\boldsymbol{p}} \\ -\tilde{\boldsymbol{p}}^\mathsf{T} \end{bmatrix} \mathsf{H}\underline{\tilde{L}}_\ell(\tilde{\boldsymbol{p}})\boldsymbol{v}\right\rangle,$$
$$= -\boldsymbol{v}^\mathsf{T}\mathsf{H}\underline{\tilde{L}}_\ell(\tilde{\boldsymbol{p}})\boldsymbol{v},$$

where the passage to (9) is due to $r(0) = \boldsymbol{p}/\|\boldsymbol{p}\| \perp \mathsf{D}\tilde{\ell}(\tilde{\boldsymbol{p}})$. In the last equality we used the fact that $J_n^\mathsf{T}\begin{bmatrix} X_{\boldsymbol{p}} \\ -\tilde{\boldsymbol{p}}^\mathsf{T} \end{bmatrix} = [I_{n-1}, -\mathbf{1}_{n-1}]\begin{bmatrix} I_{n-1} - \mathbf{1}_{n-1}\tilde{\boldsymbol{p}} \\ -\tilde{\boldsymbol{p}}^\mathsf{T} \end{bmatrix} = I_{n-1}$. $\qquad\square$

**Proposition 10.** *Let $\Phi\colon \mathbb{R}^k \to \mathbb{R} \cup \{+\infty\}$ be an entropy and $\ell\colon \mathcal{A} \to [0, +\infty]^n$ a closed admissible loss. If $\ell$ is $\Phi$-mixable, then $\forall \mathfrak{l} \subseteq [k]$ with $|\mathfrak{l}| > 1$, $\ell$ is $\Phi_{\mathfrak{l}}$-mixable and*

$$\forall \boldsymbol{q} \in \text{rbd}\,\Delta_{\mathfrak{l}}, \forall \hat{\boldsymbol{q}} \in \text{ri}\,\Delta_{\mathfrak{l}}, \ \Phi'(\boldsymbol{q}; \hat{\boldsymbol{q}} - \boldsymbol{q}) = -\infty. \tag{10}$$

Given an entropy $\Phi\colon \mathbb{R}^k \to \mathbb{R} \cup \{+\infty\}$ and a loss $\ell\colon \mathcal{A} \to [0, +\infty]$, we define

$$\mathsf{m}_\Phi(x, A, \boldsymbol{a}, \hat{\boldsymbol{q}}, \boldsymbol{\mu}) := \langle\boldsymbol{\mu}, \ell_x(A)\rangle + D_\Phi(\boldsymbol{\mu}, \hat{\boldsymbol{q}}) - \ell_x(\boldsymbol{a}),$$

where $x \in [n]$, $A \in \mathcal{A}^k$, $\boldsymbol{a} \in \mathcal{A}$, and $\boldsymbol{q}, \hat{\boldsymbol{q}} \in \Delta_k$. Reid et al. [9] showed that $\ell$ is $\Phi$ mixable if and only if

$$\widehat{\mathsf{m}_\Phi} := \inf_{A \in \mathcal{A}^k, \hat{\boldsymbol{q}} \in \Delta_k} \sup_{\boldsymbol{a}_* \in \mathcal{A}} \inf_{\boldsymbol{\mu} \in \Delta_k, x \in [n]} \mathsf{m}_\Phi(x, A, \boldsymbol{a}, \hat{\boldsymbol{q}}, \boldsymbol{\mu}) \geq 0.$$

***Proof of Proposition 10.*** [**We show that $\ell$ is $\Phi_{\mathfrak{l}}$-mixable**] Let $\mathfrak{l} \subseteq [k]$, with $|\mathfrak{l}| > 1$, $A \in \mathcal{A}^k$, and $\boldsymbol{q} \in \Delta_{\mathfrak{l}}$. Since $\ell$ is $\Phi$-mixable, the following holds

$$\exists \boldsymbol{a}_* \in \Delta_n, \forall x \in [n], \ \ell_x(\boldsymbol{a}_*) \leq \inf_{\hat{\boldsymbol{q}} \in \Delta_k}\langle\hat{\boldsymbol{q}}, \ell_x(A)\rangle + D_\Phi(\hat{\boldsymbol{q}}, \boldsymbol{q}), \tag{11}$$
$$\leq \inf_{\hat{\boldsymbol{q}} \in \Delta_{\mathfrak{l}}}\langle\hat{\boldsymbol{q}}, \ell_x(A)\rangle + D_\Phi(\hat{\boldsymbol{q}}, \boldsymbol{q}), \tag{12}$$
$$= \inf_{\hat{\boldsymbol{q}} \in \Delta_{\mathfrak{l}}}\langle\Pi_{\mathfrak{l}}\hat{\boldsymbol{q}}, \Pi_{\mathfrak{l}}\ell_x(A)\rangle + D_{\Phi_{\mathfrak{l}}}(\Pi_{\mathfrak{l}}\hat{\boldsymbol{q}}, \Pi_{\mathfrak{l}}\boldsymbol{q}),$$
$$= \inf_{\hat{\boldsymbol{\mu}} \in \Delta_{|\mathfrak{l}|}}\langle\hat{\boldsymbol{\mu}}, \ell_x(A\Pi_{\mathfrak{l}}^\mathsf{T})\rangle + D_{\Phi_{\mathfrak{l}}}(\hat{\boldsymbol{\mu}}, \Pi_{\mathfrak{l}}\boldsymbol{q}), \tag{13}$$

where in (11) we used the fact that $\Phi_{\mathfrak{l}}(\Pi_{\mathfrak{l}}\boldsymbol{q}) = \Phi(\boldsymbol{q}), \forall \boldsymbol{q} \in \Delta_{\mathfrak{l}}$. Given that $A \mapsto A\Pi_{\mathfrak{l}}^\mathsf{T}$ [resp. $\boldsymbol{q} \mapsto \Pi_{\mathfrak{l}}\boldsymbol{q}$] is onto from $\mathcal{A}^k$ to $\mathcal{A}^{|\mathfrak{l}|}$ [resp. from $\Delta_{\mathfrak{l}}$ to $\Delta_{|\mathfrak{l}|}$], (13) implies that $\ell$ is $\Phi_{\mathfrak{l}}$-mixable.

[**We show** (10)] Suppose that there exists $\hat{\boldsymbol{q}} \in \text{rbd}\,\Delta_k$ and $\boldsymbol{q} \in \text{ri}\,\Delta_k$ such that $|\Phi'(\hat{\boldsymbol{q}}; \boldsymbol{q} - \hat{\boldsymbol{q}})| < +\infty$. Let $f\colon [0, \epsilon] \to \mathbb{R}$ be defined by $f(\lambda) := \Phi(\hat{\boldsymbol{q}} + \lambda(\boldsymbol{q} - \hat{\boldsymbol{q}}))$, where $\epsilon > 0$ is such that $\hat{\boldsymbol{q}} + \epsilon(\boldsymbol{q} - \hat{\boldsymbol{q}}) \in \text{ri}\,\Delta_k$. The function $f$ is closed and convex on $\text{dom}\,f = [0, \epsilon]$ and $\lim_{\lambda\downarrow 0}\frac{f(\lambda)-f(0)}{\lambda} = f'(0; 1) = \Phi'(\hat{\boldsymbol{q}}; \boldsymbol{q} - \hat{\boldsymbol{q}})$ which is finite by assumption. Using this and the fact that $\lambda f'(0; 1) = f'(0; \lambda)$, we have $\lim_{\lambda\downarrow 0}\lambda^{-1}(f(\lambda) - f(0) - f'(0; \lambda)) = 0$. Substituting $f$ by its expression in terms of $\Phi$ in the latter equality gives

$$\lim_{\lambda\downarrow 0}\lambda^{-1}D_\Phi(\hat{\boldsymbol{q}} + \lambda(\boldsymbol{q} - \hat{\boldsymbol{q}}), \hat{\boldsymbol{q}}) = 0. \tag{14}$$

Let $\eta > 0$ and $\theta^* \in [k]$ be such that $\hat{q}_{\theta^*} = 0$. Suppose that $\ell$ is an admissible, $\Phi$-mixable loss. The fact that $\ell$ is admissible implies that there exists $(x_0, x_1, \boldsymbol{a}_0, \boldsymbol{a}_1) \in [n] \times [n] \times \mathcal{A} \times \mathcal{A}$ such that [9]

$$\boldsymbol{a}_1 \in \operatorname{argmin}\{\ell_{x_0}(\boldsymbol{a}) : \ell_{x_1}(\boldsymbol{a}) = \inf_{\hat{\boldsymbol{a}} \in \mathcal{A}} \ell_{x_1}(\hat{\boldsymbol{a}})\} \text{ and } \inf_{\boldsymbol{a} \in \mathcal{A}} \ell_{x_0}(\boldsymbol{a}) = \ell_{x_0}(\boldsymbol{a}_0) < \ell_{x_0}(\boldsymbol{a}_1). \tag{15}$$

In particular, it holds that $\ell_{x_0}(\boldsymbol{a}_0) < \ell_{x_0}(\boldsymbol{a}_1)$. Fix $A \in \mathcal{A}^k$, such that $A_{\cdot,\theta^*} = \boldsymbol{a}_0$ and $A_{\cdot,\theta} = \boldsymbol{a}_1$ for $\theta \in [k] \setminus \{\theta^*\}$. Let

$$\boldsymbol{a}_* := \operatorname*{argmax}_{\boldsymbol{a} \in \Delta_n} \inf_{\boldsymbol{\mu} \in \Delta_k, x \in [n]} \mathsf{m}_\Phi(x, A, \boldsymbol{a}, \hat{\boldsymbol{q}}, \boldsymbol{\mu}),$$

with $\hat{\boldsymbol{q}} \in \operatorname{rbd} \Delta_k$ as in (14). Note that $\boldsymbol{a}_*$ exists since $\ell$ is closed.

If $\boldsymbol{a}_*$ is such that $\ell_{x_1}(\boldsymbol{a}_*) > \ell_{x_1}(\boldsymbol{a}_1)$, then taking $\boldsymbol{\mu} = \hat{\boldsymbol{q}}$ puts all weights on experts predicting $\boldsymbol{a}_1$, while $D_\Phi(\boldsymbol{\mu}, \hat{\boldsymbol{q}}) = 0$. Therefore,

$$\widehat{\mathsf{m}_\Phi} \leq \inf_{\boldsymbol{\mu} \in \Delta_k, x \in [n]} \mathsf{m}_\Phi(x, A, \boldsymbol{a}_*, \hat{\boldsymbol{q}}, \boldsymbol{\mu}) \leq \mathsf{m}_\Phi(x_1, A, \boldsymbol{a}, \hat{\boldsymbol{q}}, \hat{\boldsymbol{q}}) < 0.$$

This contradicts the $\Phi$-mixability of $\ell$. Therefore, $\ell_{x_1}(\boldsymbol{a}_*) = \ell_{x_1}(\boldsymbol{a}_1)$, which by (15) implies $\ell_{x_0}(\boldsymbol{a}_*) \geq \ell_{x_0}(\boldsymbol{a}_1)$. For $\boldsymbol{q}^\lambda = \hat{\boldsymbol{q}} + \lambda(\boldsymbol{q} - \hat{\boldsymbol{q}})$, with $\boldsymbol{q} \in \operatorname{ri} \Delta_k$ as in (11) and $\lambda \in [0, \epsilon]$,

$$\begin{aligned}
\widehat{\mathsf{m}_\Phi} &\leq \inf_{\boldsymbol{\mu} \in \Delta_k, x \in [n]} \mathsf{m}_\Phi(x, A, \boldsymbol{a}_*, \hat{\boldsymbol{q}}, \boldsymbol{\mu}), \\
&\leq \mathsf{m}_\Phi(x_0, A, \boldsymbol{a}, \hat{\boldsymbol{q}}, \boldsymbol{q}^\lambda), \\
&= \langle \boldsymbol{q}^\lambda, \ell_{x_0}(A) \rangle + D_\Phi(\boldsymbol{q}^\lambda, \hat{\boldsymbol{q}}) - \ell_{x_0}(\boldsymbol{a}_*), \\
&= (1 - \lambda q_{\theta^*})\ell_{x_0}(\boldsymbol{a}_1) + \lambda q_{\theta^*}\ell_{x_0}(\boldsymbol{a}_0) + D_\Phi(\boldsymbol{q}^\lambda, \hat{\boldsymbol{q}}) - \ell_{x_0}(\boldsymbol{a}_*), \\
&\leq (1 - \lambda q_{\theta^*})\ell_{x_0}(\boldsymbol{a}_*) + \lambda q_{\theta^*}\ell_{x_0}(\boldsymbol{a}_0) + D_\Phi(\boldsymbol{q}^\lambda, \hat{\boldsymbol{q}}) - \ell_{x_0}(\boldsymbol{a}_*), \\
&= \lambda q_{\theta^*}(\ell_{x_0}(\boldsymbol{a}_0) - \ell_{x_0}(\boldsymbol{a}_*)) + D_\Phi(\hat{\boldsymbol{q}} + \lambda(\boldsymbol{q} - \hat{\boldsymbol{q}}), \hat{\boldsymbol{q}}).
\end{aligned}$$

Since $q_{\theta^*} > 0$ ($\boldsymbol{q} \in \operatorname{ri} \Delta_k$) and $\ell_{x_0}(\boldsymbol{a}_0) < \ell_{x_0}(\boldsymbol{a}_1) \leq \ell_{x_0}(\boldsymbol{a}_*)$, (11) implies that there exists $\lambda_* > 0$ small enough such that $\lambda_* q_{\theta^*}(\ell_{x_0}(\boldsymbol{a}_0) - \ell_{x_0}(\boldsymbol{a}_*)) + D_\Phi(\hat{\boldsymbol{q}} + \lambda_*(\boldsymbol{q} - \hat{\boldsymbol{q}}), \hat{\boldsymbol{q}}) < 0$. But this implies that $\widehat{\mathsf{m}_\Phi} < 0$ which contradicts the $\Phi$-mixability of $\ell$. Therefore, $\Phi'(\hat{\boldsymbol{q}}; \boldsymbol{q} - \hat{\boldsymbol{q}})$ is either equal to $+\infty$ or $-\infty$. The former case is not possible. In fact, since $\Phi$ is convex, it must have non-decreasing slopes; in particular, it holds that $\Phi'(\hat{\boldsymbol{q}}; \boldsymbol{q} - \hat{\boldsymbol{q}}) \leq \Phi(\boldsymbol{q} - \hat{\boldsymbol{q}}) - \Phi(\hat{\boldsymbol{q}})$. Since $\Phi$ is finite on $\Delta_k$ (by definition of an entropy), we have $\Phi'(\hat{\boldsymbol{q}}; \boldsymbol{q} - \hat{\boldsymbol{q}}) < +\infty$. Therefore, we have just shown that

$$\forall \hat{\boldsymbol{q}} \in \operatorname{rbd} \Delta_k, \forall \boldsymbol{q} \in \operatorname{ri} \Delta_k, \ \Phi'(\hat{\boldsymbol{q}}; \boldsymbol{q} - \hat{\boldsymbol{q}}) = -\infty. \tag{16}$$

Now suppose that $(\hat{\boldsymbol{q}}, \boldsymbol{q}) \in \operatorname{rbd} \Delta_\mathfrak{l} \times \operatorname{ri} \Delta_\mathfrak{l}$ for $\mathfrak{l} \subseteq [k]$, with $|\mathfrak{l}| > 1$. Note that in this case, we have $(\Phi_\mathfrak{l})'(\Pi_\mathfrak{l}\hat{\boldsymbol{q}}; \Pi_\mathfrak{l}(\boldsymbol{q} - \hat{\boldsymbol{q}})) = \Phi'(\hat{\boldsymbol{q}}; \boldsymbol{q} - \hat{\boldsymbol{q}})$. We showed in the first step of this proof that under the assumptions of the proposition, $\ell$ must be $\Phi_\mathfrak{l}$-mixable. Therefore, repeating the steps above that lead to (16) for $\Phi$, $\hat{\boldsymbol{q}}$, and $\boldsymbol{q}$ substituted by $\Phi_\mathfrak{l}$, $\Pi_\mathfrak{l}\boldsymbol{q} \in \operatorname{rbd} \Delta_{|\mathfrak{l}|}$, and $\Pi_\mathfrak{l}\boldsymbol{q} \in \operatorname{ri} \Delta_{|\mathfrak{l}|}$, we obtain $\Phi'(\hat{\boldsymbol{q}}; \boldsymbol{q} - \hat{\boldsymbol{q}}) = \Phi'_\mathfrak{l}(\Pi_\mathfrak{l}\hat{\boldsymbol{q}}; \Pi_\mathfrak{l}(\boldsymbol{q} - \hat{\boldsymbol{q}})) = -\infty$. This shows (10). $\qquad\square$

**Lemma 11.** *For $\eta > 0$, $\mathrm{S}_\eta := \eta^{-1} \mathrm{S}$ satisfies* (10) *for all $\mathfrak{l} \subseteq [k]$ such that $|\mathfrak{l}| > 1$, where $\mathrm{S}$ is the Shannon entropy.*

*Proof.* Let $\mathfrak{l} \subseteq [k]$ such that $|\mathfrak{l}| > 1$. Let $(\hat{\boldsymbol{q}}, \boldsymbol{q}) \in \operatorname{rbd} \Delta_\mathfrak{l} \times \operatorname{ri} \Delta_\mathfrak{l}$ and $\boldsymbol{q}^\lambda := \hat{\boldsymbol{q}} + \lambda(\boldsymbol{q} - \hat{\boldsymbol{q}})$, for $\lambda \in ]0, 1[$. Let $\mathfrak{J} := \{j \in \mathfrak{l} : \hat{q}_j \neq 0\}$ and $\mathfrak{K} := \mathfrak{l} \setminus \mathfrak{J}$. We have

$$\begin{aligned}
\mathrm{S}(\hat{\boldsymbol{q}}; \boldsymbol{q} - \hat{\boldsymbol{q}}) &= \lim_{\lambda \downarrow 0} \lambda^{-1} \left[ \sum_{\theta \in \mathfrak{l}} q_\theta^\lambda \log q_\theta^\lambda - \sum_{\theta' \in \mathfrak{J}} \hat{q}_{\theta'} \log \hat{q}_{\theta'} \right], \\
&= \lim_{\lambda \downarrow 0} \lambda^{-1} \left[ \sum_{\theta \in \mathfrak{J}} (q_\theta^\lambda \log q_\theta^\lambda - \hat{q}_\theta \log \hat{q}_\theta) + \sum_{\theta' \in \mathfrak{K}} q_{\theta'}^\lambda \log q_{\theta'}^\lambda \right]. \tag{17}
\end{aligned}$$

Observe that the limit of either summation term inside the bracket in (17) is equal to zero. Thus, using l'Hopital's rule we get

$$\begin{aligned}
\mathrm{S}(\hat{\boldsymbol{q}}; \boldsymbol{q} - \hat{\boldsymbol{q}}) &= \lim_{\lambda \downarrow 0} \left[ \sum_{\theta \in \mathfrak{J}} [(q_\theta - \hat{q}_\theta) \log q_\theta^\lambda + (q_\theta - \hat{q}_\theta)] + \sum_{\theta' \in \mathfrak{K}} [q_{\theta'} \log q_{\theta'}^\lambda + q_{\theta'}] \right], \\
&= \sum_{\theta \in \mathfrak{J}} (q_\theta - \hat{q}_\theta) \log \hat{q}_\theta + \sum_{\theta' \in \mathfrak{K}} q_{\theta'} \left[ \lim_{\lambda \downarrow 0} \log q_{\theta'}^\lambda \right], \tag{18}
\end{aligned}$$

where in (18) we used the fact that $\sum_{\theta\in\mathfrak{J}}(q_\theta - \hat{q}_\theta) + \sum_{\theta'\in\mathfrak{K}} q_{\theta'} = 0$. Since for all $\theta' \in \mathfrak{K}$, $\lim_{\lambda\downarrow 0} q_{\theta'}^\lambda = 0$, the right hand side of (6) is equal to $-\infty$. Therefore S satisfies (10). Since $S_\eta = \eta^{-1} S$, it is clear that $S_\eta$ also satisfies (10).

$\square$

**Lemma 12.** *Let $\Phi : \mathbb{R}^k \to \mathbb{R} \cup \{+\infty\}$ be an entropy satisfying (10) for all $\mathfrak{l} \subseteq [k]$ such that $|\mathfrak{l}| > 1$. Then for all such $\mathfrak{l}$, it holds that*

$$\forall q \in \Delta_{\mathfrak{l}}, \forall \mu \in \Delta_k \setminus \Delta_{\mathfrak{l}}, \ D_\Phi(\mu, q) = +\infty.$$

*Proof.* Let $\mu \in \Delta_k \setminus \Delta_{\mathfrak{l}}$ and $\mathfrak{J} := \{\theta \in [k] : \mu_\theta \neq 0\} \cup \mathfrak{l}$. In this case, we have $q \in \mathrm{rbd}\,\Delta_{\mathfrak{J}}$ and $q + 2^{-1}(\mu - q) \in \mathrm{ri}\,\Delta_{\mathfrak{J}}$. Thus, since $\Phi$ satisfies (10) and $\Phi'(q; \cdot)$ is 1-homogeneous [7, Prop. D.1.1.2], it follows that $2^{-1}\Phi'(q; \mu - q) = \Phi'(q; 2^{-1}(\mu - q)) = -\infty$. Hence $D_\Phi(\mu, q) = +\infty$. $\square$

**Lemma 13.** *Let $\Phi \colon \mathbb{R}^k \to \mathbb{R} \cup \{+\infty\}$ be an entropy satisfying (10) for all $\mathfrak{l} \subseteq [k]$ such that $|\mathfrak{l}| > 1$. If $\Phi$ satisfies (10), then $\partial\tilde{\Phi}(\tilde{q}) = \varnothing, \forall \tilde{q} \in \mathrm{bd}\,\tilde{\Delta}_k$. Furthermore, $\forall \mathfrak{l} \subseteq [k]$ such that $|\mathfrak{l}| > 1$,*

$$\forall d \in \mathbb{R}^k, \forall q \in \mathrm{ri}\,\Delta_{\mathfrak{l}}, \ \mathrm{Mix}_\Phi(d, q) = \mathrm{Mix}_{\Phi_{\mathfrak{l}}}(\Pi_{\mathfrak{l}} d, \Pi_{\mathfrak{l}} q).$$

*Proof.* Let $\mu \in \mathrm{rbd}\,\Delta_k$. Since $\Phi$ satisfies (10), it follows that $\forall q \in \mathrm{ri}\,\Delta_k, \tilde{\Phi}(\tilde{\mu}; \tilde{q} - \tilde{\mu}) = \Phi'(\mu; q - \mu) = -\infty$. Therefore, $\partial\tilde{\Phi}'(\tilde{\mu}) = \varnothing$ [11, Thm. 23.4].

Let $d \in \mathbb{R}^n, \mathfrak{l} \subseteq [k]$, with $|\mathfrak{l}| > 1$, and $q \in \mathrm{ri}\,\Delta_{\mathfrak{l}}$. Then

$$\begin{aligned}
\mathrm{Mix}_{\Phi_{\mathfrak{l}}}(\Pi_{\mathfrak{l}} d, \Pi_{\mathfrak{l}} q) &= \inf_{\pi \in \Delta_{|\mathfrak{l}|}} \langle \pi, \Pi_{\mathfrak{l}} d \rangle + D_{\Phi_{\mathfrak{l}}}(\pi, \Pi_{\mathfrak{l}} q), \\
&= \inf_{\mu \in \Delta_{\mathfrak{l}}} \langle \mu, d \rangle + D_\Phi(\mu, q), \\
&\leq \inf_{\mu \in \Delta_k} \langle \mu, d \rangle + D_\Phi(\mu, q), \qquad (19) \\
&= \mathrm{Mix}_\Phi(d, q).
\end{aligned}$$

To complete the proof, we need to show that (19) holds with equality. For this, it suffices to prove that $\forall \mu \in \Delta_k \setminus \Delta_{\mathfrak{l}}, D_\Phi(\mu, q) = +\infty$. This follows from Corollary 12. $\square$

**Lemma 14.** *Let $\Phi \colon \mathbb{R}^k \to \mathbb{R} \cup \{+\infty\}$ be an entropy satisfying (10) for all $\mathfrak{l} \subseteq [k]$ such that $|\mathfrak{l}| > 1$. Let $x \in [n], d \in \mathbb{R}^k$, and $q \in \Delta_k$. The infimum in*

$$\mathrm{Mix}_\Phi(d, q) = \inf_{\mu \in \Delta_k} \langle \mu, d \rangle + D_\Phi(\mu, q) \qquad (20)$$

*is attained at some $q_* \in \Delta_k$. Furthermore, if $q \in \mathrm{ri}\,\Delta_k$ and $q_*$ is the infimum of (20) then for any $s_q^* \in \mathrm{argmax}\{\langle s, \tilde{q}_* - \tilde{q} \rangle : s \in \partial\tilde{\Phi}(\tilde{q})\}$, we have*

$$\tilde{q}_* \in \partial\tilde{\Phi}^*(s_q^* - J_k^\mathsf{T} d), \qquad (21)$$

$$\mathrm{Mix}_\Phi(d, q) = d_k + \tilde{\Phi}^*(s_q^*) - \tilde{\Phi}^*(s_q^* - J_k^\mathsf{T} d). \qquad (22)$$

*Proof.* Let $q \in \mathrm{ri}\,\Delta_k$. Since $\tilde{q} \in \mathrm{int}\,\mathrm{dom}\,\tilde{\Phi} = \mathrm{int}\,\tilde{\Delta}_k$, the function $\tilde{\mu} \mapsto -\tilde{\Phi}'(\tilde{q}; \tilde{\mu} - \tilde{q})$ is lower semicontinuous [11, Cor. 24.5.1]. Given that $\tilde{\mu} \mapsto \langle \mathrm{II}_k(\tilde{\mu}), d \rangle + \tilde{\Phi}(\tilde{\mu}) - \tilde{\Phi}(\tilde{q})$ is a closed convex function, it is also lower semi-continuous. Therefore, the function

$$\tilde{\mu} \mapsto \langle \mathrm{II}_k(\tilde{\mu}), d \rangle + \tilde{\Phi}(\tilde{\mu}) - \tilde{\Phi}(\tilde{q}) - \tilde{\Phi}'(\tilde{q}; \tilde{\mu} - \tilde{q})$$

is lower semicontinuous, and thus attains its minimum on the compact set $\tilde{\Delta}_k$ at some point $\tilde{q}_*$. Using the fact that $D_\Phi(\mu, q) = D_{\tilde{\Phi}}(\tilde{\mu}, \tilde{q})$, we get that

$$q_* := \mathrm{II}_k(\tilde{q}_*) = \underset{\mu \in \Delta_k}{\mathrm{argmin}}\langle \mu, d \rangle + D_\Phi(\mu, q). \qquad (23)$$

If $q \in \mathrm{rbd}\,\Delta_k$, then either $q$ is a vertex of $\Delta_k$ or there exists $\mathfrak{l} \subsetneq [k]$ such that $q \in \mathrm{ri}\,\Delta_{\mathfrak{l}}$. In the former case, it follows from (10) that $D_\Phi(\mu, q) = +\infty$ for all $\mu \in \Delta_k \setminus \{q\}$, and thus the infimum

of (20) is trivially attained at $\boldsymbol{\mu} = \boldsymbol{q}$. Now consider the alternative — $\boldsymbol{q} \in \Delta_{\mathfrak{l}}$ with $|\mathfrak{l}| > 1$. Using Corollary 12, we have $D_\Phi(\boldsymbol{\mu}, \boldsymbol{q}) = +\infty$ for all $\boldsymbol{\mu} \in \Delta_k \setminus \Delta_{\mathfrak{l}}$. Therefore,

$$\begin{aligned}\mathrm{Mix}_\Phi(\boldsymbol{d}, \boldsymbol{q}) &= \inf_{\boldsymbol{\mu} \in \Delta_{\mathfrak{l}}} \langle \boldsymbol{\mu}, \boldsymbol{d} \rangle + D_\Phi(\boldsymbol{\mu}, \boldsymbol{q}), \\ &= \inf_{\hat{\boldsymbol{\mu}} \in \Delta_{|\mathfrak{l}|}} \langle \hat{\boldsymbol{\mu}}, \Pi_{\mathfrak{l}} \boldsymbol{d} \rangle + D_{\Phi_{\mathfrak{l}}}(\hat{\boldsymbol{\mu}}, \Pi_{\mathfrak{l}} \boldsymbol{q}), \end{aligned} \tag{24}$$

where $\Phi_{\mathfrak{l}} := \Phi \circ \Pi_{\mathfrak{l}}$. Since $\Pi_{\mathfrak{l}} \boldsymbol{q} \in \mathrm{ri}\,\Delta_{|\mathfrak{l}|}$, we can use the same argument as the previous paragraph with $\Phi$ and $\boldsymbol{q}$ replaced by $\Phi_{\mathfrak{l}}$ and $\Pi_{\mathfrak{l}} \boldsymbol{q}$, respectively, to show that the infimum in (24) is attained at some $\hat{\boldsymbol{q}}_* \in \Delta_{|\mathfrak{l}|}$. Thus, $\boldsymbol{q}_* := \Pi_{\mathfrak{l}}^\mathsf{T} \hat{\boldsymbol{q}} \in \Delta_k$ attains the infimum in (20).

Now we show the second part of the lemma. Let $\boldsymbol{q} \in \mathrm{ri}\,\Delta_k$ and $\boldsymbol{q}_*$ be the infimum of (20). Since $\tilde{\Phi}$ is convex and $\tilde{\boldsymbol{q}} = \Pi_k(\boldsymbol{q}) \in \mathrm{int}\,\tilde{\Delta}_k = \mathrm{int}\,\mathrm{dom}\,\tilde{\Phi}$, we have $\partial \tilde{\Phi}(\tilde{\boldsymbol{q}}) \neq \varnothing$ [11, Thm. 23.4]. This means that there exists $\boldsymbol{s}_{\boldsymbol{q}}^* \in \partial \tilde{\Phi}(\tilde{\boldsymbol{q}})$ such that $\langle \boldsymbol{s}_{\boldsymbol{q}}^*, \tilde{\boldsymbol{q}}_* - \tilde{\boldsymbol{q}} \rangle = \tilde{\Phi}'(\tilde{\boldsymbol{q}}; \tilde{\boldsymbol{q}}_* - \tilde{\boldsymbol{q}})$ [7, p.166]. We will now show that $\boldsymbol{s}_{\boldsymbol{q}}^* - J_k^\mathsf{T} \boldsymbol{d} \in \partial \tilde{\Phi}(\tilde{\boldsymbol{q}}_*)$, which will imply that $\tilde{\boldsymbol{q}}_* \in \partial \tilde{\Phi}^*(\boldsymbol{s}_{\boldsymbol{q}}^* - J_k^\mathsf{T} \boldsymbol{d})$ (ibid., Cor. D.1.4.4). Let $\boldsymbol{q}_* = \mathrm{argmin}_{\boldsymbol{\mu} \in \Delta_k} \langle \boldsymbol{\mu}, \boldsymbol{d} \rangle + D_\Phi(\boldsymbol{\mu}, \boldsymbol{q})$. Thus, for all $\boldsymbol{\mu} \in \Delta_k$,

$$\begin{aligned}&\langle \boldsymbol{\mu}, \boldsymbol{d} \rangle + \tilde{\Phi}(\tilde{\boldsymbol{\mu}}) - \tilde{\Phi}(\tilde{\boldsymbol{q}}) - \tilde{\Phi}'(\tilde{\boldsymbol{q}}; \tilde{\boldsymbol{\mu}} - \tilde{\boldsymbol{q}}) \geq \langle \boldsymbol{q}_*, \boldsymbol{d} \rangle + \tilde{\Phi}(\tilde{\boldsymbol{q}}_*) - \tilde{\Phi}(\tilde{\boldsymbol{q}}) - \langle \boldsymbol{s}_{\boldsymbol{q}}^*, \tilde{\boldsymbol{q}}_* - \tilde{\boldsymbol{q}} \rangle, \\ \implies \quad &\tilde{\Phi}(\tilde{\boldsymbol{\mu}}) \geq \tilde{\Phi}(\tilde{\boldsymbol{q}}_*) - \langle \tilde{\boldsymbol{\mu}} - \tilde{\boldsymbol{q}}_*, J_k^\mathsf{T} \boldsymbol{d} \rangle + \langle \boldsymbol{s}_{\boldsymbol{q}}^*, \tilde{\boldsymbol{q}} - \tilde{\boldsymbol{q}}_* \rangle + \Phi'(\tilde{\boldsymbol{q}}; \tilde{\boldsymbol{\mu}} - \tilde{\boldsymbol{q}}), \\ \implies \quad &\tilde{\Phi}(\tilde{\boldsymbol{\mu}}) \geq \tilde{\Phi}(\tilde{\boldsymbol{q}}_*) - \langle \tilde{\boldsymbol{\mu}} - \tilde{\boldsymbol{q}}_*, J_k^\mathsf{T} \boldsymbol{d} \rangle + \langle \boldsymbol{s}_{\boldsymbol{q}}^*, \tilde{\boldsymbol{q}} - \tilde{\boldsymbol{q}}_* \rangle + \langle \boldsymbol{s}_{\boldsymbol{q}}^*, \tilde{\boldsymbol{\mu}} - \tilde{\boldsymbol{q}} \rangle, \\ \implies \quad &\tilde{\Phi}(\tilde{\boldsymbol{\mu}}) \geq \tilde{\Phi}(\tilde{\boldsymbol{q}}_*) + \langle \tilde{\boldsymbol{\mu}} - \tilde{\boldsymbol{q}}_*, \boldsymbol{s}_{\boldsymbol{q}}^* - J_k^\mathsf{T} \boldsymbol{d} \rangle, \end{aligned}$$

where in the second line we used the fact that $\forall \boldsymbol{q} \in \Delta_k, \langle \boldsymbol{q}, \boldsymbol{d} \rangle = \langle \tilde{\boldsymbol{q}}, J_k^\mathsf{T} \boldsymbol{d} \rangle + d_k$, and in third line we used the fact that $\forall \boldsymbol{s} \in \partial \tilde{\Phi}(\tilde{\boldsymbol{q}})$, $\langle \boldsymbol{s}, \tilde{\boldsymbol{\mu}} - \tilde{\boldsymbol{q}} \rangle \leq \tilde{\Phi}'(\tilde{\boldsymbol{q}}; \tilde{\boldsymbol{\mu}} - \tilde{\boldsymbol{q}})$ (ibid.). This shows that $\boldsymbol{s}_{\boldsymbol{q}}^* - J_k^\mathsf{T} \boldsymbol{d} \in \partial \tilde{\Phi}(\tilde{\boldsymbol{q}}_*)$.

Substituting $\tilde{\Phi}'(\tilde{\boldsymbol{q}}; \tilde{\boldsymbol{q}}_* - \tilde{\boldsymbol{q}})$ by $\langle \boldsymbol{s}_{\boldsymbol{q}}^*, \boldsymbol{q}_* - \boldsymbol{q} \rangle$ in the expression for $\mathrm{Mix}_\Phi(\boldsymbol{d}, \boldsymbol{q})$, we get

$$\begin{aligned}\mathrm{Mix}_\Phi(\boldsymbol{d}, \boldsymbol{q}) &= d_k + \langle \tilde{\boldsymbol{q}}_*, J_k^\mathsf{T} \boldsymbol{d} \rangle + \tilde{\Phi}(\tilde{\boldsymbol{q}}_*) - \tilde{\Phi}(\tilde{\boldsymbol{q}}) - \langle \boldsymbol{s}_{\boldsymbol{q}}^*, \tilde{\boldsymbol{q}}_* - \tilde{\boldsymbol{q}} \rangle, \\ &= d_k + \langle \boldsymbol{s}_{\boldsymbol{q}}^*, \tilde{\boldsymbol{q}} \rangle - \tilde{\Phi}(\tilde{\boldsymbol{q}}) - [\langle \boldsymbol{s}_{\boldsymbol{q}}^* - J_k^\mathsf{T} \boldsymbol{d}, \tilde{\boldsymbol{q}}_* \rangle - \tilde{\Phi}(\tilde{\boldsymbol{q}}_*)], \\ &= d_k + \tilde{\Phi}^*(\boldsymbol{s}_{\boldsymbol{q}}^*) - \tilde{\Phi}^*(\boldsymbol{s}_{\boldsymbol{q}}^* - J_k^\mathsf{T} \boldsymbol{d}), \end{aligned}$$

where in the last line we used the fact that $\tilde{\Phi}$ is a closed convex function, and thus $\forall \tilde{\boldsymbol{q}} \in \tilde{\Delta}_k$, $\boldsymbol{s} \in \partial \tilde{\Phi}(\tilde{\boldsymbol{q}}) \implies \tilde{\Phi}^*(\boldsymbol{s}) = \langle \boldsymbol{s}, \tilde{\boldsymbol{q}} \rangle - \tilde{\Phi}(\tilde{\boldsymbol{q}})$ (ibid., Cor. E.1.4.4).

$\square$

**Lemma 15.** *Let $\boldsymbol{q} \in \Delta_k$. For any sequence $(\boldsymbol{d}_m)$ in $[0, +\infty[^k$ converging to $\boldsymbol{d} \in [0, +\infty]^k$ coordinate-wise and any entropy $\Phi \colon \mathbb{R}^k \to \mathbb{R} \cup \{+\infty\}$ satisfying (10) for $\mathfrak{l} \subseteq [k]$ such that $|\mathfrak{l}| > 1$,*

$$\lim_{m \to \infty} \mathrm{Mix}_\Phi(\boldsymbol{d}_m, \boldsymbol{q}) = \mathrm{Mix}_\Phi(\boldsymbol{d}, \boldsymbol{q}). \tag{25}$$

***Proof of Lemma 15.*** Let $\boldsymbol{q} \in \Delta_k$ and $\Phi \colon \mathbb{R}^k \to \mathbb{R} \cup \{+\infty\}$ be an entropy as in the statement of the Lemma. Let $(\boldsymbol{d}_m) \subset \mathbb{R}^k$ such that $\boldsymbol{d}_m \overset{m \to \infty}{\to} \boldsymbol{d} \in \mathbb{R}^k$. in $[0, +\infty[^k$. Let $\mathfrak{l} := \{\theta \in [k] : d_\theta < +\infty\}$. If $\mathfrak{l} = \varnothing$ then the result holds trivially since, on the one hand, $\mathrm{Mix}_\Phi(\boldsymbol{d}, \boldsymbol{q}) = +\infty$ and on the other hand $\mathrm{Mix}_\Phi(\boldsymbol{d}_m, \boldsymbol{q}) \geq \min_{\theta \in [k]} d_{m,\theta} \overset{m \to \infty}{\to} +\infty$.

Assume now that $\mathfrak{l} \neq \varnothing$. Then

$$\mathrm{Mix}_\Phi(\boldsymbol{d}_m, \boldsymbol{q}) = \inf_{\boldsymbol{\mu} \in \Delta_k} \langle \boldsymbol{\mu}, \boldsymbol{d}_m \rangle + D_\Phi(\boldsymbol{\mu}, \boldsymbol{q}), \tag{26}$$

$$\leq \inf_{\hat{\boldsymbol{\mu}} \in \Delta_{\mathfrak{l}}} \langle \hat{\boldsymbol{\mu}}, \boldsymbol{d} \rangle + D_\Phi(\hat{\boldsymbol{\mu}}, \boldsymbol{q}), \tag{27}$$

$$< +\infty, \tag{28}$$

where the last inequality stems from the fact that $\Pi_{\mathfrak{l}} \boldsymbol{d}_m$ is a finite vector in $\mathbb{R}^{|\mathfrak{l}|}$. Therefore, (28) implies that the sequence $\alpha_m := \mathrm{Mix}_\Phi(\boldsymbol{d}_m, \boldsymbol{q})$ is bounded. We will show that $(\alpha_m)$ converges in $\mathbb{R}$ and that its limit is exactly $\mathrm{Mix}_\Phi(\boldsymbol{d}, \boldsymbol{q})$. Let $(\hat{\alpha}_m)$ be any convergent subsequence of $(\alpha_m)$, and let

$(\hat{\boldsymbol{d}}_m)$ be the corresponding subsequence of $(\boldsymbol{d}_m)$. Consider the infimum in (100) with $\boldsymbol{d}_m$ is replaced by $\hat{\boldsymbol{d}}_m$. From Lemma 14, this infimum is attained at some $\boldsymbol{q}_m \in \Delta_k$. Since $\Delta_k$ is compact, we may assume without loss of generality that $\boldsymbol{q}_m$ converges to some $\bar{\boldsymbol{q}} \in \Delta_k$. Observe that $\bar{\boldsymbol{q}}$ must be $\Delta_{\mathfrak{l}}$; suppose that $\exists \theta_* \in \bar{\mathfrak{l}}$ such that $\bar{q}_{\theta_*} > 0$. Then

$$\hat{\alpha}_m \geq \langle \boldsymbol{q}_m, \hat{\boldsymbol{d}}_m \rangle,$$
$$\geq q_{m,\theta_*} \hat{d}_{m,\theta_*} \overset{m\to\infty}{\to} +\infty.$$

This would contradict the fact that $\alpha_m$ is bounded, and thus $\bar{\boldsymbol{q}} \in \Delta_{\mathfrak{l}}$. Using this, we get

$$\mathrm{Mix}_\Phi(\hat{\boldsymbol{d}}_m, \boldsymbol{q}) = \langle \boldsymbol{q}_m, \hat{\boldsymbol{d}}_m \rangle + D_\Phi(\boldsymbol{q}_m, \boldsymbol{q}),$$
$$\geq \langle \Pi_{\mathfrak{l}} \boldsymbol{q}_m, \Pi_{\mathfrak{l}} \hat{\boldsymbol{d}}_m \rangle + D_\Phi(\boldsymbol{q}_m, \boldsymbol{q}),$$
$$\overset{m\to\infty}{\to} \langle \Pi_{\mathfrak{l}} \bar{\boldsymbol{q}}, \Pi_{\mathfrak{l}} \boldsymbol{d} \rangle + D_\Phi(\bar{\boldsymbol{q}}, \boldsymbol{q}),$$
$$= \langle \bar{\boldsymbol{q}}, \boldsymbol{d} \rangle + D_\Phi(\bar{\boldsymbol{q}}, \boldsymbol{q}), \tag{29}$$
$$\geq \inf_{\hat{\boldsymbol{\mu}} \in \Delta_{\mathfrak{l}}} \langle \hat{\boldsymbol{\mu}}, \boldsymbol{d} \rangle + D_\Phi(\hat{\boldsymbol{\mu}}, \boldsymbol{q}). \tag{30}$$

where in (29) we use the fact that $\bar{\boldsymbol{q}} \in \Delta_{\mathfrak{l}}$. Combining (30) with (27) shows that $\hat{\alpha}_m$ converges to $\mathrm{Mix}_\Phi(\boldsymbol{d}, \boldsymbol{q}) = \inf_{\hat{\boldsymbol{\mu}} \in \Delta_{\mathfrak{l}}} \langle \hat{\boldsymbol{\mu}}, \boldsymbol{d} \rangle + D_\Phi(\hat{\boldsymbol{\mu}}, \boldsymbol{q})$. Since $(\hat{\alpha}_m)$ was any convergent subsequence of $(\alpha_m)$ (which is bounded), the result follows. $\qquad\square$

## C  Proofs of Results in the Main Body

### C.1  Proof of Theorem 4

**Theorem 4** *Any loss $\ell \colon \mathcal{A} \to [0, +\infty]^n$ such that $\mathrm{dom}\,\ell \neq \varnothing$, has a proper support loss $\underline{\ell}$ with the same Bayes risk, $\underline{L}_\ell$, as $\ell$.*

*Proof.* We will construct a proper support loss $\underline{\ell}$ of $\ell$.

Let $\boldsymbol{p} \in \mathrm{ri}\,\Delta_n$ ($-\boldsymbol{p} \in \mathrm{int}\,\mathrm{dom}\,\sigma_{\mathscr{S}_\ell}$). Since the support function of a non-empty set is closed and convex, we have $\sigma_{\mathscr{S}_\ell}^{**} = \sigma_{\mathscr{S}_\ell}$ [7, Prop. C.2.1.2]. Pick any $\boldsymbol{v} \in \partial\sigma_{\mathscr{S}_\ell}(-\boldsymbol{p}) = \partial\sigma_{\mathscr{S}_\ell}^{**}(-\boldsymbol{p}) \neq \varnothing$. Since $\sigma_{\mathscr{S}_\ell}^* = \iota_{\mathscr{S}_\ell}$ [11], we can apply Proposition 1-(iv) with $f$ replaced by $\sigma_{\mathscr{S}_\ell}^*$ to obtain $\langle -\boldsymbol{p}, \boldsymbol{v} \rangle = \sigma_{\mathscr{S}_\ell}(-\boldsymbol{p}) + \iota_{\mathscr{S}_\ell}(\boldsymbol{v})$. The fact that $\langle -\boldsymbol{p}, \boldsymbol{v} \rangle$ and $\sigma_{\mathscr{S}_\ell}(-\boldsymbol{p})$ are both finite implies that $\iota_{\mathscr{S}_\ell}(\boldsymbol{v}) = 0$. Therefore, $\boldsymbol{v} \in \mathscr{S}_\ell$ and $\langle \boldsymbol{p}, \boldsymbol{v} \rangle = -\sigma_{\mathscr{S}_\ell}(-\boldsymbol{p}) = \underline{L}_\ell(\boldsymbol{p})$. Define $\underline{\ell}(\boldsymbol{p}) := \boldsymbol{v} \in \mathscr{S}_\ell$.

Now let $\boldsymbol{p} \in \mathrm{rbd}\,\Delta_n$ and $\boldsymbol{q} := \mathbf{1}_n/n$. Since the $\underline{L}_\ell$ is a closed concave function and $\boldsymbol{q} \in \mathrm{int}\,\mathrm{dom}\,\underline{L}_\ell$, it follows that $\underline{L}_\ell(\boldsymbol{p} + m^{-1}(\boldsymbol{q} - \boldsymbol{p})) \overset{m\to\infty}{\to} \underline{L}_\ell(\boldsymbol{p})$ [7, Prop. B.1.2.5]. Note that $\boldsymbol{q}_m := \boldsymbol{p} + m^{-1}(\boldsymbol{q} - \boldsymbol{p}) \in \mathrm{ri}\,\Delta_n, \forall m \in \mathbb{N}$. Now let $v_{x,m} := \underline{\ell}_x(\boldsymbol{q}_m)$, where $\underline{\ell}(\boldsymbol{q}_m)$ is as constructed in the previous paragraph. If $(v_{1,m})$ is bounded [resp. unbounded], we can extract a subsequence $(v_{1,\varphi_1(m)})$ which converges [resp. diverges to $+\infty$], where $\varphi_1 : \mathbb{N} \to \mathbb{N}$ is an increasing function. By repeating this process for $(v_{2,\varphi_1(m)})$ and so on, we can construct an increasing function $\varphi := \varphi_n \circ \cdots \circ \varphi_1 : \mathbb{N} \to \mathbb{N}$, such that $\boldsymbol{v}_m := [v_{x,\varphi(m)}]_{x \in [n]}^{\mathsf{T}}$ has a well defined (coordinate-wise) limit in $[0, +\infty]^n$. Define $\underline{\ell}(\boldsymbol{p}) := \lim_{m\to\infty} \boldsymbol{v}_m$. By continuity of the inner product, we have

$$\langle \boldsymbol{p}, \underline{\ell}(\boldsymbol{p}) \rangle = \lim_{m\to\infty} \langle \boldsymbol{q}_{\varphi(m)}, \boldsymbol{v}_m \rangle = \lim_{m\to\infty} \langle \boldsymbol{q}_{\varphi(m)}, \underline{\ell}(\boldsymbol{q}_{\varphi(m)}) \rangle,$$
$$= \lim_{m\to\infty} \underline{L}_\ell(\boldsymbol{q}_{\varphi(m)}) = \underline{L}_\ell(\boldsymbol{p}).$$

By construction, $\forall m \in \mathbb{N}, \boldsymbol{p}_m := \boldsymbol{q}_{\varphi(m)} \in \mathrm{ri}\,\Delta_n$ and $\underline{\ell}(\boldsymbol{p}_m) = \boldsymbol{v}_m \overset{m\to\infty}{\to} \underline{\ell}(\boldsymbol{p})$. Therefore, $\underline{\ell}$ is support loss of $\ell$.

It remains to show that it is proper; that is $\forall \boldsymbol{p} \in \Delta_n, \forall \boldsymbol{q} \in \Delta_n, \langle \boldsymbol{p}, \underline{\ell}(\boldsymbol{p}) \rangle \leq \langle \boldsymbol{p}, \underline{\ell}(\boldsymbol{q}) \rangle$. Let $\boldsymbol{q} \in \mathrm{ri}\,\Delta_n$. We just showed that $\forall \boldsymbol{p} \in \Delta_n, \langle \boldsymbol{p}, \underline{\ell}(\boldsymbol{p}) \rangle = \underline{L}_\ell(\boldsymbol{p})$ and that $\underline{\ell}(\boldsymbol{q}) \in \mathscr{S}_\ell$. Using the fact that $\underline{L}_\ell(\boldsymbol{p}) = \inf_{\boldsymbol{z} \in \mathscr{S}_\ell} \langle \boldsymbol{p}, \boldsymbol{z} \rangle$, we obtain $\langle \boldsymbol{p}, \underline{\ell}(\boldsymbol{p}) \rangle \leq \langle \boldsymbol{p}, \underline{\ell}(\boldsymbol{q}) \rangle$.

Now let $\boldsymbol{q} \in \mathrm{rbd}\,\Delta_k$. Since $\underline{\ell}$ is a support loss, we know that there exists a sequence $(\boldsymbol{q}_m) \subset \mathrm{ri}\,\Delta_n$ such that $\underline{\ell}(\boldsymbol{q}_m) \overset{m\to\infty}{\to} \underline{\ell}(\boldsymbol{q})$. But as we established in the previous paragraph, $\langle \boldsymbol{p}, \underline{\ell}(\boldsymbol{p}) \rangle \leq \langle \boldsymbol{p}, \underline{\ell}(\boldsymbol{q}_m) \rangle$. By passing to the limit $m \to \infty$, we obtain $\langle \boldsymbol{p}, \underline{\ell}(\boldsymbol{p}) \rangle \leq \langle \boldsymbol{p}, \underline{\ell}(\boldsymbol{q}) \rangle$. Therefore $\underline{\ell}$ is a proper loss with Bayes risk $\underline{L}_\ell$. $\qquad\square$

## C.2 Proofs of Theorem 5 and Proposition 12

For a set $\mathcal{C}$, we denote $\mathrm{co}\,\mathcal{C}$ and $\overline{\mathrm{co}}\mathcal{C}$ its *convex hull* and *closed convex hull*, respectively.

**Definition 16** ([7]). *Let $\mathcal{C}$ be non-empty convex set in $\mathbb{R}^n$. We say that $\boldsymbol{u} \in \mathcal{C}$ is an extreme point of $\mathcal{C}$ if there are no two different points $\boldsymbol{u}_1$ and $\boldsymbol{u_2}$ in $\mathcal{C}$ and $\lambda \in ]0, 1[$ such that $\boldsymbol{u} = \lambda \boldsymbol{u}_1 + (1 - \lambda)\boldsymbol{u}_2$.*

We denote the set of extreme points of a set $\mathcal{C}$ by $\mathrm{ext}\,\mathcal{C}$.

**Lemma 17.** *Let $\ell \colon \mathcal{A} \to [0, +\infty]^n$ be a closed loss. Then $\mathrm{ext}\,\overline{\mathrm{co}}\mathscr{S}_\ell \subseteq \mathcal{S}_\ell$.*

*Proof.* Since $\mathrm{co}\,\mathscr{S}_\ell \subseteq \mathbb{R}^n$ is connected, $\mathrm{co}\,\mathscr{S}_\ell = \{\boldsymbol{v} + \sum_{k=1}^n \alpha_k \ell(\boldsymbol{a}_k) \colon (\boldsymbol{a}_{k \in [n]}, \boldsymbol{\alpha}, \boldsymbol{v}) \in \mathcal{A}^n \times \Delta_n \times [0, +\infty[^n\}$ [7, Prop. A.1.3.7].

We claim that $\overline{\mathrm{co}}\mathscr{S}_\ell = \mathrm{co}\,\mathscr{S}_\ell$. Let $(\boldsymbol{z}_m) \coloneqq (\boldsymbol{v}_m + \sum_{k=1}^n \alpha_{m,k} \ell(\boldsymbol{a}_{m,k}))$ be a convergent sequence in $[0, +\infty[^n$, where $(\boldsymbol{\alpha}_m)$, $([\boldsymbol{a}_{m,k}]_{k \in [n]})$ and $(\boldsymbol{v}_m)$ are sequences in $\Delta_n$, $\mathcal{A}^n$, and $[0, +\infty[^n$, respectively. Since $\Delta_n$ is compact, we may assume, by extracting a subsequence if necessary, that $\boldsymbol{\alpha}_m \overset{m \to \infty}{\to} \boldsymbol{\alpha}^* \in \Delta_n$. Let $\mathcal{K} \coloneqq \{k \in [n] \colon \alpha_k^* \neq 0\}$. Since $\boldsymbol{z}_m$ converges, $([[\ell(\boldsymbol{a}_{m,k})]_{k \in \mathcal{K}}, \boldsymbol{v}_m])$ is a bounded sequence in $[0, +\infty[^{n|\mathcal{K}|+n}$. Since $\ell$ is closed, we may assume, by extraction a subsequence if necessary, that $\forall k \in \mathcal{K}$, $\ell(\boldsymbol{a}_{m,k}) \overset{m \to \infty}{\to} \ell(\boldsymbol{a}_k^*)$ and $\boldsymbol{v}_m \overset{m \to \infty}{\to} \boldsymbol{v}^*$, where $[\boldsymbol{a}_k^*]_{k \in \mathcal{K}} \in \mathcal{A}^{|\mathcal{K}|}$ and $\boldsymbol{v}^* \in [0, +\infty[^n$. Consequently,

$$\boldsymbol{v}^* + \sum_{k=1}^n \alpha_k^* \ell(\boldsymbol{a}_k^*) = \lim_{m \to \infty} \left[ \boldsymbol{v}_{m,k} + \sum_{k \in \mathcal{K}} \alpha_{m,k} \ell(\boldsymbol{a}_{m,k}) \right],$$
$$\leq \lim_{m \to \infty} \boldsymbol{z}_m,$$

where the last inequality is coordinate-wise. Therefore, there exists $\boldsymbol{v}' \in [0, +\infty[^n$ such that $\lim_{m \to \infty} \boldsymbol{z}_m = \boldsymbol{v}' + \boldsymbol{v}^* + \sum_{k=1}^n \alpha_k^* \ell(\boldsymbol{a}_k^*) \in \mathrm{co}\,\mathscr{S}_\ell$. This shows that $\overline{\mathrm{co}}\mathscr{S}_\ell \subset \mathrm{co}\,\mathscr{S}_\ell$, and thus $\overline{\mathrm{co}}\mathscr{S}_\ell = \mathrm{co}\,\mathscr{S}_\ell$ which proves our first claim.

By definition of an extreme point, $\mathrm{ext}\,\overline{\mathrm{co}}\mathscr{S}_\ell \subseteq \overline{\mathrm{co}}\mathscr{S}_\ell$. Let $\boldsymbol{e} \in \mathrm{ext}\,\overline{\mathrm{co}}\mathscr{S}_\ell$ and $(\boldsymbol{a}_{k \in [n]}, \boldsymbol{\alpha}, \boldsymbol{v}) \in \mathcal{A}^n \times \Delta_n \times [0, +\infty[^n$ such that $\boldsymbol{e} = \sum_{k=1}^n \alpha_k \ell(\boldsymbol{a}_k) + \boldsymbol{v}$. If there exists $i, j \in [n]$ such that $\alpha_i \alpha_j \neq 0$ or $\alpha_i v_j \neq 0$ then $\boldsymbol{e}$ would violate the definition of an extreme point. Therefore, the only possible extreme points are of the form $\{\ell(\boldsymbol{a}) : \boldsymbol{a} \in \mathrm{dom}\,\ell\} = \mathcal{S}_\ell$. $\qquad\square$

**Theorem 5** *Let $\ell \colon \mathcal{A} \to [0, +\infty]^n$ be a loss and $\underline{\ell}$ be a proper support loss of $\ell$. If the Bayes risk $\underline{L}_\ell$ is differentiable on $]0, +\infty[^n$, then $\underline{\ell}$ is uniquely defined on $\mathrm{ri}\,\Delta_n$ and*

$$\forall \boldsymbol{p} \in \mathrm{dom}\,\underline{\ell}, \quad \exists \boldsymbol{a}_* \in \mathrm{dom}\,\ell, \quad \ell(\boldsymbol{a}_*) = \underline{\ell}(\boldsymbol{p}),$$
$$\forall \boldsymbol{a} \in \mathrm{dom}\,\ell, \quad \exists (\boldsymbol{p}_m) \subset \mathrm{ri}\,\Delta_n, \quad \underline{\ell}(\boldsymbol{p}_m) \overset{m \to \infty}{\to} \ell(\boldsymbol{a}) \text{ coordinate-wise.}$$

*Proof.* Let $\boldsymbol{p} \in \mathrm{ri}\,\Delta_n$ and suppose that $\underline{L}_\ell$ is differentiable at $\boldsymbol{p}$. In this case, $\sigma_{\mathscr{S}_\ell}$ is differentiable at $-\boldsymbol{p}$, which implies [7, Cor. D.2.1.4]

$$\mathcal{F}(\boldsymbol{p}) \coloneqq \partial \sigma_{\mathscr{S}_\ell}(-\boldsymbol{p}) = \{\nabla \sigma_{\mathscr{S}_\ell}(-\boldsymbol{p})\}. \tag{31}$$

On the other hand, the fact that $\sigma_{\mathscr{S}_\ell} = \sigma_{\overline{\mathrm{co}}\mathscr{S}_\ell}$ [7, Prop. C.2.2.1], implies $\mathcal{F}(\boldsymbol{p}) = \partial \sigma_{\mathscr{S}_\ell}(-\boldsymbol{p}) = \partial \sigma_{\overline{\mathrm{co}}\mathscr{S}_\ell}(-\boldsymbol{p})$. The latter being an *exposed face* of $\overline{\mathrm{co}}\mathscr{S}_\ell$ implies that every extreme point of $\mathcal{F}(\boldsymbol{p})$ is also an extreme point of $\overline{\mathrm{co}}\mathscr{S}_\ell$ [7, Prop. A.2.3.7, Prop. A.2.4.3]. Therefore, from (31), $\underline{\ell}(\boldsymbol{p}) = \nabla \sigma_{\mathscr{S}_\ell}(-\boldsymbol{p})$ is the only extreme point of $\mathcal{F}(\boldsymbol{p}) \subset \overline{\mathrm{co}}\mathscr{S}_\ell$. From Lemma 17, there exists $\boldsymbol{a}_* \in \mathcal{A}$ such that $\ell(\boldsymbol{a_*}) = \underline{\ell}(\boldsymbol{p})$. In this paragraph, we showed the following

$$\forall \boldsymbol{p} \in \mathrm{ri}\,\Delta_n, \exists \boldsymbol{a}_* \in \mathrm{dom}\,\ell, \ \ell(\boldsymbol{a}_*) = \underline{\ell}(\boldsymbol{p}). \tag{32}$$

For the rest of this proof we will assume that $\underline{L}_\ell$ is differentiable on $]0, +\infty[^n$. Let $\boldsymbol{p} \in \mathrm{rbd}\,\Delta_n \cap \mathrm{dom}\,\underline{\ell}$. Since $\underline{\ell}$ is a support loss, there exists $(\boldsymbol{p}_m)$ in $\mathrm{ri}\,\Delta_n$ such that $(\underline{\ell}(\boldsymbol{p}_m))_m$ converges to $\underline{\ell}(\boldsymbol{p})$. From (32) it holds that $\forall \boldsymbol{p}_m \in \mathrm{ri}\,\Delta_n, \exists \boldsymbol{a}_m \in \mathcal{A}, \ell(\boldsymbol{a}_m) = \underline{\ell}(\boldsymbol{p}_m)$. Since $(\ell(\boldsymbol{a}_m))_m$ converges and $\ell$ is closed, there exists $\boldsymbol{a}_* \in \mathcal{A}$ such that $\ell(\boldsymbol{a}_*) = \lim_{m \to \infty} \ell(\boldsymbol{a}_m) = \underline{\ell}(\boldsymbol{p})$.

Now let $\boldsymbol{a} \in \mathrm{dom}\,\ell$ and $f(\boldsymbol{p}, x) \coloneqq \underline{\ell}_x(\boldsymbol{p}) - \ell_x(\boldsymbol{a})$. Since $\ell(\boldsymbol{a}) \in \mathscr{S}_\ell$ and $\underline{\ell}$ is proper, we have for all $\boldsymbol{p} \in \mathrm{ri}\,\Delta_n, \mathbb{E}_{x \sim \boldsymbol{p}}[f(\boldsymbol{p}, x)] \leq 0$ and $-\infty < f(\boldsymbol{p}, x), \forall x \in [n]$. Therefore, Lemma 5 implies that for

all $m \in \mathbb{N} \setminus \{0\}$ there exists $\boldsymbol{p}_m \in \operatorname{ri} \Delta_n$, such that $\forall x \in [n], \underline{\ell}_x(\boldsymbol{p}_m) \le \ell_x(\boldsymbol{a}) + 1/m$. On one hand, since $(\underline{\ell}(\boldsymbol{p}_m))$ is bounded (from the previous inequality), we may assume by extracting a subsequence if necessary, that $(\underline{\ell}(\boldsymbol{p}_m))_m$ converges. On the other hand, since $\boldsymbol{p}_m \in \operatorname{ri} \Delta_n$, (32) implies that there exists $\boldsymbol{a}_m \in \operatorname{dom} \ell$ such that $\underline{\ell}(\boldsymbol{p}_m) = \ell(\boldsymbol{a}_m)$. Since $\ell$ is closed and $(\ell(\boldsymbol{a}_m))_m$ converges, there exists $\boldsymbol{a}_* \in \mathcal{A}$, such that $\ell(\boldsymbol{a}_*) = \lim_{m \to \infty} \ell(\boldsymbol{a}_m) = \lim_{m \to \infty} \underline{\ell}(\boldsymbol{p}_m) \le \ell(\boldsymbol{a})$. But since $\ell$ is admissible, the latter component-wise inequality implies that $\ell(\boldsymbol{a}_*) = \ell(\boldsymbol{a}) = \lim_{m \to \infty} \underline{\ell}(\boldsymbol{p})$. $\qquad \square$

**Lemma 18.** *Let $\ell \colon \mathcal{A} \to [0, +\infty]^n$ be a loss satisfying Assumption 1. If $\underline{L}_\ell$ is not differentiable at $\boldsymbol{p}$ then there exist $\boldsymbol{a}_0, \boldsymbol{a}_1 \in \operatorname{dom} \ell$, such that $\ell(\boldsymbol{a}_0) \ne \ell(\boldsymbol{a}_1)$ and $\underline{L}_\ell(\boldsymbol{p}) = \langle \boldsymbol{p}, \ell(\boldsymbol{a}_0) \rangle = \langle \boldsymbol{p}, \ell(\boldsymbol{a}_1) \rangle$.*

*Proof.* Suppose $\underline{L}_\ell$ is not differentiable at $\boldsymbol{p} \in \operatorname{ri} \Delta_n$. Then from the definition of the Bayes risk, $\sigma_{\mathscr{S}_\ell}$ is not differentiable at $-\boldsymbol{p}$. This implies that $\mathcal{F}(\boldsymbol{p}) \coloneqq \partial \sigma_{\mathscr{S}_\ell}(-\boldsymbol{p})$ has more than one element [7, Cor. D.2.1.4]. Since $\sigma_{\mathscr{S}_\ell} = \sigma_{\overline{\operatorname{co}} \mathscr{S}_\ell}$ (ibid.. Prop. C.2.2.1), $\mathcal{F}(\boldsymbol{p}) = \partial \sigma_{\overline{\operatorname{co}} \mathscr{S}_\ell}(-\boldsymbol{p})$ is a subset of $\overline{\operatorname{co}} \mathscr{S}_\ell$ and every extreme point of $\mathcal{F}(\boldsymbol{p})$ is also an extreme point of $\overline{\operatorname{co}} \mathscr{S}_\ell$ (ibid., Prop. A.2.3.7). Thus, from Lemma 17, we have $\operatorname{ext} \mathcal{F}(\boldsymbol{p}) \subset \mathcal{S}_\ell$. On the other hand, since $-\boldsymbol{p} \in \operatorname{int} \operatorname{dom} \sigma_{\mathscr{S}_\ell}$, $\mathcal{F}(\boldsymbol{p})$ is a compact, convex set [11, Thm. 23.4], and thus $\mathcal{F}(\boldsymbol{p}) = \operatorname{co}(\operatorname{ext} \mathcal{F}(\boldsymbol{p}))$ [7, Thm. A.2.3.4]. Hence, the fact that $\mathcal{F}(\boldsymbol{p})$ has more than one element implies there exists $\boldsymbol{a}_0, \boldsymbol{a}_1 \in \mathcal{A}$ such that $\ell(\boldsymbol{a}_0), \ell(\boldsymbol{a}_1) \in \operatorname{ext} \mathcal{F}(\boldsymbol{p}) \subseteq \mathcal{F}(\boldsymbol{p})$ and $\ell(\boldsymbol{a}_0) \ne \ell(\boldsymbol{a}_1)$. Since $\mathcal{F}(\boldsymbol{p}) = \partial \sigma_{\mathscr{S}_\ell}(-\boldsymbol{p})$, Proposition 1-(iv) and the fact that $\sigma^*_{\mathscr{S}_\ell} = \iota_{\mathscr{S}_\ell}$ imply that $\underline{L}_\ell(\boldsymbol{p}) = \langle \boldsymbol{p}, \ell(\boldsymbol{a}_0) \rangle = \langle \boldsymbol{p}, \ell(\boldsymbol{a}_1) \rangle$. $\qquad \square$

**Proposition 12** *Let $\Phi \colon \mathbb{R}^k \to \mathbb{R} \cup \{+\infty\}$ be an entropy and $\ell \colon \mathcal{A} \to [0, +\infty]^n$. If $\ell$ is $\Phi$-mixable, then the Bayes risk satisfies $\underline{L}_\ell \in C^1(]0, +\infty[^n)$. If, additionally, $\underline{L}_\ell$ is twice differentiable on $]0, +\infty[^n$, then $\Phi$ must be strictly convex on $\Delta_k$.*

*Proof.* Let $\mathfrak{l} = \{1, 2\}$. Since $\ell$ is $\Phi$-mixable, it must be $\Phi_{\mathfrak{l}}$-mixable, where $\Phi_{\mathfrak{l}} \coloneqq \Phi_{\mathfrak{l}} \circ \Pi_{\mathfrak{l}}^{\mathsf{T}} : \mathbb{R}^2 \to \mathbb{R} \cup \{+\infty\}$ (Proposition 10). Let $\Psi \coloneqq \Phi_{\mathfrak{l}}$.

For $w \in ]0, +\infty[$ and $z \in \operatorname{int} \operatorname{dom} \tilde{\Psi}^* = \mathbb{R}$ (see appendix E), we define $(\tilde{\Psi}^*)'_\infty(w) \coloneqq \lim_{t \to +\infty} [\tilde{\Psi}^*(z + tw) - \tilde{\Psi}^*(z)]/t$. The value of $(\tilde{\Psi}^*)'_\infty(w)$ does not depend on the choice of $z$, and it holds that $(\tilde{\Psi}^*)'_\infty(w) = \sigma_{\operatorname{dom} \tilde{\Psi}}(w)$ and $(\tilde{\Psi}^*)'_\infty(-w) = \sigma_{\operatorname{dom} \tilde{\Psi}}(-w)$ [7, Prop. C.1.2.2]. In our case, we have $\operatorname{dom} \tilde{\Psi} = [0, 1]$ (by definition of $\tilde{\Psi}$), which implies that $\sigma_{\operatorname{dom} \tilde{\Psi}}(1) = 1$ and $\sigma_{\operatorname{dom} \tilde{\Psi}}(-1) = 0$. Therefore, $(\tilde{\Psi}^*)'_\infty(1) + (\tilde{\Psi}^*)'_\infty(-1) = 1$. As a result $\tilde{\Psi}^*$ cannot be affine. For all $\delta > 0$, let $g_\delta \colon \mathbb{R} \times \{-1, 0, +1\} \to \mathbb{R}$ be defined by

$$g_\delta(s, u) \coloneqq [\tilde{\Psi}^*(s + \delta(u + 1)/2) - \tilde{\Psi}^*(s + \delta(u - 1)/2)]/\delta.$$

Since $\tilde{\Psi}^*$ is convex it must have non-decreasing slopes (ibid., p.13). Combining this with the fact that $\tilde{\Psi}^*$ is not affine implies that

$$\exists s^*_\delta \in \mathbb{R}, \ g_\delta(s^*_\delta, -1) < g_\delta(s^*_\delta, +1). \tag{33}$$

The fact that $\tilde{\Psi}^*$ has non-decreasing slopes also implies that

$$g_\delta(s^*_\delta, +1) = [\tilde{\Psi}^*(s^*_\delta + \delta) - \tilde{\Psi}^*(s^*_\delta)]/\delta \le \lim_{t \to \infty} [\tilde{\Psi}^*(s^*_\delta + t) - \tilde{\Psi}^*(s^*_\delta)]/t = (\tilde{\Psi}^*)'_\infty(1) = 1.$$

Similarly, we have $0 = -(\tilde{\Psi}^*)'_\infty(-1) \le g_\delta(s^*_\delta, -1)$. Let $\tilde{\mu} \in \partial \tilde{\Psi}^*(s^*_\delta)$. Since $\tilde{\Psi}$ is a closed convex function the following equivalence holds $\tilde{\mu} \in \partial \tilde{\Psi}^*(s^*_\delta) \iff s^*_\delta \in \partial \tilde{\Psi}(\tilde{\mu})$ (ibid., Cor. D.1.4.4). Thus, if $\tilde{\mu} \in \{0, 1\} = \operatorname{bd} \tilde{\Delta}_2$, then $\partial \tilde{\Psi}(\tilde{\mu}) \ne \varnothing$, which is not possible since $\ell$ is $\Psi$-mixable (Lemma 13).

[**We show $\underline{L}_\ell \in C^1(]0, +\infty[^n)$**] We will now show that $\underline{L}_\ell$ is continuously differentiable on $]0, +\infty[^n$. Since $\underline{L}_\ell$ is 1-homogeneous, it suffices to check the differentiability on $\operatorname{ri} \Delta_n$. Suppose $\underline{L}_\ell$ is not differentiable at $\boldsymbol{p} \in \operatorname{ri} \Delta_n$. From Lemma 18, there exists $\boldsymbol{a}_0, \boldsymbol{a}_1 \in \mathcal{A}$ such that $\ell(\boldsymbol{a}_0), \ell(\boldsymbol{a}_1) \in \partial \sigma_{\mathscr{S}_\ell}(-\boldsymbol{p})$ and $\ell(\boldsymbol{a}_0) \ne \ell(\boldsymbol{a}_1)$. Let $A \coloneqq [\boldsymbol{a}_0, \boldsymbol{a}_1] \in \mathbb{R}^{n \times 2}$, $\delta \coloneqq \min\{|\ell_x(\boldsymbol{a}_0) - \ell_x(\boldsymbol{a}_1)| : x \in [n], |\ell_x(\boldsymbol{a}_0) - \ell_x(\boldsymbol{a}_1)| > 0\}$, and $s^*_\delta \in \mathbb{R}$ as in (33). We denote $g^- \coloneqq g_\delta(s^*_\delta, -1)$ and $g^+ \coloneqq g_\delta(s^*_\delta, +1) \in ]0, 1]$. Let $\tilde{\mu} \in \partial \tilde{\Psi}^*(s^*_\delta) \in \operatorname{int} \tilde{\Delta}_2$ and $\boldsymbol{\mu} = \Pi_2(\tilde{\mu}) \in \operatorname{ri} \Delta_2$. From the fact that $\ell$ is $\Psi$-mixable, $J_2^{\mathsf{T}} \ell_x(A) = \ell_x(\boldsymbol{a}_0) - \ell_x(\boldsymbol{a}_1)$, and (8), there must exist $\boldsymbol{a}_* \in \mathcal{A}$ such that for all $x \in [n]$,

$$\ell_x(\boldsymbol{a}_*) \le \operatorname{Mix}_\Psi(\ell_x(A), \boldsymbol{\mu}),$$
$$= \ell_x(a_1) + \tilde{\Psi}^*(s^*_\delta) - \tilde{\Psi}^*(s^*_\delta - \ell_x(\boldsymbol{a}_0) + \ell_x(\boldsymbol{a}_1)),$$

and by letting $\mathrm{sgn}$ be the *sign* function

$$\leq \ell_x(a_1) + g_\delta(s_\delta^*, -\mathrm{sgn}[\ell_x(\boldsymbol{a}_0) - \ell_x(\boldsymbol{a}_1)])[\ell_x(\boldsymbol{a}_0) - \ell_x(\boldsymbol{a}_1)], \tag{34}$$

where in (34) we used the fact that $\tilde{\Psi}^*$ has non-decreasing slopes and the definition of $\delta$. When $\ell_x(\boldsymbol{a}_0) \leq \ell_x(\boldsymbol{a}_1)$, (34) becomes $\ell_x(\boldsymbol{a}_*) \leq (1 - g^+)\ell_x(\boldsymbol{a}_1) + g^+\ell_x(\boldsymbol{a}_0)$. Otherwise, we have $\ell_x(\boldsymbol{a}_*) \leq (1 - g^-)\ell_x(\boldsymbol{a}_1) + g^-\ell_x(\boldsymbol{a}_0) < (1 - g^+)\ell_x(\boldsymbol{a}_1) + g^+\ell_x(\boldsymbol{a}_0)$. Since $\ell$ is admissible, there must exist at least one $x \in [n]$ such that $\ell_x(\boldsymbol{a}_0) > \ell_x(\boldsymbol{a}_1)$. Combining this with the fact that $p_x > 0, \forall x \in [n]$ ($\boldsymbol{p} \in \mathrm{ri}\,\Delta_n$), implies that $\langle \boldsymbol{p}, \ell(\boldsymbol{a}_*)\rangle < \langle \boldsymbol{p}, (1 - g^+)\ell(\boldsymbol{a}_1) + g^+\ell(\boldsymbol{a}_0)\rangle = \underline{L}_\ell(\boldsymbol{p})$. This contradicts the fact that $\ell(\boldsymbol{a}_*) \in \mathscr{S}_\ell$. Therefore, $\underline{L}_\ell$ must be differentiable at $\boldsymbol{p}$. As argued earlier, this implies that $\underline{L}_\ell$ must be differentiable on $]0, +\infty[^n$. Combining this with the fact that $\underline{L}_\ell$ is concave on $]0, +\infty[^n$, implies that $\underline{L}_\ell$ is continuously differentiable on $]0, +\infty[^n$ (ibid., Rmk. D.6.2.6).

[**We show** $\tilde{\Phi}^* \in C^1(\mathbb{R}^{k-1})$] Suppose that $\tilde{\Phi}^*$ is not differentiable at some $\boldsymbol{s}^* \in \mathbb{R}^{k-1}$. Then there exists $\boldsymbol{d} \in \mathbb{R}^{k-1} \setminus \{\boldsymbol{0}_{\tilde{k}}\}$ such that $-(\tilde{\Phi}^*)'(\boldsymbol{s}^*; -\boldsymbol{d}) < (\tilde{\Phi}^*)'(\boldsymbol{s}^*; \boldsymbol{d})$. Since $\boldsymbol{s}^* \in \mathrm{int}\,\mathrm{dom}\,\tilde{\Phi}^*$, $(\tilde{\Phi}^*)'(\boldsymbol{s}^*, \cdot)$ is finite and convex [7, Prop. D.1.1.2], and thus it is continuous on $\mathrm{dom}\,\tilde{\Phi}^* = \mathbb{R}^{k-1}$ (ibid., Rmk. B.3.1.3). Consequently, there exists $\delta^* > 0$ such that

$$\forall \hat{\boldsymbol{d}} \in \mathbb{R}^{k-1}, \|\hat{\boldsymbol{d}} - \boldsymbol{d}\| \leq \delta^* \implies -(\tilde{\Phi}^*)'(\boldsymbol{s}^*; -\hat{\boldsymbol{d}}) < (\tilde{\Phi}^*)'(\boldsymbol{s}^*; \hat{\boldsymbol{d}}) \tag{35}$$

Let $g\colon \{-1, 1\} \to \mathbb{R}$ be such that

$$g(u) := \sup_{\|\hat{\boldsymbol{d}} - \boldsymbol{d}\| \leq \delta^*} u \cdot (\tilde{\Phi}^*)'(\boldsymbol{s}^*; u\hat{\boldsymbol{d}}).$$

Note that since $\tilde{\Phi}^*$ has increasing slopes ($\tilde{\Phi}^*$ is convex), $g(1) \leq \sup_{\|\hat{\boldsymbol{d}} - \boldsymbol{d}\| \leq \delta^*} (\tilde{\Phi}^*)'_\infty(\hat{\boldsymbol{d}}) = \sup_{\|\hat{\boldsymbol{d}} - \boldsymbol{d}\| \leq \delta^*} \sigma_{\mathrm{dom}\,\tilde{\Phi}}(\hat{\boldsymbol{d}}) \leq 1$, where the last inequality holds because $\tilde{\Delta}_k \subset \mathcal{B}(\boldsymbol{0}_{\tilde{k}}, 1)$, and thus $\sigma_{\mathrm{dom}\,\tilde{\Phi}}(\hat{\boldsymbol{d}}) = \sigma_{\tilde{\Delta}_k}(\hat{\boldsymbol{d}}) \leq \sigma_{\mathcal{B}(\boldsymbol{0}_{\tilde{k}}, 1)}(\hat{\boldsymbol{d}}) = 1$. Let $\Delta g := g(1) - g(-1)$. From (35), it is clear that $\Delta g > 0$.

Suppose that $\underline{L}_\ell$ is twice differentiable on $]0, +\infty[^n$ and let $\underline{\ell}$ be a support loss of $\ell$. By definition of a support loss, $\forall \boldsymbol{p} \in \mathrm{ri}\,\Delta_k, \tilde{\underline{\ell}}(\tilde{\boldsymbol{p}}) = \underline{\ell}(\boldsymbol{p}) = \nabla \underline{L}_\ell(\boldsymbol{p})$ (where $\tilde{\underline{\ell}} := \underline{\ell} \circ \mathrm{II}_n$). Thus, since $\underline{L}_\ell$ is twice differentiable on $]0, +\infty[^n$, $\tilde{\underline{\ell}}$ is differentiable on $\mathrm{int}\,\tilde{\Delta}_n$. Furthermore, $\underline{\ell}$ is continuous on $\mathrm{ri}\,\Delta_k$ given that $\underline{L}_\ell \in C^1(]0, +\infty[^n)$ as shown in the first part of this proof. We may assume without loss of generality that $\ell$ is not a constant function. Thus, from Theorem 5, $\underline{\ell}$ is not a constant function either. Consequently, the mean value theorem applied to $\underline{\ell}$ (see e.g. [12, Thm. 5.10]) between any two points in $\mathrm{ri}\,\Delta_n$ with distinct images under $\underline{\ell}$, implies that there exists $(\tilde{\boldsymbol{p}}_*, \boldsymbol{v}_*) \in \mathrm{int}\,\tilde{\Delta}_n \times \mathbb{R}^{n-1}$, such that $\mathrm{D}\tilde{\underline{\ell}}(\tilde{\boldsymbol{p}}_*)\boldsymbol{v}_* \neq \boldsymbol{0}_{\tilde{n}}$. For the rest of the proof let $(\tilde{\boldsymbol{p}}, \boldsymbol{v}) := (\tilde{\boldsymbol{p}}_*, \boldsymbol{v}_*)$ and define $\mathfrak{I} := \{x \in [n] : \mathrm{D}\tilde{\underline{\ell}}_x(\tilde{\boldsymbol{p}})\boldsymbol{v} \neq 0\}$. From Lemma 8, we have $\langle \boldsymbol{p}, \mathrm{D}\tilde{\underline{\ell}}(\tilde{\boldsymbol{p}})\rangle = \boldsymbol{0}_{\tilde{n}}^\mathsf{T}$, which implies that there exists $x \in \mathfrak{I}, \mathrm{D}\tilde{\underline{\ell}}_x(\tilde{\boldsymbol{p}})\boldsymbol{v} > 0$. Thus, the set

$$\mathfrak{K} := \left\{ x \in \mathfrak{I} : \mathrm{D}\tilde{\underline{\ell}}_x(\tilde{\boldsymbol{p}})\boldsymbol{v} > 0 \right\} \tag{36}$$

is non-empty. From this and the fact that $\boldsymbol{p} \in \mathrm{ri}\,\Delta_n$, it follows that

$$\sum_{x' \in \mathfrak{K}} p_{x'} \mathrm{D}\tilde{\underline{\ell}}_{x'}(\tilde{\boldsymbol{p}})\boldsymbol{v} > 0. \tag{37}$$

Let $\tilde{\boldsymbol{p}}^t := \tilde{\boldsymbol{p}} + t\boldsymbol{v}$. From Taylor's Theorem (see e.g. [?, §151]) applied to the function $t \mapsto \tilde{\underline{\ell}}(\tilde{\boldsymbol{p}}^t)$, there exists $\epsilon^* > 0$ and functions $\delta_x : [-\epsilon^*, \epsilon^*] \to \mathbb{R}^n, x \in [n]$, such that $\lim_{t \to 0} t^{-1}\delta_x(t) = 0$ and

$$\forall |t| \leq \epsilon^*, \quad \underline{\ell}_x(\boldsymbol{p}^t) = \underline{\ell}_x(\boldsymbol{p}) + t\mathrm{D}\tilde{\underline{\ell}}_x(\boldsymbol{p})\boldsymbol{v} + \delta_x(t). \tag{38}$$

For $x \in [n]$, let $\boldsymbol{d}_x \in \mathbb{R}^{\tilde{k}}$ and suppose that $\|\boldsymbol{d}_x - \boldsymbol{d}\| \leq \delta^*$ (we will define $\boldsymbol{d}_x$ explicitly later). By shrinking $\epsilon^*$ if necessary, we may assume that

$$\forall x \in \mathfrak{I}, \forall \theta \in [k], \forall |t| \leq \epsilon^*, \quad d_\theta t^{-1} \delta_x(t) \leq \frac{\delta^* |\mathrm{D}\tilde{\underline{\ell}}_x(\tilde{\boldsymbol{p}})\boldsymbol{v}|}{\sqrt{n}}, \tag{39}$$

$$\forall x \notin \mathfrak{I}, \quad \tilde{\Phi}^*(\boldsymbol{s}^*) - \tilde{\Phi}^* \left( \boldsymbol{s}^* - \left[ \delta_x \left( \epsilon^* \frac{d_\theta}{\|\boldsymbol{d}\|} \right) \right]_{\theta \in [\tilde{k}]} \right) \leq \epsilon^* \frac{\Delta g}{4\|\boldsymbol{d}\|} \sum_{x' \in \mathfrak{K}} p_{x'} \mathrm{D}\tilde{\underline{\ell}}_{x'}(\tilde{\boldsymbol{p}})\boldsymbol{v}, \tag{40}$$

$$\forall x \in [n], \quad \tilde{\Phi}^*(\boldsymbol{s}^*) - \tilde{\Phi}^* \left( \boldsymbol{s}^* - \epsilon^* \frac{\mathrm{D}\tilde{\underline{\ell}}_x(\tilde{\boldsymbol{p}})\boldsymbol{v}}{\|\boldsymbol{d}\|} \boldsymbol{d}_x \right) \leq -(\tilde{\Phi}^*)' \left( \boldsymbol{s}^*; -\epsilon^* \frac{\mathrm{D}\tilde{\underline{\ell}}_x(\tilde{\boldsymbol{p}})\boldsymbol{v}}{\|\boldsymbol{d}\|} \boldsymbol{d}_x \right)$$
$$+ \epsilon^* \frac{\Delta g}{4\|\boldsymbol{d}\|} \sum_{x' \in \mathfrak{K}} p_{x'} \mathrm{D}\tilde{\underline{\ell}}_{x'}(\tilde{\boldsymbol{p}})\boldsymbol{v}, \tag{41}$$

where (41) is satisfied for small enough $\epsilon^*$ because of (37) and the fact that

$$\frac{1}{\epsilon} \left( \tilde{\Phi}^*(\boldsymbol{s}^*) - \tilde{\Phi}^*(\boldsymbol{s}^* - \epsilon \frac{\mathrm{D}\tilde{\underline{\ell}}_x(\tilde{\boldsymbol{p}})\boldsymbol{v}}{\|\boldsymbol{d}\|} \boldsymbol{d}_x) \right) \underset{\epsilon \to 0}{\to} -(\tilde{\Phi}^*)' \left( \boldsymbol{s}^*; -\frac{\mathrm{D}\tilde{\underline{\ell}}_x(\tilde{\boldsymbol{p}})\boldsymbol{v}}{\|\boldsymbol{d}\|} \boldsymbol{d}_x \right),$$

and (40) is also satisfied for small enough $\epsilon^*$ because $\tilde{\Phi}^*(\boldsymbol{s}^*) - \tilde{\Phi}^* \left( \boldsymbol{s}^* - \left[ \delta_x \left( \epsilon \frac{d_\theta}{\|\boldsymbol{d}\|} \right) \right]_{\theta \in [\tilde{k}]} \right) = O \left( \max_{\{\theta \in [\tilde{k}]\}} \left| \delta_x \left( \epsilon \frac{d_\theta}{\|\boldsymbol{d}\|} \right) \right| \right) = o(\epsilon)$, where the first equality is due to the fact that $(\lambda, \boldsymbol{z}) \mapsto \frac{1}{\lambda} \left( \tilde{\Phi}^*(\boldsymbol{s}^*) - \tilde{\Phi}^*(\boldsymbol{s}^* - \lambda \boldsymbol{z}) \right)$ is uniformly bounded on compact subsets of $\mathbb{R} \times \mathbb{R}^{\tilde{k}}$ (by continuity of the directional derivative $(\tilde{\Phi}^*)'(\boldsymbol{s}^*; \cdot)$).

If $\mathrm{D}\tilde{\underline{\ell}}_x(\tilde{\boldsymbol{p}})\boldsymbol{v} \leq 0$, then by the positive homogeneity of the directional derivative, the definition of the function $g$, and (41), we get

$$\tilde{\Phi}^*(\boldsymbol{s}^*) - \tilde{\Phi}^* \left( \boldsymbol{s}^* - \epsilon^* \frac{\mathrm{D}\tilde{\underline{\ell}}_x(\tilde{\boldsymbol{p}})\boldsymbol{v}}{\|\boldsymbol{d}\|} \boldsymbol{d}_x \right) \leq \epsilon^* \frac{\mathrm{D}\tilde{\underline{\ell}}_x(\tilde{\boldsymbol{p}})\boldsymbol{v}}{\|\boldsymbol{d}\|} g(1) + \epsilon^* \frac{\Delta g}{4\|\boldsymbol{d}\|} \sum_{x' \in \mathfrak{K}} p_{x'} \mathrm{D}\tilde{\underline{\ell}}_{x'}(\tilde{\boldsymbol{p}})\boldsymbol{v}. \tag{42}$$

On the other hand, if $\mathrm{D}\tilde{\underline{\ell}}_x(\tilde{\boldsymbol{p}})\boldsymbol{v} > 0$, then from the monotonicity of the slopes of $\tilde{\Phi}^*$, the positive homogeneity of the directional derivative, and the definition of the function $g$, it follows that

$$\frac{1}{\epsilon^*} \left( \tilde{\Phi}^*(\boldsymbol{s}^*) - \tilde{\Phi}^*(\boldsymbol{s}^* - \epsilon^* \frac{\mathrm{D}\tilde{\underline{\ell}}_x(\tilde{\boldsymbol{p}})\boldsymbol{v}}{\|\boldsymbol{d}\|} \boldsymbol{d}_x) \right) \leq -(\tilde{\Phi}^*)' \left( \boldsymbol{s}^*; -\frac{\mathrm{D}\tilde{\underline{\ell}}_x(\tilde{\boldsymbol{p}})\boldsymbol{v}}{\|\boldsymbol{d}\|} \boldsymbol{d}_x \right),$$
$$= -\frac{\mathrm{D}\tilde{\underline{\ell}}_x(\tilde{\boldsymbol{p}})\boldsymbol{v}}{\|\boldsymbol{d}\|} (\tilde{\Phi}^*)' \left( \boldsymbol{s}^*; -\boldsymbol{d}_x \right),$$
$$\leq \frac{\mathrm{D}\tilde{\underline{\ell}}_x(\tilde{\boldsymbol{p}})\boldsymbol{v}}{\|\boldsymbol{d}\|} g(-1),$$
$$= \frac{\mathrm{D}\tilde{\underline{\ell}}_x(\tilde{\boldsymbol{p}})\boldsymbol{v}}{\|\boldsymbol{d}\|} \left( -\Delta g + g(1) \right). \tag{43}$$

Let $\lambda_\theta := \epsilon^* \frac{d_\theta}{\|\boldsymbol{d}\|}$, for $\theta \in [\tilde{k}]$. From Theorem 5, there exists $[\boldsymbol{a}_\theta]_{\theta \in [k]} \in \mathcal{A}^k$, such that

$$\ell(\boldsymbol{a}_k) = \underline{\ell}(\boldsymbol{p}) \quad \text{and} \quad \forall \theta \in [\tilde{k}], \ \ell(\boldsymbol{a}_\theta) = \underline{\ell}(\boldsymbol{p}^{\lambda_\theta}) = \underline{\ell}(\boldsymbol{p}) + \epsilon^* \frac{d_\theta}{\|\boldsymbol{d}\|} \mathrm{D}\tilde{\underline{\ell}}(\tilde{\boldsymbol{p}})\boldsymbol{v} + \delta \left( \epsilon^* \frac{d_\theta}{\|\boldsymbol{d}\|} \right), \tag{44}$$

where $[\delta(\cdot)]_x := \delta_x(\cdot)$ for $x \in [n]$.

From the fact that $\ell$ is $\Phi$-mixable, it follows that there exists $\boldsymbol{a}_* \in \mathcal{A}$ such that for all $x \in [n]$,

$$\ell_x(\boldsymbol{a}_*) \leq \mathrm{Mix}_\Phi(\ell_x(\boldsymbol{a}_{1:k}), \boldsymbol{\mu}) = \ell_x(\boldsymbol{a}_k) + \tilde{\Phi}^*(\boldsymbol{s}^*) - \tilde{\Phi}^* \left( \boldsymbol{s}^* - J_k^\top \ell_x(\boldsymbol{a}_{1:k}) \right). \tag{45}$$

For $x \in [n]$, we now define $\boldsymbol{d}_x \in \mathbb{R}^{\tilde{k}}$ explicitly as

$$\forall \theta \in [\tilde{k}], \quad d_{x,\theta} := \begin{cases} d_\theta + \frac{\|\boldsymbol{d}\|}{\epsilon^* [\mathrm{D}\tilde{\underline{\ell}}_x(\tilde{\boldsymbol{p}})\boldsymbol{v}]} \delta_x \left( \epsilon^* \frac{d_\theta}{\|\boldsymbol{d}\|} \right), & \text{if } x \in \mathfrak{I} \\ d_\theta, & \text{otherwise.} \end{cases}$$

From (39), we have $\|\boldsymbol{d}_x - \boldsymbol{d}\| \leq \delta^*, \forall x \in [n]$. Furthermore, from (44) and the fact that for all $x \in [n], J_k^\top \ell_x(\boldsymbol{a}_{1:k}) = [\ell_x(\boldsymbol{a}_\theta) - \ell_x(\boldsymbol{a}_k)]_{\theta \in [\tilde{k}]}$, we have

$$J_k^\top \ell_x(\boldsymbol{a}_{1:k}) = \begin{cases} \epsilon^* \frac{\mathrm{D}\tilde{\underline{\ell}}_x(\tilde{\boldsymbol{p}})\boldsymbol{v}}{\|\boldsymbol{d}\|} \boldsymbol{d}_x, & \text{if } x \in \mathfrak{I}; \\ \left[ \delta_x \left( \epsilon^* \frac{d_\theta}{\|\boldsymbol{d}\|} \right) \right]_{\theta \in [\tilde{k}]}, & \text{otherwise.} \end{cases} \tag{46}$$

Using this, together with (42) and (43), we get $\forall x \in \mathfrak{I}$,

$$\tilde{\Phi}^*(\boldsymbol{s}^*) - \tilde{\Phi}^*\left(\boldsymbol{s}^* - J_k^{\mathsf{T}}\ell_x(\boldsymbol{a}_{1:k})\right) = \tilde{\Phi}^*(\boldsymbol{s}^*) - \tilde{\Phi}^*\left(\boldsymbol{s}^* - \epsilon^*\frac{\mathsf{D}\tilde{\ell}_x(\tilde{\boldsymbol{p}})\boldsymbol{v}}{\|\boldsymbol{d}\|}\boldsymbol{d}_x\right),$$

$$\leq \epsilon^*\frac{\mathsf{D}\tilde{\ell}_x(\tilde{\boldsymbol{p}})\boldsymbol{v}}{\|\boldsymbol{d}\|}g(1) - \epsilon^*\Delta g\frac{\mathsf{D}\tilde{\ell}_x(\tilde{\boldsymbol{p}})\boldsymbol{v}}{\|\boldsymbol{d}\|}\mathbb{1}_{\{\mathsf{D}\tilde{\ell}_x(\tilde{\boldsymbol{p}})\boldsymbol{v}>0\}}$$

$$+ \epsilon^*\frac{\Delta g}{4\|\boldsymbol{d}\|}\mathbb{1}_{\{\mathsf{D}\tilde{\ell}_x(\tilde{\boldsymbol{p}})\boldsymbol{v}\leq 0\}}\sum_{x'\in\mathfrak{K}}p_{x'}\mathsf{D}\tilde{\ell}_{x'}(\tilde{\boldsymbol{p}})\boldsymbol{v}. \quad (47)$$

Combining (45), (46), and (47) yields

$$\langle\boldsymbol{p},\ell(\boldsymbol{a}_*)\rangle \leq \langle\boldsymbol{p},\ell(\boldsymbol{a}_k)\rangle + \frac{\epsilon^*}{\|\boldsymbol{d}\|}\langle\boldsymbol{p},\mathsf{D}\tilde{\ell}(\tilde{\boldsymbol{p}})\boldsymbol{v}\rangle g(1) - \frac{3\epsilon^*\Delta g}{4\|\boldsymbol{d}\|}\sum_{x'\in\mathfrak{K}}p_{x'}\mathsf{D}\tilde{\ell}_{x'}(\tilde{\boldsymbol{p}})\boldsymbol{v}$$

$$+ \sum_{x\notin\mathfrak{I}}p_x\left(\tilde{\Phi}^*(\boldsymbol{s}^*) - \tilde{\Phi}^*\left(\boldsymbol{s}^* - \left[\delta_x\left(\epsilon^*\frac{d_\theta}{\|\boldsymbol{d}\|}\right)\right]_{\theta\in[\tilde{k}]}\right)\right),$$

using (40) and the fact that $\langle\boldsymbol{p},\mathsf{D}\tilde{\ell}(\tilde{\boldsymbol{p}})\rangle = \boldsymbol{0}_{\tilde{n}}^{\mathsf{T}}$ (see Lemma 8), we get

$$\leq \langle\boldsymbol{p},\ell(\boldsymbol{a}_k)\rangle - \frac{\epsilon^*\Delta g}{2\|\boldsymbol{d}\|}\sum_{x'\in\mathfrak{K}}p_{x'}\mathsf{D}\tilde{\ell}_{x'}(\tilde{\boldsymbol{p}})\boldsymbol{v}, \quad (48)$$

$$< \langle\boldsymbol{p},\underline{\ell}(\boldsymbol{p})\rangle, \quad (49)$$

where in (49) we used (37) and the fact that $\underline{\ell}(\boldsymbol{p}) = \ell(\boldsymbol{a}_k)$ (see (45)). Equation 49 shows that $\ell(\boldsymbol{a}^*) \notin \mathscr{S}_\ell$, which is a contradiction. $\qquad\square$

## C.3 Proof of Theorem 7

**Theorem 7** *Let $\eta > 0$, and let $\ell: \mathcal{A} \to [0,+\infty]^n$ a loss. Suppose that $\mathrm{dom}\,\ell = \mathcal{A}$ and that $\underline{L}_\ell$ is twice differentiable on $]0,+\infty[^n$. If $\underline{\eta}_\ell > 0$ then $\ell$ is $\underline{\eta}_\ell$-mixable. In particular, $\eta_\ell \geq \underline{\eta}_\ell$.*

*Proof.* Let $\eta := \underline{\eta}_\ell$. We will show that $\exp(-\eta\mathscr{S}_\ell)$ is convex, which will imply that $\ell$ is $\eta$-mixable [5].

Since $\underline{\eta}_\ell = \inf_{\tilde{\boldsymbol{p}}\in\mathrm{int}\,\tilde{\Delta}_n}(\lambda_{\max}([\mathsf{H}\underline{\tilde{L}}_{\log}(\tilde{\boldsymbol{p}})]^{-1}\mathsf{H}\underline{\tilde{L}}_\ell(\tilde{\boldsymbol{p}})))^{-1} > 0$, $\eta\underline{L}_\ell - \underline{L}_{\log}$ is convex on $\mathrm{ri}\,\Delta_n$ [14, Thm. 10]. Let $\boldsymbol{p} \in \mathrm{ri}\,\Delta_n$ and define

$$\Lambda(\boldsymbol{r}) := \underline{L}_{\log}(\boldsymbol{r}) + \langle\boldsymbol{r},\eta\underline{\ell}(\boldsymbol{p}) - \ell_{\log}(\boldsymbol{p})\rangle, \ \boldsymbol{r} \in \mathrm{ri}\,\Delta_n.$$

Since $\Lambda$ is equal to $\underline{L}_{\log}$ plus an affine function, it follows that $\eta\underline{L}_\ell - \Lambda$ is also convex on $\mathrm{ri}\,\Delta_n$. On the one hand, since $\underline{\ell}$ and $\ell_{\log}$ are proper losses, we have $\langle\boldsymbol{p},\underline{\ell}(\boldsymbol{p})\rangle = \underline{L}_\ell(\boldsymbol{p})$ and $\langle\boldsymbol{p},\ell_{\log}(\boldsymbol{p})\rangle = \underline{L}_{\log}(\boldsymbol{p})$ which implies that

$$\eta\underline{L}_\ell(\boldsymbol{p}) - \Lambda(\boldsymbol{p}) = 0. \quad (50)$$

On the other hand, since $\underline{L}_\ell$ and $\underline{L}_{\log}$ are differentiable we have $\underline{\ell}(\boldsymbol{p}) = \nabla\underline{L}_\ell(\boldsymbol{p})$ and $\nabla\underline{L}_{\log}(\boldsymbol{p}) = \ell_{\log}(\boldsymbol{p})$, which yields $\eta\nabla\underline{L}_\ell(\boldsymbol{p}) - \nabla\Lambda(\boldsymbol{p}) = \boldsymbol{0}_n$. This implies that $\eta\underline{L}_\ell - \Lambda$ attains a minimum at $\boldsymbol{p}$ [7, Thm. D.2.2.1]. Combining this fact with (50) gives $\eta\underline{L}_\ell(\boldsymbol{r}) \geq \Lambda(\boldsymbol{r}), \forall\boldsymbol{r} \in \mathrm{ri}\,\Delta_n$, or equivalently $-\eta\underline{L}_\ell \leq -\Lambda$. By Proposition 1-(iii), this implies

$$[-\eta\underline{L}_\ell]^* \geq [-\Lambda]^*. \quad (51)$$

Using Proposition 1-(ii), we get $[-\Lambda]^*(\boldsymbol{s}) = [-\underline{L}_{\log}]^*(\boldsymbol{s} - \ell_{\log}(\boldsymbol{p}) + \eta\underline{\ell}(\boldsymbol{p}))$ for $\boldsymbol{s} \in \mathbb{R}^n$. Since $-\eta\underline{L}_\ell(\boldsymbol{u}) = -\underline{L}_\ell(\eta\boldsymbol{u}) = \sigma_{\mathscr{S}_\ell}(-\eta\boldsymbol{u})$ and $\sigma^*_{\mathscr{S}_\ell} = \iota_{\mathscr{S}_\ell}$, Proposition 1-(v) implies $[-\eta\underline{L}_\ell]^*(\boldsymbol{s}) = \iota_{\mathscr{S}_\ell}(-\boldsymbol{s}/\eta)$. Similarly, we have $[-\underline{L}_{\log}]^*(\boldsymbol{s}) = \iota_{\mathscr{S}_{\log}}(-\boldsymbol{s})$. Therefore, (51) implies

$$\forall\boldsymbol{s} \in \mathbb{R}^n, \quad \iota_{\mathscr{S}_\ell}(-\boldsymbol{s}/\eta) \geq \iota_{\mathscr{S}_{\log}}(-\boldsymbol{s} + \ell_{\log}(\boldsymbol{p}) - \eta\underline{\ell}(\boldsymbol{p})).$$

This inequality implies that if $\boldsymbol{s} \in -\eta\mathscr{S}_\ell$, then $\boldsymbol{s} \in -\mathscr{S}_{\log} + \ell_{\log}(\boldsymbol{p}) - \eta\underline{\ell}(\boldsymbol{p})$. In particular, if $\boldsymbol{u} \in e^{-\eta\mathscr{S}_\ell}$ then

$$\boldsymbol{u} \in e^{-\mathscr{S}_{\log}+\ell_{\log}(\boldsymbol{p})-\eta\underline{\ell}(\boldsymbol{p})} \subseteq \mathcal{H}_{\tau(\boldsymbol{p}),1} = \{\boldsymbol{v} \in \mathbb{R}^n : \langle\boldsymbol{v},\boldsymbol{p}\odot e^{\eta\underline{\ell}(\boldsymbol{p})}\rangle \leq 1\}. \quad (52)$$

To see the set inclusion in (52), consider $s \in -\mathscr{S}_{\log} + \ell_{\log}(p) - \eta\underline{\ell}(p)$, then by definition of the superprediction set $\mathscr{S}_{\log}$ there exists $r \in \Delta_n$ and $v \in [0, +\infty[^n$, such that $s = \log r - \log p - \eta\underline{\ell}(p) - v$. Thus,

$$\langle e^s, p \odot e^{\eta\underline{\ell}(p)} \rangle = \langle r, e^{-v} \rangle \le 1, \tag{53}$$

where the inequality is true because $r \in \Delta_n$ and $v \in [0, +\infty[^n$. The above argument shows that $e^{-\eta\mathscr{S}_\ell} \subseteq \mathcal{H}_{\tau(p),1}$, where $\tau(p) := p \odot e^{\eta\underline{\ell}(p)}$. Furthermore, $e^{-\eta\mathscr{S}_\ell} \subseteq \mathcal{H}_{\tau(p),1} \cap ]0, +\infty[^n$, since all elements of $e^{-\eta\mathscr{S}_\ell}$ have non-negative, finite components. The latter set inclusion still holds for $\hat{p} \in \text{rbd}\, \Delta_n$. In fact, from the definition of a support loss, there exists a sequence $(p_m)$ in $\text{ri}\, \Delta_n$ converging to $\hat{p}$ such that $\underline{\ell}(p_m) \overset{m \to \infty}{\to} \underline{\ell}(\hat{p})$. Equation 53 implies that for $u \in e^{-\eta\mathscr{S}_\ell}$, $\langle u, p_m \odot e^{\eta\underline{\ell}(p_m)} \rangle \le 1$. Since the inner product is continuous, by passage to the limit, we obtain $\langle u, \hat{p} \odot e^{\eta\underline{\ell}(\hat{p})} \rangle \le 1$. Therefore,

$$e^{-\eta\mathscr{S}_\ell} \subseteq \bigcap_{p \in \Delta_n} \mathcal{H}_{\tau(p),1} \cap ]0, +\infty[^n. \tag{54}$$

Now suppose $u \in \bigcap_{p \in \Delta_n} \mathcal{H}_{\tau(p),1} \cap ]0, +\infty[^n$; that is, for all $p \in \Delta_n$,

$$1 \ge \left\langle u, p \odot e^{\eta\underline{\ell}(p)} \right\rangle = \left\langle p, u \odot e^{\eta\underline{\ell}(p)} \right\rangle = \left\langle p, e^{\eta\underline{\ell}(p) + \log u} \right\rangle,$$
$$\ge e^{\langle p, \eta\underline{\ell}(p) \rangle + \langle p, \log u \rangle}, \tag{55}$$

where the first equality is obtained merely by expanding the expression of the inner product, and the second inequality is simply Jensen's Inequality. Since $u \mapsto e^u$ is strictly convex, the Jensen's inequality in (55) is strict unless $\exists (c, p) \in \mathbb{R} \times \Delta_n$, such that

$$\eta\underline{\ell}(p) + \log u = c\mathbf{1}_n. \tag{56}$$

By substituting (56) into (55), we get $1 \ge \exp(c)$, and thus $c \le 0$. Furthermore, (56) together with the fact that $u \in ]0, +\infty[^n$ imply that $p \in \text{dom}\,\underline{\ell}$, and thus there exists $a \in \text{dom}\,\ell$ such that $\ell(a) = \underline{\ell}(p)$ (Theorem 5). Using this and rearranging (56), we get $u = \exp(-\eta\ell(a) + c\mathbf{1})$. Since $c \le 0$, this means that $u \in \exp(-\eta\mathscr{S}_\ell)$. Suppose now that (56) does not hold. In this case, (55) must be a strict inequality for all $p \in \Delta_n$. By applying the log on both side of (55),

$$\forall p \in \Delta_n, \eta\underline{L}_\ell(p) + \langle p, \log u \rangle = \langle p, \eta\underline{\ell}(p) \rangle + \langle p, \log u \rangle < 0. \tag{57}$$

Since $p \mapsto \underline{L}_\ell(p) = -\sigma_{\mathscr{S}_\ell}(-p)$ is a closed concave function, the map $g: p \mapsto \eta\underline{L}_\ell(p) + \langle p, \log u \rangle$ is also closed and concave, and thus upper semi-continuous. Since $\Delta_n$ is compact, the function $g$ must attain its maximum in $\Delta_n$. Due to (57) this maximum is negative; there exists $c_1 > 0$ such that

$$\forall p \in \Delta_k, \langle p, \eta\underline{\ell}(p) \rangle - \langle p, -\log u \rangle \le -c_1. \tag{58}$$

Let $f(p, x) := \eta\underline{\ell}_x(p) + \log u_x + c_1$, for $x \in [n]$. It follows from (58) that for all $p \in \Delta_n$, $\mathbb{E}_{x \sim p} f(p, x) \le 0$ and $\forall x \in [n], -\infty < f(p, x)$. Thus, Lemma 6 applied to $f$ with $\epsilon = c_1/2$, implies that there exists $p_* \in \text{ri}\, \Delta_n$, such that $\eta\underline{\ell}(p_*) \le -\log u - c_1/2 \le -\log u$. From this inequality, $p_* \in \text{dom}\,\underline{\ell}$, and therefore, there exists $a_* \in \text{dom}\,\ell$ such that $\ell(a_*) = \underline{\ell}(p_*)$ (Theorem 5). This shows that $\eta\ell(a_*) \le -\log u$, which implies that $u \in \exp -\eta\mathscr{S}_\ell$. Therefore, $\bigcap_{p \in \Delta_n} \mathcal{H}_{\tau(p),1} \cap ]0, +\infty[^n \subseteq e^{-\eta\mathscr{S}_\ell}$. Combining this with (54) shows that $e^{-\eta\mathscr{S}_\ell} = \bigcap_{p \in \Delta_n} \mathcal{H}_{\tau(p),1} \cap ]0, +\infty[^n$. Since $e^{-\eta\mathscr{S}_\ell}$ is the intersection of convex set, it is a itself convex set. Since $\text{dom}\,\ell = \mathcal{A}$ by assumption, it follows that $\mathscr{S}_\ell = \mathscr{S}_\ell^\infty$, and thus $e^{-\eta\mathscr{S}_\ell^\infty}$ is convex. This last fact implies that $\ell$ is $\eta$-mixable [5]. $\square$

## C.4 Proof of Theorem 10

We start by the following characterization of $\Delta$-differentiability (this was defined on page 5 of the main body of the paper).

**Lemma 19.** *Let* $\Phi \colon \mathbb{R}^k \to \mathbb{R} \cup \{+\infty\}$ *be an entropy. Then* $\Phi$ *is* $\Delta$*-differentiable if and only if* $\forall \mathfrak{l} \subseteq [k]$ *such that* $|\mathfrak{l}| > 1$, $\tilde{\Phi}_\mathfrak{l} := \Phi \circ \text{II}_k \circ [\Pi_\mathfrak{l}^{\tilde{k}}]^\mathsf{T}$ *is differentiable on* $\text{int}\, \tilde{\Delta}_{|\mathfrak{l}|}$.

*Proof.* This is a direct consequence of Proposition B.4.2.1 in [7], since 1) $\tilde{\Phi}_{\mathfrak{l}}$ is convex; and 2)

$$\tilde{\Phi}'_{\mathfrak{l}}(\tilde{\boldsymbol{u}}; \tilde{\boldsymbol{v}} - \tilde{\boldsymbol{u}}) = \tilde{\Phi}'([\Pi_{\mathfrak{l}}^{\tilde{k}}]^{\mathsf{T}}\tilde{\boldsymbol{u}}; [\Pi_{\mathfrak{l}}^{\tilde{k}}]^{\mathsf{T}}(\boldsymbol{v} - \boldsymbol{u})),$$
$$= \Phi'(\amalg_k[\Pi_{\mathfrak{l}}^{\tilde{k}}]^{\mathsf{T}}\tilde{\boldsymbol{u}}; \amalg_k[\Pi_{\mathfrak{l}}^{\tilde{k}}]^{\mathsf{T}}(\boldsymbol{v} - \boldsymbol{u})),$$

for all $\tilde{\boldsymbol{u}}, \tilde{\boldsymbol{v}} \in \text{int } \tilde{\Delta}_{|\mathfrak{l}|}$ and $\tilde{\Phi} := \Phi \circ \amalg_k$. $\qquad\square$

**Theorem 10** *Let $\Phi : \mathbb{R} \to \mathbb{R} \cup \{+\infty\}$ be a $\Delta$-differentiable entropy. Let $\ell : \mathcal{A} \to [0, +\infty]^n$ be a loss (not necessarily finite) such that $\underline{L}_\ell$ is twice differentiable on $]0, +\infty[^n$. If $\ell$ is $(\eta, \Phi)$-mixable then the GAA achieves a constant regret in the $\mathfrak{G}_\ell^n(\mathcal{A}, k)$ game; for any sequence $(x^t, \boldsymbol{a}_{1:k}^t)_{t=1}^T$,*

$$\text{Loss}_{\text{GAA}}^\ell(T) - \min_{\theta \in [k]} \text{Loss}_\theta^\ell(T) \leq R_\ell^\Phi := \inf_{\boldsymbol{q} \in \Delta_k} \max_{\theta \in [k]} D_\Phi(\boldsymbol{e}_\theta, \boldsymbol{q}) / \eta_\ell^\Phi,$$

*where $\boldsymbol{e}_\theta$ is the $\theta$th basis element of $\mathbb{R}^k$.*

*Proof.* For all $\mathfrak{l} \subseteq [k]$ such that $|\mathfrak{l}| > 1$, let $\tilde{\Phi} := \Phi \circ \amalg_k$ and $\tilde{\Phi}_{\mathfrak{l}} := \tilde{\Phi} \circ [\Pi_{\mathfrak{l}}^{\tilde{k}}]^{\mathsf{T}}$. From Lemma 14 the infimum involved in the definition of the expert distribution $\boldsymbol{q}^t$ in Algorithm 2 is indeed attained. It remains to verify that this minimum is unique. This will become clear in what follows.

Let $\mathfrak{l}^0 = [k]$ and $\mathfrak{I}^t := \{\theta \in [k] : \ell_{x^t}(\boldsymbol{a}_\theta^t) < +\infty\}, t \in [T]$. For $t \in [T]$, we define the non-increasing sequence of subsets $(\mathfrak{l}^t)$ of $[k]$ defined by $\mathfrak{l}^t := \mathfrak{I}^t \cap \mathfrak{l}^{t-1}$. We show by induction that $\boldsymbol{q}^t \in \Delta_{\mathfrak{l}^t}$ and

$$\nabla \tilde{\Phi}_{\mathfrak{l}^t}(\Pi_{\mathfrak{l}^t}^{\tilde{k}} \tilde{\boldsymbol{q}}^t) = \Pi_{\mathfrak{l}^t}^{\tilde{k}} \left( \nabla \tilde{\Phi}(\tilde{\boldsymbol{q}}^0) - \sum_{s=1}^t J_k^{\mathsf{T}} \ell_{x^s}(A^s) \right), \tag{59}$$

where $A^s := [\boldsymbol{a}_\theta^s] \in \mathcal{A}^k, s \in \mathbb{N}$. Suppose that (59) holds true up to some $t \geq 1$. We will now show that it holds for $t + 1$. To simplify expressions, we denote $\tilde{\boldsymbol{x}}_{\mathfrak{l}} := \Pi_{\mathfrak{l}}^{\tilde{k}} \tilde{\boldsymbol{x}} \in \mathbb{R}^{\mathfrak{l}}$ for $\tilde{\boldsymbol{x}} \in \mathbb{R}^{\tilde{k}}$, and $\boldsymbol{z}^t := \ell_{x^t}(A^t), t \in [T]$. From the definition of $\boldsymbol{q}^t$ in Algorithm 2, we have

$$\boldsymbol{q}^{t+1} \in \mathcal{M} := \underset{\boldsymbol{\mu} \in \Delta_k}{\text{Argmin}} \langle \boldsymbol{\mu}, \boldsymbol{z}^{t+1} \rangle + D_\Phi(\boldsymbol{\mu}, \boldsymbol{q}^t).$$

Using the definition of $\mathfrak{I}^{t+1}$,

$$\mathcal{M} = \underset{\boldsymbol{\mu} \in \Delta_{\mathfrak{l}^{t+1}}}{\text{Argmin}} \langle \boldsymbol{\mu}, \boldsymbol{z}^{t+1} \rangle + D_\Phi(\boldsymbol{\mu}, \boldsymbol{q}^t),$$
$$= \underset{\boldsymbol{\mu} \in \Delta_{\mathfrak{l}^{t+1}}}{\text{Argmin}} \langle \boldsymbol{\mu}, \boldsymbol{z}^{t+1} \rangle + \tilde{\Phi}_{\mathfrak{l}^t}(\tilde{\boldsymbol{\mu}}_{\mathfrak{l}^t}) - \tilde{\Phi}_{\mathfrak{l}^t}(\tilde{\boldsymbol{q}}_{\mathfrak{l}}^t) - \tilde{\Phi}'_{\mathfrak{l}^t}(\tilde{\boldsymbol{q}}_{\mathfrak{l}^t}^t; \tilde{\boldsymbol{\mu}}_{\mathfrak{l}^t} - \tilde{\boldsymbol{q}}_{\mathfrak{l}^t}^t).$$

Now using the facts that $\boldsymbol{q}^t \in \Delta_{\mathfrak{l}^t}, \boldsymbol{\mu} \in \Delta_{\mathfrak{l}^{t+1}} \subseteq \Delta_{\mathfrak{l}^t}$, $\Phi$ is $\Delta$-differentiable, and Lemma 19, we have

$$\mathcal{M} = \underset{\boldsymbol{\mu} \in \Delta_{\mathfrak{l}^{t+1}}}{\text{Argmin}} \langle \boldsymbol{\mu}, \boldsymbol{z}^{t+1} \rangle + \tilde{\Phi}_{\mathfrak{l}^{t+1}}(\tilde{\boldsymbol{\mu}}_{\mathfrak{l}^{t+1}}) - \tilde{\Phi}_{\mathfrak{l}^t}(\tilde{\boldsymbol{q}}_{\mathfrak{l}^t}^t) - \langle \tilde{\boldsymbol{\mu}}_{\mathfrak{l}^t} - \tilde{\boldsymbol{q}}_{\mathfrak{l}^t}^t, \nabla \tilde{\Phi}_{\mathfrak{l}^t}(\tilde{\boldsymbol{q}}_{\mathfrak{l}^t}^t) \rangle.$$

Using the facts that $\langle \boldsymbol{\mu}, \boldsymbol{z}^{t+1} \rangle = z_k^{t+1} + \langle \tilde{\boldsymbol{\mu}}_{\mathfrak{l}^{t+1}}, \Pi_{\mathfrak{l}^{t+1}}^{\tilde{k}} J_k^{\mathsf{T}} \boldsymbol{z}^{t+1} \rangle$, for $\tilde{\boldsymbol{\mu}} \in \tilde{\Delta}_{\mathfrak{l}^{t+1}}$, and $\langle \tilde{\boldsymbol{\mu}}_{\mathfrak{l}^t}, \nabla \tilde{\Phi}_{\mathfrak{l}^t}(\tilde{\boldsymbol{q}}_{\mathfrak{l}^t}^t) \rangle \mathcal{M} = \langle \tilde{\boldsymbol{\mu}}_{\mathfrak{l}^{t+1}}, \Pi_{\mathfrak{l}^{t+1}}^{\mathfrak{l}^t} \nabla \tilde{\Phi}_{\mathfrak{l}^t}(\tilde{\boldsymbol{q}}_{\mathfrak{l}^t}^t) \rangle$ (since $\boldsymbol{\mu} \in \Delta_{\mathfrak{l}^{t+1}}$)

$$\mathcal{M} = \underset{\boldsymbol{\mu} \in \Delta_{\mathfrak{l}^{t+1}}}{\text{Argmin}} \langle \tilde{\boldsymbol{\mu}}_{\mathfrak{l}^{t+1}}, -\Pi_{\mathfrak{l}^{t+1}}^{\mathfrak{l}^t} \nabla \tilde{\Phi}_{\mathfrak{l}^t}(\tilde{\boldsymbol{q}}_{\mathfrak{l}^t}^t) + \Pi_{\mathfrak{l}^{t+1}}^{\tilde{k}} J_k^{\mathsf{T}} \boldsymbol{z}^{t+1} \rangle + \tilde{\Phi}_{\mathfrak{l}^{t+1}}(\tilde{\boldsymbol{\mu}}_{\mathfrak{l}^{t+1}})$$
$$+ \langle \tilde{\boldsymbol{q}}_{\mathfrak{l}^t}^t, \nabla \tilde{\Phi}_{\mathfrak{l}^t}(\tilde{\boldsymbol{q}}_{\mathfrak{l}^t}^t) \rangle - \tilde{\Phi}_{\mathfrak{l}^t}(\tilde{\boldsymbol{q}}_{\mathfrak{l}^t}^t),$$

and since the last two terms are independent of $\boldsymbol{\mu}$,

$$\mathcal{M} = \underset{\boldsymbol{\mu} \in \Delta_{\mathfrak{l}^{t+1}}}{\text{Argmin}} \langle \tilde{\boldsymbol{\mu}}_{\mathfrak{l}^{t+1}}, -\Pi_{\mathfrak{l}^{t+1}}^{\mathfrak{l}^t} \nabla \tilde{\Phi}_{\mathfrak{l}^t}(\tilde{\boldsymbol{q}}_{\mathfrak{l}^t}^t) + \Pi_{\mathfrak{l}^{t+1}}^{\tilde{k}} J_k^{\mathsf{T}} \boldsymbol{z}^{t+1} \rangle + \tilde{\Phi}_{\mathfrak{l}^{t+1}}(\tilde{\boldsymbol{\mu}}_{\mathfrak{l}^{t+1}}).$$

Now using Fenchel duality property in Proposition 1-(iv),

$$\mathcal{M} = \{ \boldsymbol{\mu} \in \Delta_{\mathfrak{l}^{t+1}} : \Pi_{\mathfrak{l}^{t+1}}^{\tilde{k}} \circ \Pi_k(\boldsymbol{\mu}) = \tilde{\boldsymbol{\mu}}_{\mathfrak{l}^{t+1}} \in \partial \tilde{\Phi}_{\mathfrak{l}^{t+1}}^*(\Pi_{\mathfrak{l}^{t+1}}^{\mathfrak{l}^t} \nabla \tilde{\Phi}_{\mathfrak{l}^t}(\tilde{\boldsymbol{q}}_{\mathfrak{l}^t}^t) - \Pi_{\mathfrak{l}^{t+1}}^{\tilde{k}} J_k^{\mathsf{T}} \boldsymbol{z}^{t+1}) \}.$$

Finally, due to Lemma 10 and Proposition 12, $\tilde{\Phi}_{\mathfrak{l}^{t+1}}^*$ is differentiable on $\mathbb{R}^{|\mathfrak{l}^{t+1}|-1}$, and thus

$$\mathcal{M} = \{ \amalg_k \circ [\Pi_{\mathfrak{l}^{t+1}}^{\tilde{k}}]^{\mathsf{T}} \circ \nabla \tilde{\Phi}_{\mathfrak{l}^{t+1}}^*(\Pi_{\mathfrak{l}^{t+1}}^{\mathfrak{l}^t} \nabla \tilde{\Phi}_{\mathfrak{l}^t}(\tilde{\boldsymbol{q}}_{\mathfrak{l}^t}^t) - \Pi_{\mathfrak{l}^{t+1}}^{\tilde{k}} J_k^{\mathsf{T}} \boldsymbol{z}^{t+1}) \}. \tag{60}$$

From (60), we obtain

$$\nabla \tilde{\Phi}_{\mathfrak{l}^{t+1}}(\Pi_{\mathfrak{l}^{t+1}}^{\tilde{k}} \tilde{\boldsymbol{q}}^{t+1}) = \Pi_{\mathfrak{l}^{t+1}}^{\mathfrak{l}^t} \nabla \tilde{\Phi}_{\mathfrak{l}^t}(\tilde{\boldsymbol{q}}_{\mathfrak{l}^t}^t) - \Pi_{\mathfrak{l}^{t+1}}^{\tilde{k}} J_k^{\mathsf{T}} \boldsymbol{z}^{t+1}. \tag{61}$$

Thus using the induction assumption and the fact that $\Pi_{\mathfrak{l}^{t+1}}^{\mathfrak{l}^t} \Pi_{\mathfrak{l}^t}^{\tilde{k}} = \Pi_{\mathfrak{l}^{t+1}}^{\tilde{k}}$ (since $\mathfrak{l}^{t+1} \subseteq \mathfrak{l}^t$), the result follows, i.e. (59) is true for all $t \in [T]$. Furthermore, $\boldsymbol{q}^{t+1} \in \Delta_{\mathfrak{l}^{t+1}}$, since $\Pi_{\mathfrak{l}^{t+1}}^{\tilde{k}} \tilde{\boldsymbol{q}}^{t+1} \in \operatorname{dom} \tilde{\Phi}_{\mathfrak{l}^{t+1}} \subseteq \tilde{\Delta}_{|\mathfrak{l}^{t+1}|}$. Using the same arguments as above, one arrives at

$$\operatorname{Mix}_\Phi(\boldsymbol{q}^t, \boldsymbol{z}^{t+1}) = z_k^{t+1} + \inf_{\tilde{\boldsymbol{\mu}} \in \Delta_{\mathfrak{l}^{t+1}}} \langle \tilde{\boldsymbol{\mu}}_{\mathfrak{l}^{t+1}}, -\Pi_{\mathfrak{l}^{t+1}}^{\mathfrak{l}^t} \nabla \tilde{\Phi}_{\mathfrak{l}^t}(\tilde{\boldsymbol{q}}_{\mathfrak{l}^t}^t) + \Pi_{\mathfrak{l}^{t+1}}^{\tilde{k}} J_k^{\mathsf{T}} \boldsymbol{z}^{t+1} \rangle + \tilde{\Phi}_{\mathfrak{l}^{t+1}}(\tilde{\boldsymbol{\mu}}_{\mathfrak{l}^{t+1}})$$
$$+ \langle \tilde{\boldsymbol{q}}_{\mathfrak{l}^t}^t, \nabla \tilde{\Phi}_{\mathfrak{l}^t}(\tilde{\boldsymbol{q}}_{\mathfrak{l}^t}^t) \rangle - \tilde{\Phi}_{\mathfrak{l}^t}(\tilde{\boldsymbol{q}}_{\mathfrak{l}^t}^t).$$

Using the Fenchel duality property Proposition 1-(vi) and (60),

$$= z_k^{t+1} + \tilde{\Phi}_{\mathfrak{l}^t}^*(\nabla \tilde{\Phi}_{\mathfrak{l}^t}(\tilde{\boldsymbol{q}}_{\mathfrak{l}^t}^t)) - \tilde{\Phi}_{\mathfrak{l}^{t+1}}^*(\Pi_{\mathfrak{l}^{t+1}}^{\mathfrak{l}^t} \nabla \tilde{\Phi}_{\mathfrak{l}^t}(\tilde{\boldsymbol{q}}_{\mathfrak{l}^t}^t) - \Pi_{\mathfrak{l}^{t+1}}^{\tilde{k}} J_k^{\mathsf{T}} \boldsymbol{z}^{t+1}). \tag{62}$$

On the other hand, $\Phi$-mixability implies that there exists $\boldsymbol{a}_*^t \in \mathcal{A}^t$, such that for all $x^t \in [n]$,

$$\forall t \in [T], \ell_{x^t}(\boldsymbol{a}_*^t) \leq \operatorname{Mix}_\Phi(\boldsymbol{q}^{t-1}, \boldsymbol{z}^t),$$

Summing this inequality for $t = 1, \ldots, T$ yields,

$$\sum_{t=1}^{T} \ell_{x^t}(\boldsymbol{a}_*^t) \leq \sum_{t=1}^{T} \operatorname{Mix}_\Phi(\boldsymbol{q}^{t-1}, \boldsymbol{z}^t),$$

and thus using (62) and (61) yields

$$\sum_{t=1}^{T} \ell_{x^t}(\boldsymbol{a}_*^t) \leq \sum_{t=1}^{T} \ell_{x^t}(\boldsymbol{a}_k^t) + \tilde{\Phi}^*(\nabla \tilde{\Phi}(\tilde{\boldsymbol{q}}^0)) - \tilde{\Phi}_{\mathfrak{l}^T}^*(\Pi_{\mathfrak{l}^T}^{\mathfrak{l}^{T-1}} \nabla \tilde{\Phi}_{\mathfrak{l}^{T-1}}(\tilde{\boldsymbol{q}}_{\mathfrak{l}^{T-1}}^{T-1}) - \Pi_{\mathfrak{l}^T}^{\tilde{k}} J_k^{\mathsf{T}} \boldsymbol{z}^T).$$

Finally, using (59) together with the fact that $\Pi_{\mathfrak{l}^T}^{\mathfrak{l}^{T-1}} \Pi_{\mathfrak{l}^{T-1}}^{\tilde{k}} = \Pi_{\mathfrak{l}^T}^{\tilde{k}}$

$$\sum_{t=1}^{T} \ell_{x^t}(\boldsymbol{a}_*^t) \leq \sum_{t=1}^{T} \ell_{x^t}(\boldsymbol{a}_k^t) + \tilde{\Phi}^*(\nabla \tilde{\Phi}(\tilde{\boldsymbol{q}}^0)) - \tilde{\Phi}_{\mathfrak{l}^T}^*\left(\Pi_{\mathfrak{l}^T}^{\tilde{k}} \left(\nabla \tilde{\Phi}(\tilde{\boldsymbol{q}}^0) - \sum_{t=1}^{T} J_k^{\mathsf{T}} \ell_{x^t}(A^t)\right)\right).$$

Using the definition of the Fenchel dual and Proposition 1-(vi) again, the above inequality becomes

$$\sum_{t=1}^{T} \ell_{x^t}(\boldsymbol{a}_*^t) \leq \sum_{t=1}^{T} \ell_{x^t}(\boldsymbol{a}_k^t) + \langle \tilde{\boldsymbol{q}}^0, \nabla \tilde{\Phi}(\tilde{\boldsymbol{q}}^0)) \rangle - \tilde{\Phi}(\tilde{\boldsymbol{q}}^0)$$

$$- \sup_{\boldsymbol{\pi} \in \Delta_{|\mathfrak{l}^T|}} \left[\left\langle \tilde{\boldsymbol{\pi}}, \Pi_{\mathfrak{l}^T}^{\tilde{k}} \left(\nabla \tilde{\Phi}(\tilde{\boldsymbol{q}}^0) - \sum_{t=1}^{T} J_k^{\mathsf{T}} \ell_{x^t}(A^t)\right)\right\rangle - \tilde{\Phi}_{\mathfrak{l}^T}(\tilde{\boldsymbol{\pi}})\right],$$

$$= \sum_{t=1}^{T} \ell_{x^t}(\boldsymbol{a}_k^t) + \langle \tilde{\boldsymbol{q}}^0, \nabla \tilde{\Phi}(\tilde{\boldsymbol{q}}^0)) \rangle - \tilde{\Phi}(\tilde{\boldsymbol{q}}^0)$$

$$+ \inf_{\boldsymbol{\mu} \in \Delta_{\mathfrak{l}^T}} \left[\left\langle \tilde{\boldsymbol{\mu}}, \sum_{t=1}^{T} J_k^{\mathsf{T}} \ell_{x^t}(A^t) - \nabla \tilde{\Phi}(\tilde{\boldsymbol{q}}^0)\right\rangle + \tilde{\Phi}(\tilde{\boldsymbol{\mu}})\right]. \tag{63}$$

Using the fact that $\forall \theta \in [k] \setminus \mathfrak{l}^T, \sum_{t=1}^{T} \ell_{x^t}(\boldsymbol{a}_\theta^t) = +\infty$ (by definition of $(\mathfrak{l}^t)$), the right hand side of (63) becomes

$$\sum_{t=1}^{T} \ell_{x^t}(\boldsymbol{a}_k^t) + \langle \tilde{\boldsymbol{q}}^0, \nabla \tilde{\Phi}(\tilde{\boldsymbol{q}}^0)) \rangle - \tilde{\Phi}(\tilde{\boldsymbol{q}}^0) + \inf_{\boldsymbol{\mu} \in \Delta_k} \left[\left\langle \tilde{\boldsymbol{\mu}}, \sum_{t=1}^{T} J_k^{\mathsf{T}} \ell_{x^t}(A^t) - \nabla \tilde{\Phi}(\tilde{\boldsymbol{q}}^0)\right\rangle + \tilde{\Phi}(\tilde{\boldsymbol{\mu}})\right].$$

Thus, we get

$$\forall \boldsymbol{\mu} \in \Delta_k, \sum_{t=1}^{T} \ell_{x^t}(\boldsymbol{a}_*^t) \leq \sum_{t=1}^{T} \ell_{x^t}(\boldsymbol{a}_k^t) + \left\langle \tilde{\boldsymbol{\mu}}, \sum_{t=1}^{T} J_k^{\mathsf{T}} \ell_{x^t}(A^t)\right\rangle$$

$$+ \tilde{\Phi}(\tilde{\boldsymbol{\mu}}) - \tilde{\Phi}(\tilde{\boldsymbol{q}}^0) - \langle \tilde{\boldsymbol{\mu}} - \tilde{\boldsymbol{q}}^0, \nabla \tilde{\Phi}(\tilde{\boldsymbol{q}}^0) \rangle.$$

Using the facts that $\sum_{t=1}^{T} \ell_{x^t}(\boldsymbol{a}_k^t) + \left\langle \tilde{\boldsymbol{\mu}}, \sum_{t=1}^{T} J_k^\mathsf{T} \ell_{x^t}(A^t) \right\rangle = \left\langle \boldsymbol{\mu}, \sum_{t=1}^{T} \ell_{x^t}(A^t) \right\rangle$ and the definition of the divergence,

$$\forall \boldsymbol{\mu} \in \Delta_k, \ \sum_{t=1}^{T} \ell_{x^t}(\boldsymbol{a}_*^t) \leq \left\langle \boldsymbol{\mu}, \sum_{t=1}^{T} \ell_{x^t}(A^t) \right\rangle + D_\Phi(\boldsymbol{\mu}, \boldsymbol{q}^0),$$

which for $\boldsymbol{\mu} = \boldsymbol{e}_\theta$ implies

$$\forall \theta \in [k], \ \sum_{t=1}^{T} \ell_{x^t}(\boldsymbol{a}_*^t) \leq \sum_{t=1}^{T} \ell_{x^t}(\boldsymbol{a}_\theta^t) + D_\Phi(\boldsymbol{e}_\theta, \boldsymbol{q}^0). \tag{64}$$

When instead of $\Phi$-mixability, we have $(\eta, \Phi)$-mixability, the last term in (64) becomes $\frac{D_\Phi(\boldsymbol{e}_\theta, \boldsymbol{q}^0)}{\eta}$ and the desired result follows.

$\square$

## C.5 Proof of Theorem 11

We require the following result:

**Proposition 20.** *For the Shannon entropy* S*, it holds that* $\tilde{\mathrm{S}}^*(\boldsymbol{v}) = \log(\langle \exp(\boldsymbol{v}), \mathbf{1}_{\tilde{k}} \rangle + 1), \forall \boldsymbol{v} \in \mathbb{R}^{k-1}$, *and* $\mathrm{S}^\star(\boldsymbol{z}) = \log\langle \exp(\boldsymbol{z}), \mathbf{1}_k \rangle, \forall \boldsymbol{z} \in \mathbb{R}^k$.

*Proof.* Given $\boldsymbol{v} \in \mathbb{R}^{k-1}$, we first derive the expression of the Fenchel dual $\tilde{\mathrm{S}}^*(\boldsymbol{v}) := \sup_{\tilde{\boldsymbol{q}} \in \tilde{\Delta}_k} \langle \tilde{\boldsymbol{q}}, \boldsymbol{v} \rangle - \tilde{\mathrm{S}}(\tilde{\boldsymbol{q}})$. Setting the gradient of $\tilde{\boldsymbol{q}} \mapsto \langle \tilde{\boldsymbol{q}}, \boldsymbol{v} \rangle - \tilde{\mathrm{S}}(\tilde{\boldsymbol{q}})$ to $\mathbf{0}_{\tilde{k}}$ gives $\boldsymbol{v} = \nabla \tilde{\mathrm{S}}(\tilde{\boldsymbol{q}})$. For $\boldsymbol{q} \in ]0, +\infty[^k$, we have $\nabla \mathrm{S}(\boldsymbol{q}) = \log \boldsymbol{q} + \mathbf{1}_k$, and from appendix A we know that $\nabla \tilde{\mathrm{S}}(\tilde{\boldsymbol{q}}) = J_k^\mathsf{T} \nabla \mathrm{S}(\boldsymbol{q})$. Therefore,

$$\boldsymbol{v} = \nabla \tilde{\mathrm{S}}(\tilde{\boldsymbol{q}}) \implies \boldsymbol{v} = J_k^\mathsf{T} \nabla \mathrm{S}(\boldsymbol{q}) \implies \boldsymbol{v} = \log \frac{\tilde{\boldsymbol{q}}}{q_k},$$

where the right most equality is equivalent to $\tilde{\boldsymbol{q}}/q_k = \exp(\boldsymbol{v})$. Since $\langle \tilde{\boldsymbol{q}}, \mathbf{1}_{\tilde{k}} \rangle = 1 - q_k$, we get $q_k = (\langle \exp(\boldsymbol{v}), \mathbf{1}_{\tilde{k}} \rangle + 1)^{-1}$. Therefore, the supremum in the definition of $\tilde{\mathrm{S}}^*(\boldsymbol{v})$ is attained at $\tilde{\boldsymbol{q}}_* = \exp(\boldsymbol{v})(\langle \exp(\boldsymbol{v}), \mathbf{1}_{\tilde{k}} \rangle + 1)^{-1}$. Hence $\tilde{\mathrm{S}}^*(\boldsymbol{v}) = \langle \tilde{\boldsymbol{q}}_*, \boldsymbol{v} \rangle - \langle \tilde{\boldsymbol{q}}_*, \log \tilde{\boldsymbol{q}}_* \rangle = \log(\langle \exp(\boldsymbol{v}), \mathbf{1}_{\tilde{k}} \rangle + 1)$. Finally, using (2) we get $\mathrm{S}^\star(\boldsymbol{z}) = \log\langle \exp(\boldsymbol{z}), \mathbf{1}_k \rangle$, for $\boldsymbol{z} \in \mathbb{R}^k$. $\square$

**Theorem 11** *Let* $\eta > 0$. *A loss* $\ell \colon \mathcal{A} \to [0, +\infty]^n$ *is* $\eta$-*mixable if and only if* $\ell$ *is* $(\eta, \mathrm{S})$-*mixable.*

*Proof.*

**Claim 1.** *For all* $\boldsymbol{q} \in \Delta_k$, $A := \boldsymbol{a}_{1:k} \in \mathbb{R}^k$, *and* $x \in [n]$

$$-\eta^{-1} \log \langle \exp(-\eta \ell_x(A)), \boldsymbol{q} \rangle = \mathrm{Mix}_{\mathrm{S}}^\eta(\ell_x(A), \boldsymbol{q}). \tag{65}$$

*Let* $\boldsymbol{q} \in \mathrm{ri}\,\Delta_k$. From Proposition 20, the Shannon entropy is such that $\mathrm{S}^\star$ is differentiable on $\mathbb{R}^k$, and thus it follows from Lemma 14 ((21)-(22)) that for any $\boldsymbol{d} \in [0, +\infty[^k$

$$\mathrm{Mix}_{\mathrm{S}}(\boldsymbol{d}, \boldsymbol{q}) = \mathrm{S}^\star(\nabla \mathrm{S}(\boldsymbol{q})) - \mathrm{S}^\star(\nabla \mathrm{S}(\boldsymbol{q}) - \boldsymbol{d}). \tag{66}$$

By definition of S, $\nabla \mathrm{S}(\boldsymbol{q}) = \log \boldsymbol{q} + \mathbf{1}_k$, and due to Proposition 20, $\mathrm{S}^\star(\boldsymbol{z}) = \log\langle \exp \boldsymbol{z}, \mathbf{1}_k \rangle, \boldsymbol{z} \in \mathbb{R}^k$. Therefore,

$$\nabla \mathrm{S}(\boldsymbol{q}) - \eta \boldsymbol{d} = \log(\exp(-\eta \boldsymbol{d}) \odot \boldsymbol{q}) + \mathbf{1}_k. \tag{67}$$

On the other hand, from [9] we also have

$$\mathrm{Mix}_{\mathrm{S}}^\eta(\boldsymbol{d}, \boldsymbol{q}) = \eta^{-1} \mathrm{Mix}_{\mathrm{S}}(\eta \boldsymbol{d}, \boldsymbol{q}), \ \eta > 0. \tag{68}$$

Combining (66)-(68), yields

$$-\eta^{-1} \log \langle \exp(-\eta \boldsymbol{d}, \boldsymbol{q} \rangle = \mathrm{Mix}_{\mathrm{S}}^\eta(\boldsymbol{d}, \boldsymbol{q}). \tag{69}$$

*Suppose now that $q \in \operatorname{ri} \Delta_{\mathfrak{l}}$ for $\mathfrak{l} \subseteq [k]$ such that $|\mathfrak{l}| > 1$.* By repeating the argument above for $S_{\mathfrak{l}} := S \circ \Pi_{\mathfrak{l}}^{\mathsf{T}}$, we get

$$\forall d \in [0, +\infty[^n, \ \operatorname{Mix}_{S_{\mathfrak{l}}}^{\eta}(\Pi_{\mathfrak{l}} d, \Pi_{\mathfrak{l}} q) = -\eta^{-1} \log\langle \exp(-\eta \Pi_{\mathfrak{l}} d), \Pi_{\mathfrak{l}} q\rangle,$$
$$= -\eta^{-1} \log\langle \exp(-\eta d), q\rangle. \tag{70}$$

Fix $x \in [n]$ and let $\hat{d} := \ell_x(A) \in [0, +\infty]^k$. Let $(\hat{d}_m) \subset [0, +\infty[^k$ be any sequence converging to $\hat{d}$. Lemma 15, $\operatorname{Mix}_S^{\eta}(\hat{d}_m, q) \overset{m \to \infty}{\to} \operatorname{Mix}_S^{\eta}(\hat{d}, q)$. Using this with (70) gives

$$-\eta^{-1} \log \langle \exp(-\eta \ell_x(A)), q\rangle = \lim_{m \to \infty} -\eta^{-1} \log\langle \exp(-\eta \hat{d}_m), q\rangle,$$
$$= \lim_{m \to \infty} \operatorname{Mix}_S^{\eta}(\hat{d}_m, q),$$
$$= \operatorname{Mix}_S^{\eta}(\hat{d}, q) = \operatorname{Mix}_S^{\eta}(\ell_x(A), q). \tag{71}$$

*It remains to check the case where $q$ is a vertex*; Without loss of generality assume that $q = e_1$ and let $\mu \in \Delta_k \setminus \{e_1\}$. Then there exists $\mathfrak{l}_* \subset [k]$, such that $(e_1, \mu) \in (\operatorname{rbd} \Delta_{\mathfrak{l}_*}) \times (\operatorname{ri} \Delta_{\mathfrak{l}_*})$ and by Lemma 11, $S'(e_1; \mu - e_1) = -\infty$. Therefore, $\forall q \in \Delta_k \setminus \{e_1\}, D_{S_\eta}(q, e_1) = +\infty$, which implies

$$\forall x \in [n], \operatorname{Mix}_S^{\eta}(\ell_x(A), e_1) = \inf_{q \in \Delta_k} \langle q, \ell_x(A)\rangle + D_{S_\eta}(q, e_1),$$
$$= \langle e_1, \ell_x(A)\rangle + D_{S_\eta}(e_1, e_1),$$
$$= \langle e_1, \ell_x(A)\rangle,$$
$$= \ell_x(a_1) = -\eta^{-1} \log \langle \exp(-\eta \ell_x(A)), e_1\rangle. \tag{72}$$

Combining (72) and (71) proves the claim in (65). The desired equivalence follows trivially from the definitions of $\eta$-mixability and $(\eta, S)$-mixability.

$\square$

## C.6 Proof of Theorem 13

We need the following lemma to show Theorem 13.

**Lemma 21.** *Let $\Phi$ be as in Theorem 13. Then $\eta_\ell \Phi - S$ is convex on $\Delta_k$ only if $\Phi$ satisfies (10).*

*Proof.* Let $\hat{q} \in \operatorname{rbd} \Delta_k$. Suppose that there exists $q \in \operatorname{ri} \Delta_k$ such that $\Phi'(\hat{q}; q - \hat{q}) > -\infty$. Since $\Phi$ is convex, it must have non-decreasing slopes; in particular, it holds that $\Phi'(\hat{q}; q - \hat{q}) \le \Phi(q) - \Phi(\hat{q})$. Therefore, since $\Phi$ is finite on $\Delta_k$ (by definition of an entropy), we have $\Phi'(\hat{q}; q - \hat{q}) < +\infty$. Since by assumption $\eta_\ell \Phi - S$ is convex and finite on the simplex, we can use the same argument to show that $[\eta_\ell \Phi - S]'(\hat{q}; q - \hat{q}) = \eta_\ell \Phi'(\hat{q}; q - \hat{q}) - S'(\hat{q}; q - \hat{q}) < +\infty$. This is a contradiction since $S'(\hat{q}; q - \hat{q}) = -\infty$ (Lemma 11). Therefore, it must hold that $\Phi'(\hat{q}; q - \hat{q}) = -\infty$.

Suppose now that $(\hat{q}, q) \in (\operatorname{rbd} \Delta_{\mathfrak{l}}) \times (\operatorname{ri} \Delta_{\mathfrak{l}})$ for $\mathfrak{l} \subseteq [k]$, with $|\mathfrak{l}| > 1$. Let $\Phi_{\mathfrak{l}} := \Phi \circ \Pi_{\mathfrak{l}}^{\mathsf{T}}$ and $S_{\mathfrak{l}} := S \circ \Pi_{\mathfrak{l}}^{\mathsf{T}}$. Since $\eta_\ell \Phi - S$ is convex on $\Delta_k$ and $\Pi_{\mathfrak{l}}$ is a linear function, $\eta_\ell \Phi_{\mathfrak{l}} - S_{\mathfrak{l}}$ is convex on $\Delta_{|\mathfrak{l}|}$. Repeating the steps above for $\Phi$ and $S$ substituted by $\Phi_{\mathfrak{l}}$ and $S_{\mathfrak{l}}$, respectively, we get that $(\Phi_{\mathfrak{l}})'(\Pi_{\mathfrak{l}} \hat{q}; \Pi_{\mathfrak{l}} q - \Pi_{\mathfrak{l}} \hat{q}) = -\infty$. Since $(\Phi_{\mathfrak{l}})'(\Pi_{\mathfrak{l}} \hat{q}; \Pi_{\mathfrak{l}} q - \Pi_{\mathfrak{l}} \hat{q}) = \Phi'(\hat{q}; q - \hat{q})$ the proof is completed. $\square$

**Theorem 13** *Let $\eta > 0$, $\ell \colon \mathcal{A} \to [0, +\infty]^n$ a $\eta$-mixable loss, and $\Phi \colon \mathbb{R}^k \to \mathbb{R} \cup \{+\infty\}$ an entropy. If $\eta \Phi - S$ is convex on $\Delta_k$, then $\ell$ is $\Phi$-mixable.*

*Proof.* Assume $\eta_\ell \Phi - S$ is convex on $\Delta_k$. For this to hold, it is necessary that $\eta_\ell > 0$ since $-S$ is strictly concave. Let $\eta := \eta_\ell$ and $S_\eta := \eta^{-1} S$. Then $\tilde{S}_\eta = \eta^{-1}\tilde{S}$ and $\tilde{\Phi} - \tilde{S}_\eta = (\Phi - S_\eta) \circ \amalg_k$ is convex on $\tilde{\Delta}_k$, since $\Phi - S_\eta$ is convex on $\Delta_k$ and $\amalg_k$ is affine.

Let $x \in [n]$, $A := [a_\theta]_{\theta \in [k]}$, and $q \in \Delta_k$. Suppose that $q \in \operatorname{ri} \Delta_k$ and let $s_q^* \in \partial \tilde{\Phi}(\tilde{q})$ be as in Proposition 14. Note that if $\ell_x(a_\theta) = +\infty, \forall \theta \in [k]$, then the $\Phi$-mixability condition (8) is trivially satisfied. Suppose, without loss of generality, that $\ell_x(a_k) < +\infty$. Let $(d_m) \subset$

$[0, +\infty[^k$ be any sequence such that $\boldsymbol{d}_m \overset{m\to\infty}{\to} \boldsymbol{d} := \ell_x(A) \in [0, +\infty]^k$. From Lemmas 11 and 15, $\mathrm{Mix}_\Psi(\boldsymbol{d}_m, \boldsymbol{q}) \overset{m\to\infty}{\to} \mathrm{Mix}_\Psi(\boldsymbol{d}, \boldsymbol{q})$ for $\Psi \in \{\Phi, \mathrm{S}_\eta\}$.

Let $\tilde{\Upsilon}_{\boldsymbol{q}} : \mathbb{R}^{k-1} \to \mathbb{R} \cup \{+\infty\}$ be defined by

$$\tilde{\Upsilon}_{\boldsymbol{q}}(\tilde{\boldsymbol{\mu}}) := \tilde{\mathrm{S}}_\eta(\tilde{\boldsymbol{\mu}}) + \langle \tilde{\boldsymbol{\mu}}, \boldsymbol{s}_{\boldsymbol{q}}^* - \nabla \tilde{\mathrm{S}}_\eta(\tilde{\boldsymbol{q}}) \rangle - \tilde{\Phi}^*(\boldsymbol{s}_{\boldsymbol{q}}^*) + \tilde{\mathrm{S}}_\eta^*(\nabla \tilde{\mathrm{S}}_\eta(\tilde{\boldsymbol{q}})),$$

and it's Fenchel dual follows from Proposition 1 (i+ii):

$$\tilde{\Upsilon}_{\boldsymbol{q}}^*(\boldsymbol{v}) = \tilde{\mathrm{S}}_\eta^*(\boldsymbol{v} - \boldsymbol{s}_{\boldsymbol{q}}^* + \nabla \tilde{\mathrm{S}}_\eta(\tilde{\boldsymbol{q}})) + \tilde{\Phi}^*(\boldsymbol{s}_{\boldsymbol{q}}^*) - \tilde{\mathrm{S}}_\eta^*(\nabla \tilde{\mathrm{S}}_\eta(\tilde{\boldsymbol{q}})),$$

After substituting $\boldsymbol{v}$ by $\boldsymbol{s}_{\boldsymbol{q}}^* - J_k^\mathsf{T} \boldsymbol{d}$ in the expression of $\tilde{\Upsilon}_{\boldsymbol{q}}^*$ and rearranging, we get

$$\tilde{\mathrm{S}}_\eta^*(\nabla \tilde{\mathrm{S}}_\eta(\tilde{\boldsymbol{q}})) - \tilde{\mathrm{S}}_\eta^*(\nabla \tilde{\mathrm{S}}_\eta(\tilde{\boldsymbol{q}}) - J_k^\mathsf{T} \boldsymbol{d}_m) = \tilde{\Phi}^*(\boldsymbol{s}_{\boldsymbol{q}}^*) - \tilde{\Upsilon}_{\boldsymbol{q}}^*(\boldsymbol{s}_{\boldsymbol{q}}^* - J_k^\mathsf{T} \boldsymbol{d}_m). \tag{73}$$

Since $\boldsymbol{s}_{\boldsymbol{q}}^* \in \partial \tilde{\Phi}(\tilde{\boldsymbol{q}})$ and $\tilde{\Phi}$ is a closed convex function, combining Proposition 1-(iv) and the fact that $\tilde{\Phi}^{**} = \tilde{\Phi}$ [7, Cor. E.1.3.6] yields $\langle \tilde{\boldsymbol{q}}, \boldsymbol{s}_{\boldsymbol{q}}^* \rangle - \tilde{\Phi}^*(\boldsymbol{s}_{\boldsymbol{q}}^*) = \tilde{\Phi}(\tilde{\boldsymbol{q}})$. Thus, after substituting $\tilde{\boldsymbol{\mu}}$ by $\tilde{\boldsymbol{q}}$ in the expression of $\tilde{\Upsilon}_{\boldsymbol{q}}$, we get

$$\tilde{\Phi}(\tilde{\boldsymbol{q}}) = \tilde{\Upsilon}_{\boldsymbol{q}}(\tilde{\boldsymbol{q}}). \tag{74}$$

On the other hand, $\tilde{\Phi} - \tilde{\Upsilon}_{\boldsymbol{q}}$ is convex on $\tilde{\Delta}_k$, since $\tilde{\Upsilon}_{\boldsymbol{q}}$ is equal to $\tilde{\mathrm{S}}_\eta$ plus an affine function. Thus, $\partial[\tilde{\Phi} - \tilde{\Upsilon}_{\boldsymbol{q}}](\tilde{\boldsymbol{q}}) + \partial \tilde{\Upsilon}_{\boldsymbol{q}}(\tilde{\boldsymbol{q}}) = \partial \tilde{\Phi}(\tilde{\boldsymbol{q}})$, since $\tilde{\Phi}$ and $\tilde{\Upsilon}_{\boldsymbol{q}}$ are both convex (ibid., Thm. D.4.1.1). Since $\tilde{\Upsilon}_{\boldsymbol{q}}$ is differentiable at $\tilde{\boldsymbol{q}}$, we have $\partial \tilde{\Upsilon}_{\boldsymbol{q}}(\tilde{\boldsymbol{q}}) = \{\nabla \tilde{\Upsilon}_{\boldsymbol{q}}(\tilde{\boldsymbol{q}})\} = \{\boldsymbol{s}_{\boldsymbol{q}}^*\}$. Furthermore, since $\boldsymbol{s}_{\boldsymbol{q}}^* \in \partial \tilde{\Phi}(\tilde{\boldsymbol{q}})$, then $\boldsymbol{0}_{\tilde{k}} \in \partial \tilde{\Phi}(\boldsymbol{q}) - \partial \tilde{\Upsilon}_{\boldsymbol{q}}(\tilde{\boldsymbol{q}}) = \partial[\tilde{\Phi} - \tilde{\Upsilon}_{\boldsymbol{q}}](\tilde{\boldsymbol{q}})$. Hence, $\tilde{\Phi} - \tilde{\Upsilon}_{\boldsymbol{q}}$ attains a minimum at $\tilde{\boldsymbol{q}}$ (ibid., Thm. D.2.2.1). Due to this and (74), $\tilde{\Phi} \geq \tilde{\Upsilon}_{\boldsymbol{q}}$, which implies that $\tilde{\Phi}^* \leq \tilde{\Upsilon}_{\boldsymbol{q}}^*$ (Proposition 1-(iii)). Using this in (73) gives for all $m \in \mathbb{N}$

$$\begin{aligned} \tilde{\mathrm{S}}_\eta^*(\nabla \tilde{\mathrm{S}}_\eta(\tilde{\boldsymbol{q}})) - \tilde{\mathrm{S}}_\eta^*(\nabla \tilde{\mathrm{S}}_\eta(\tilde{\boldsymbol{q}}) - J_k^\mathsf{T} \boldsymbol{d}_m) &\leq \tilde{\Phi}^*(\boldsymbol{s}_{\boldsymbol{q}}^*) - \tilde{\Phi}^*(\boldsymbol{s}_{\boldsymbol{q}}^* - J_k^\mathsf{T} \boldsymbol{d}_m), \\ \implies \mathrm{Mix}_{\mathrm{S}}^\eta(\boldsymbol{d}_m, \boldsymbol{q}) &\leq \mathrm{Mix}_\Phi(\boldsymbol{d}_m, \boldsymbol{q}), \end{aligned}$$

where the implication is obtained by adding $[\boldsymbol{d}_m]_k$ on both sides of the first inequality and using Proposition 14.

Suppose now that $\boldsymbol{q} \in \mathrm{ri}\,\Delta_{\mathfrak{l}}$, with $|\mathfrak{l}| > 1$, and let $\Phi_{\mathfrak{l}} := \Phi \circ \Pi_{\mathfrak{l}}^\mathsf{T}$ and $\mathrm{S}_{\mathfrak{l}} := \mathrm{S} \circ \Pi_{\mathfrak{l}}^\mathsf{T}$. Note that since $\eta_\ell \Phi - \mathrm{S}$ is convex on $\Delta_k$ and $\Pi_{\mathfrak{l}}$ is a linear function, $\eta_\ell \Phi_{\mathfrak{l}} - \mathrm{S}_{\mathfrak{l}}$ is convex on $\Delta_{|\mathfrak{l}|}$. Repeating the steps above for $\Phi$, $\mathrm{S}$, $\boldsymbol{q}$, and $A$ substituted by $\Phi_{\mathfrak{l}}$, $\mathrm{S}_{\mathfrak{l}}$, $\Pi_{\mathfrak{l}}\boldsymbol{q}$, and $A\Pi_{\mathfrak{l}}^\mathsf{T}$, respectively, yields

$$\begin{aligned} \mathrm{Mix}_{\mathrm{S}_{\mathfrak{l}}}^\eta(\Pi_{\mathfrak{l}}\boldsymbol{d}_m, \Pi_{\mathfrak{l}}\boldsymbol{q}) &\leq \mathrm{Mix}_{\Phi_{\mathfrak{l}}}(\Pi_{\mathfrak{l}}\boldsymbol{d}_m, \Pi_{\mathfrak{l}}\boldsymbol{q}), \\ \implies \mathrm{Mix}_{\mathrm{S}}^\eta(\boldsymbol{d}_m, \boldsymbol{q}) &\leq \mathrm{Mix}_\Phi(\boldsymbol{d}_m, \boldsymbol{q}), \\ \implies \mathrm{Mix}_{\mathrm{S}}^\eta(\ell_x(A), \boldsymbol{q}) &\leq \mathrm{Mix}_\Phi(\ell_x(A), \boldsymbol{q}), \end{aligned} \tag{75}$$

where the first implication follows from Lemma 13, since $\mathrm{S}_\eta$ and $\Phi$ both satisfy (10) (see Lemmas 11 and 21), and (75) is obtained by passage to the limit $m \to \infty$. Since $\eta = \eta_\ell > 0$, $\ell$ is $\eta$-mixable, which implies that $\ell$ is $\mathrm{S}_\eta$-mixable (Theorem 11). Therefore, there exists $\boldsymbol{a}_* \in \mathcal{A}$, such that

$$\ell_x(\boldsymbol{a}_*) \leq \mathrm{Mix}_{\mathrm{S}}^\eta(\ell_x(A), \boldsymbol{q}) \leq \mathrm{Mix}_\Phi(\ell_x(A), \boldsymbol{q}). \tag{76}$$

To complete the proof (that is, to show that $\ell$ is $\Phi$-mixable), it remains to consider the case where $\boldsymbol{q}$ is a vertex of $\Delta_k$. Without loss of generality assume that $\boldsymbol{q} = \boldsymbol{e}_1$ and let $\boldsymbol{\mu} \in \Delta_k \setminus \{\boldsymbol{e}_1\}$. Thus, there exists $\mathfrak{l}_* \subseteq [k]$, with $|\mathfrak{l}_*| > 1$, such that $(\boldsymbol{e}_1, \boldsymbol{\mu}) \in (\mathrm{rbd}\,\Delta_{\mathfrak{l}_*}) \times (\mathrm{ri}\,\Delta_{\mathfrak{l}_*})$, and Lemma 21 implies that $\Phi'(\boldsymbol{e}_1; \boldsymbol{\mu} - \boldsymbol{e}_1) = -\infty$. Therefore, $\forall \boldsymbol{q} \in \Delta_k \setminus \{\boldsymbol{e}_1\}, D_\Phi(\boldsymbol{q}, \boldsymbol{e}_1) = +\infty$, which implies

$$\begin{aligned} \forall x \in [n], \mathrm{Mix}_\Phi(\ell_x(A), \boldsymbol{e}_1) &= \inf_{\boldsymbol{q} \in \Delta_k} \langle \boldsymbol{q}, \ell_x(A) \rangle + D_\Phi(\boldsymbol{q}, \boldsymbol{e}_1), \\ &= \langle \boldsymbol{e}_1, \ell_x(A) \rangle + D_\Phi(\boldsymbol{e}_1, \boldsymbol{e}_1) = \langle \boldsymbol{e}_1, \ell_x(A) \rangle, \\ &= \ell_x(\boldsymbol{a}_1). \end{aligned} \tag{77}$$

The $\Phi$-mixability condition (8) is trivially satisfied in this case. Combining (76) and (77) shows that $\ell$ is $\Phi$-mixable. $\qquad\square$

## C.7 Proof of Theorem 14

The following Lemma gives necessary regularity conditions on the entropy $\Phi$ under the assumptions of Theorem 14.

**Lemma 22.** *Let $\Phi$ and $\ell$ be as in Theorem 14. Then the following holds*

    *(i) $\tilde{\Phi}$ is strictly concave on $\operatorname{int} \tilde{\Delta}_k$.*

    *(ii) $\tilde{\Phi}^*$ is be continuously differentiable on $\mathbb{R}^{k-1}$.*

    *(iii) $\tilde{\Phi}^*$ is twice differentiable on $\mathbb{R}^{k-1}$ and $\forall \tilde{q} \in \operatorname{int} \tilde{\Delta}_k$, $\mathsf{H}\tilde{\Phi}^*(\nabla\tilde{\Phi}(\tilde{q})) = (\mathsf{H}\tilde{\Phi}(\tilde{q}))^{-1}$.*

    *(iv) For the Shannon entropy, we have $(\mathsf{H}\tilde{\mathsf{S}}(\tilde{q}))^{-1} = \mathsf{H}\tilde{\mathsf{S}}^*(\nabla\tilde{\mathsf{S}}(\tilde{q})) = \operatorname{diag}\tilde{q} - \tilde{q}\tilde{q}^{\mathsf{T}}$.*

*Proof.* Since $\ell$ is $\Phi$-mixable and $\underline{L}_\ell$ is twice differentiable on $]0, +\infty[^n$, $\tilde{\Phi}^*$ is continously differentiable on $\mathbb{R}^{n-1}$ (Proposition 12). Therefore, $\tilde{\Phi}$ is strictly convex on $\operatorname{ri} \Delta_k$ [7, Thm. E.4.1.2].

The differentiability of $\tilde{\Phi}$ and $\tilde{\Phi}^*$ implies $\nabla\tilde{\Phi}^*(\nabla\tilde{\Phi}(\tilde{q})) = \tilde{q}$ (ibid.). Since $\tilde{\Phi}$ is twice differentiable on $\operatorname{int} \tilde{\Delta}_k$ (by assumption), the latter equation implies that $\tilde{\Phi}^*$ is twice differentiable on $\nabla\tilde{\Phi}(\operatorname{int} \tilde{\Delta}_k)$. Using the chain rule, we get $\mathsf{H}\tilde{\Phi}^*(\nabla\tilde{\Phi}(u))\mathsf{H}\tilde{\Phi}(u) = I_{\tilde{k}}$. Multiplying both sides of the equation by $(\mathsf{H}\tilde{\Phi}(u))^{-1}$ from the right gives the expression in (iii). Note that $\mathsf{H}\tilde{\Phi}(\cdot)$ is in fact invertible on $\operatorname{int} \tilde{\Delta}_k$ since $\tilde{\Phi}$ is strictly convex on $\operatorname{int} \tilde{\Delta}_k$. It remains to show that $\nabla\tilde{\Phi}(\operatorname{int} \tilde{\Delta}_k) = \mathbb{R}^{k-1}$. This set equality follows from 1) $[\tilde{q} \in \partial\tilde{\Phi}^*(s) \iff s \in \partial\tilde{\Phi}(\tilde{q})]$ (ibid., Cor. E.1.4.4); 2) $\operatorname{dom} \tilde{\Phi}^* = \mathbb{R}^{k-1}$; and 3) $\forall \tilde{q} \in \operatorname{bd} \tilde{\Delta}_k, \partial\tilde{\Phi}(\tilde{q}) = \varnothing$ (Lemma 13).

For the Shannon entropy, we have $\tilde{\mathsf{S}}^*(v) = \log(\langle\exp(v), \mathbf{1}_{\tilde{k}}\rangle + 1)$ (Proposition 20) and $\nabla\tilde{\mathsf{S}}(\tilde{q}) = \log\frac{\tilde{q}}{q_k}$, for $(v, \tilde{q}) \in \mathbb{R}^{k-1} \times \tilde{\Delta}_k$. Thus $(\mathsf{H}\tilde{\mathsf{S}}(\tilde{q}))^{-1} = \mathsf{H}\tilde{\mathsf{S}}^*(\nabla\tilde{\mathsf{S}}(\tilde{q})) = \operatorname{diag}\tilde{q} - \tilde{q}\tilde{q}^{\mathsf{T}}$. $\qquad\square$

To show Theorem 14, we analyze a particular parameterized curve defined in the next lemma.

**Lemma 23.** *Let $\ell\colon \Delta_n \to [0, +\infty]^n$ be a proper loss whose Bayes risk $\underline{L}_\ell$ is twice differentiable on $]0, +\infty[^n$, and let $\Phi$ be an entropy such that $\tilde{\Phi}$ and $\tilde{\Phi}^*$ are twice differentiable on $\operatorname{int} \tilde{\Delta}_k$ and $\mathbb{R}^{k-1}$, respectively. For $(\tilde{p}, \tilde{q}, V) \in \operatorname{int} \tilde{\Delta}_n \times \operatorname{int} \tilde{\Delta}_k \times \mathbb{R}^{\tilde{n} \times \tilde{k}}$, let $\beta\colon \mathbb{R} \to \mathbb{R}^n$ be the curve defined by*

$$\forall x \in [n], \quad \beta_x(t) = \tilde{\ell}_x(\tilde{p}) + \tilde{\Phi}^*(\nabla\tilde{\Phi}(\tilde{q})) - \tilde{\Phi}^*(\nabla\tilde{\Phi}(\tilde{q}) - J_k^{\mathsf{T}}\tilde{\ell}_x(\tilde{P}^t)), \tag{78}$$

*where $\tilde{P}^t = [\tilde{p}\mathbf{1}_{\tilde{k}}^{\mathsf{T}} + tV, \tilde{p}] \in \mathbb{R}^{\tilde{n} \times k}$ and $t \in \{s \in \mathbb{R} : \forall j \in [\tilde{k}], \tilde{p} + sV_{\cdot,j} \in \operatorname{int} \tilde{\Delta}_n\}$. Then*

$$\beta(0) = \tilde{\ell}(\tilde{p}),$$
$$\dot{\beta}(0) = \mathsf{D}\tilde{\ell}(\tilde{p})V\tilde{q},$$
$$\frac{d}{dt}\left\langle p, \dot{\beta}(t) \right\rangle\bigg|_{t=0} = -\sum_{j=1}^{k-1} q_j V_{\cdot,j}^{\mathsf{T}}\mathsf{H}\underline{L}_\ell(\tilde{p})V_{\cdot,j} - \operatorname{tr}(\operatorname{diag}(p)\,\mathsf{D}\tilde{\ell}(\tilde{p})V(\mathsf{H}\tilde{\Phi}(\tilde{q}))^{-1}(\mathsf{D}\tilde{\ell}(\tilde{p})V)^{\mathsf{T}}).$$
$$\tag{79}$$

*Proof.* Since $\tilde{P}^t = [\tilde{p}\mathbf{1}_{\tilde{k}}^{\mathsf{T}} + tV, \tilde{p}] \in \mathbb{R}^{\tilde{n} \times k}$, $\tilde{P}^0 = \tilde{p}\mathbf{1}_k^{\mathsf{T}}$ and $\tilde{\ell}_x(\tilde{P}^0) = \tilde{\ell}_x(\tilde{p})\mathbf{1}_k$. As a result, $J_k^{\mathsf{T}}\tilde{\ell}_x(\tilde{P}^0) = \mathbf{0}_{\tilde{k}}$, and thus $\beta_x(0) = \tilde{\ell}_x(\tilde{p}) + \tilde{\Phi}^*(\nabla\tilde{\Phi}(\tilde{q})) - \tilde{\Phi}^*(\nabla\tilde{\Phi}(\tilde{q}) - \mathbf{0}_{\tilde{k}}) = \tilde{\ell}_x(\tilde{p})$. This shows that $\beta(0) = \tilde{\ell}(\tilde{p})$. Let $\gamma_x(t) := \nabla\tilde{\Phi}(\tilde{q}) - J_k^{\mathsf{T}}\tilde{\ell}_x(\tilde{P}^t)$. For $j \in [k-1]$,

$$\frac{d}{dt}[\gamma_x(t)]_j = \frac{d}{dt}\left([\nabla\tilde{\Phi}(\tilde{q})]_j - [J_k^{\mathsf{T}}\tilde{\ell}_x(\tilde{P}^t)]_j\right),$$
$$= -\frac{d}{dt}\left(\tilde{\ell}_x(\tilde{P}_{\cdot,j}^t) - \tilde{\ell}_x(\tilde{P}_{\cdot,k}^t)\right),$$
$$= -\frac{d}{dt}\left(\tilde{\ell}_x(\tilde{p} + tV_{\cdot,j}) - \tilde{\ell}_x(\tilde{p})\right), \quad \left(\text{since } \frac{d}{dt}\ell_x(\tilde{P}_{\cdot,k}^t) = \frac{d}{dt}\tilde{\ell}_x(\tilde{p}) = 0\right)$$
$$= -\mathsf{D}\tilde{\ell}_x(\tilde{P}_{\cdot,j}^t)V_{\cdot,j}.$$

From the definition of $\tilde{P}^t$, $\tilde{P}^0_{\cdot,j} = \tilde{\boldsymbol{p}}$, $\forall j \in [\tilde{k}]$, and therefore, $\dot{\gamma}_x(0) = -(\mathsf{D}\tilde{\ell}_x(\tilde{\boldsymbol{p}})V)^{\mathsf{T}}$. By differentiating $\beta_x$ in (78) and using the chain rule, $\dot{\beta}_x(t) = -(\dot{\gamma}_x(t))^{\mathsf{T}}\nabla\tilde{\Phi}^*(\gamma_x(t))$. By setting $t = 0$, $\dot{\beta}_x(0) = -(\dot{\gamma}_x(0))^{\mathsf{T}}\nabla\tilde{\Phi}^*(\nabla\tilde{\Phi}(\tilde{\boldsymbol{q}})) = \mathsf{D}\tilde{\ell}_x(\tilde{\boldsymbol{p}})V\tilde{\boldsymbol{q}}$. Thus, $\dot{\beta}(0) = \mathsf{D}\tilde{\ell}(\tilde{\boldsymbol{p}})V\tilde{\boldsymbol{q}}$. Furthermore,

$$
\begin{aligned}
\frac{d}{dt}\left\langle \boldsymbol{p}, \dot{\beta}(t) \right\rangle\Big|_{t=0} &= \frac{d}{dt}\sum_{x=1}^{n} p_x \left( \sum_{j=1}^{k-1} \mathsf{D}\tilde{\ell}_x(\tilde{P}^t_{\cdot,j})V_{\cdot,j}[\nabla\tilde{\Phi}^*(\gamma_x(t))]_j \right)\Bigg|_{t=0}, \\
&= \sum_{j=1}^{k-1} \frac{d}{dt}\left( \sum_{x=1}^{n} p_x \mathsf{D}\tilde{\ell}_x(\tilde{P}^t_{\cdot,j})V_{\cdot,j}[\nabla\tilde{\Phi}^*(\gamma_x(t))]_j \right)\Bigg|_{t=0}, \\
&= \sum_{j=1}^{k-1}\left( \frac{d}{dt}\left\langle \boldsymbol{p}, \mathsf{D}\tilde{\ell}(\tilde{P}^t_{\cdot,j})V_{\cdot,j}q_j \right\rangle\Big|_{t=0} + \sum_{x=1}^{n} p_x \mathsf{D}\tilde{\ell}_x(\tilde{\boldsymbol{p}})V_{\cdot,j}\frac{d}{dt}[\nabla\tilde{\Phi}^*(\gamma_x(t))]_j\Big|_{t=0} \right), \\
&= -\sum_{j=1}^{k-1} q_j V_{\cdot,j}^{\mathsf{T}}\mathsf{H}\underline{\tilde{L}}_\ell(\tilde{\boldsymbol{p}})V_{\cdot,j} - \sum_{x=1}^{n}\sum_{\substack{i=1 \\ j=1}}^{k-1} p_x \mathsf{D}\tilde{\ell}_x(\tilde{\boldsymbol{p}})V_{\cdot,j}[\mathsf{H}\tilde{\Phi}^*(\nabla\tilde{\Phi}(\tilde{\boldsymbol{q}}))]_{j,i}\mathsf{D}\tilde{\ell}_x(\tilde{\boldsymbol{p}})V_{\cdot,i}, \\
&= -\sum_{j=1}^{k-1} q_j V_{\cdot,j}^{\mathsf{T}}\mathsf{H}\underline{\tilde{L}}_\ell(\tilde{\boldsymbol{p}})V_{\cdot,j} - \operatorname{tr}(\operatorname{diag}(\boldsymbol{p})\,\mathsf{D}\tilde{\ell}(\tilde{\boldsymbol{p}})V\mathsf{H}\tilde{\Phi}^*(\nabla\tilde{\Phi}(\tilde{\boldsymbol{q}}))(\mathsf{D}\tilde{\ell}(\tilde{\boldsymbol{p}})V)^{\mathsf{T}}), \\
&= -\sum_{j=1}^{k-1} q_j V_{\cdot,j}^{\mathsf{T}}\mathsf{H}\underline{\tilde{L}}_\ell(\tilde{\boldsymbol{p}})V_{\cdot,j} - \operatorname{tr}(\operatorname{diag}(\boldsymbol{p})\,\mathsf{D}\tilde{\ell}(\tilde{\boldsymbol{p}})V(\mathsf{H}\tilde{\Phi}(\tilde{\boldsymbol{q}}))^{-1}(\mathsf{D}\tilde{\ell}(\tilde{\boldsymbol{p}})V)^{\mathsf{T}}),
\end{aligned}
$$

where in the third equality we used Lemma 6, in the fourth equality we used Lemma 9, and in the sixth equality we used Lemma 22-(iii).

$\square$

In next lemma, we state a necessary condition for $\Phi$-mixability in terms of the parameterized curve $\beta$ defined in Lemma 23.

**Lemma 24.** *Let $\ell$, $\Phi$, and $\beta$ be as in Lemma 23. If $\exists(\tilde{\boldsymbol{p}}, \tilde{\boldsymbol{q}}, V) \in \operatorname{int}\tilde{\Delta}_n \times \operatorname{int}\tilde{\Delta}_k \times \mathbb{R}^{\tilde{n}\times\tilde{k}}$ such that the curve $\gamma(t) := \tilde{\ell}(\tilde{\boldsymbol{p}} + tV\tilde{\boldsymbol{q}})$ satisfies $\frac{d}{dt}\langle\boldsymbol{p}, \dot{\beta}(t) - \dot{\gamma}(t)\rangle\big|_{t=0} < 0$, then $\ell$ is not $\Phi-$mixable. In particular, $\exists P \in \operatorname{ri}\Delta_n^k$, such that $[\operatorname{Mix}_\Phi(\ell_x(P), \boldsymbol{q})]_{x\in[n]}^{\mathsf{T}}$ lies outside $\mathscr{S}_\ell$.*

*Proof.* First note that for any triplet $(\tilde{\boldsymbol{p}}, \tilde{\boldsymbol{q}}, V) \in \operatorname{int}\tilde{\Delta}_n \times \operatorname{int}\tilde{\Delta}_k \times \mathbb{R}^{\tilde{n}\times\tilde{k}}$, the map $t \mapsto \langle\boldsymbol{p}, \dot{\beta}(t) - \dot{\gamma}(t)\rangle$ is differentiable at 0. This follows from Lemmas 6 and 23. Let $r(t) := \mathrm{II}_n(\tilde{\boldsymbol{p}} + tV\tilde{\boldsymbol{q}})$ and $\delta(t) := \langle r(t), \beta(t) - \gamma(t)\rangle$. Then

$$
\dot{\delta}(t) = \left\langle r(t), \dot{\beta}(t) - \dot{\gamma}(t) \right\rangle + \langle V\tilde{\boldsymbol{q}}, \beta(t) - \gamma(t)\rangle.
$$

Since $t \mapsto \langle\boldsymbol{p}, \dot{\beta}(t) - \dot{\gamma}(t)\rangle$ is differentiable at 0, it follows from Lemma 6 that $t \mapsto \dot{\delta}(t)$ is also differentiable at 0, and thus

$$
\begin{aligned}
\ddot{\delta}(0) &= \frac{d}{dt}\left\langle r(t), \dot{\beta}(t) - \dot{\gamma}(t) \right\rangle\Big|_{t=0} + \left\langle J_n V\tilde{\boldsymbol{q}}, \dot{\beta}(0) - \dot{\gamma}(0) \right\rangle, \\
&= \left\langle \dot{r}(0), \dot{\beta}(0) - \dot{\gamma}(0) \right\rangle + \frac{d}{dt}\left\langle \boldsymbol{p}, \dot{\beta}(t) - \dot{\gamma}(t) \right\rangle\Big|_{t=0}, \qquad\qquad (80) \\
&= \left\langle J_n V\tilde{\boldsymbol{q}}, \dot{\beta}(0) - \dot{\gamma}(0) \right\rangle + \frac{d}{dt}\left\langle \boldsymbol{p}, \dot{\beta}(t) - \dot{\gamma}(t) \right\rangle\Big|_{t=0}, \\
&= \frac{d}{dt}\left\langle \boldsymbol{p}, \dot{\beta}(t) - \dot{\gamma}(t) \right\rangle\Big|_{t=0} < 0, \qquad\qquad\qquad\qquad\qquad (81)
\end{aligned}
$$

where (80) and (81) hold because $\dot{\beta}(0) = \mathsf{D}\tilde{\ell}(\tilde{\boldsymbol{p}})V\tilde{\boldsymbol{q}} = \dot{\gamma}(0)$ (see Lemma 23). According to Taylor's theorem (see e.g. [? , §151]), there exists $\epsilon > 0$ and $h : [-\epsilon, \epsilon] \to \mathbb{R}$ such that

$$\forall |t| \le \epsilon, \ \delta(t) = \delta(0) + t\dot{\delta}(0) + \frac{t^2}{2}\ddot{\delta}(0) + h(t)t^2, \tag{82}$$

and $\lim_{t \to 0} h(t) = 0$. From Lemma 23, $\beta(0) = \gamma(0) = 0$ and $\dot{\beta}(0) = \dot{\gamma}(0)$. Therefore, $\delta(0) = \dot{\delta}(0) = 0$ and (82) becomes $\delta(t) = \frac{t^2}{2}\ddot{\delta}(0) + h(t)t^2$. Due to (81) and the fact that $\lim_{t \to 0} h(t) = 0$, we can choose $\epsilon_* > 0$ small enough such that $\delta(\epsilon_*) = \frac{\epsilon_*^2}{2}\ddot{\delta}(0) + h(\epsilon_*)\epsilon_*^2 < 0$. This means that $\langle \mathrm{II}_n(\tilde{\boldsymbol{p}} + \epsilon_* V\tilde{\boldsymbol{q}}), \beta(\epsilon_*) \rangle < \langle \mathrm{II}_n(\tilde{\boldsymbol{p}} + \epsilon_* V\tilde{\boldsymbol{q}}), \tilde{\ell}(\tilde{\boldsymbol{p}} + \epsilon_* V\tilde{\boldsymbol{q}}) \rangle = \langle \mathrm{II}_n(\tilde{\boldsymbol{p}} + \epsilon_* V\tilde{\boldsymbol{q}}), \ell(\mathrm{II}_n(\tilde{\boldsymbol{p}} + \epsilon_* V\tilde{\boldsymbol{q}})) \rangle$. Therefore, $\beta(\epsilon_*)$ must lie outside the superprediction set. Thus, the mixability condition (8) does not hold for $P^{\epsilon_*} = \mathrm{II}_n[\tilde{\boldsymbol{p}}\mathbf{1}_{\tilde{k}}^{\mathsf{T}} + \epsilon_* V, \ \tilde{\boldsymbol{p}}] \in \mathrm{ri}\,\Delta_n^k$. This completes the proof. $\square$

**Theorem 14** *Let* $\ell \colon \mathcal{A} \to [0, +\infty]^n$ *be a loss such that* $\underline{L}_\ell$ *is twice differentiable on* $]0, +\infty[^n$*, and* $\Phi \colon \mathbb{R}^k \to \mathbb{R} \cup \{+\infty\}$ *an entropy such that* $\tilde{\Phi} := \Phi \circ \mathrm{II}_k$ *is twice differentiable on* $\mathrm{int}\,\tilde{\Delta}_k$*. Then* $\ell$ *is* $\Phi$*-mixable only if* $\underline{\eta_\ell}\Phi - \mathrm{S}$ *is convex on* $\Delta_k$*.*

*Proof.* We will prove the contrapositive; suppose that $\underline{\eta_\ell}\Phi - \mathrm{S}$ is not convex on $\Delta_k$ and we show that $\ell$ cannot be $\Phi$-mixable. Note first that from Lemma 22-(iii), $\tilde{\Phi}^*$ is twice differentiable on $\mathbb{R}^{k-1}$. Thus Lemmas 23 and 24 apply. Let $\underline{\ell}$ be a proper support loss of $\ell$ and suppose that $\underline{\eta_\ell}\Phi - \mathrm{S}$ is not convex on $\Delta_k$, This implies that $\underline{\eta_\ell}\tilde{\Phi} - \tilde{\mathrm{S}}$ is not convex on $\mathrm{int}\,\tilde{\Delta}_k$, and by Lemma 3 there exists $\tilde{\boldsymbol{q}}_* \in \mathrm{int}\,\tilde{\Delta}_k$, such that $1 > \underline{\eta_\ell}\lambda_{\min}(\mathsf{H}\tilde{\Phi}(\tilde{\boldsymbol{q}}_*)(\mathsf{H}\tilde{\mathrm{S}}(\tilde{\boldsymbol{q}}_*))^{-1})$. From this and the definition of $\underline{\eta_\ell}$, there exists $\tilde{\boldsymbol{p}}_* \in \mathrm{int}\,\tilde{\Delta}_n$ such that

$$1 > \frac{\lambda_{\min}(\mathsf{H}\tilde{\Phi}(\tilde{\boldsymbol{q}}_*)(\mathsf{H}\tilde{\mathrm{S}}(\tilde{\boldsymbol{q}}_*))^{-1})}{\lambda_{\max}([\mathsf{H}\underline{L}_{\log}(\tilde{\boldsymbol{p}}_*)]^{-1}\mathsf{H}\underline{L}_\ell(\tilde{\boldsymbol{p}}_*))} = \frac{\lambda_{\min}(\mathsf{H}\tilde{\Phi}(\tilde{\boldsymbol{q}}_*)(\mathrm{diag}\,(\tilde{\boldsymbol{q}}_*) - \tilde{\boldsymbol{q}}_*\tilde{\boldsymbol{q}}_*^{\mathsf{T}}))}{\lambda_{\max}([\mathsf{H}\underline{L}_{\log}(\tilde{\boldsymbol{p}}_*)]^{-1}\mathsf{H}\underline{L}_\ell(\tilde{\boldsymbol{p}}_*))}, \tag{83}$$

where the equality is due to Lemma 22-(iv). For the rest of this proof let $(\tilde{\boldsymbol{p}}, \tilde{\boldsymbol{q}}) = (\tilde{\boldsymbol{p}}^*, \tilde{\boldsymbol{q}}^*)$. By assumption, $\underline{L}_\ell$ twice differentiable and concave on $\mathrm{int}\,\tilde{\Delta}_n$, and thus $-\mathsf{H}\underline{L}_\ell(\tilde{\boldsymbol{p}})$ is symmetric positive semi-definite. Therefore, their exists a symmetric positive semi-definite matrix $\Lambda_{\boldsymbol{p}}$ such that $\Lambda_{\boldsymbol{p}}\Lambda_{\boldsymbol{p}} = -\mathsf{H}\underline{L}_\ell(\tilde{\boldsymbol{p}})$. From Lemma 22-(i), $\tilde{\Phi}$ is strictly convex on $\mathrm{int}\,\tilde{\Delta}_k$, and so there exists a symmetric positive definite matrix $K_{\boldsymbol{q}}$ such that $K_{\boldsymbol{q}}K_{\boldsymbol{q}} = \mathsf{H}\tilde{\Phi}(\tilde{\boldsymbol{q}})$. Let $\boldsymbol{w} \in \mathbb{R}^{n-1}$ be the unit norm eigenvector of $[\mathsf{H}\underline{L}_{\log}(\tilde{\boldsymbol{p}})]^{-1}\underline{L}_\ell(\tilde{\boldsymbol{p}})$ associated with $\lambda_*^\ell := \lambda_{\max}([\mathsf{H}\underline{L}_{\log}(\tilde{\boldsymbol{p}})]^{-1}\mathsf{H}\underline{L}_\ell(\tilde{\boldsymbol{p}}))$. Suppose that $c_\ell := \boldsymbol{w}^{\mathsf{T}}\mathsf{H}\underline{L}_\ell(\tilde{\boldsymbol{p}})\boldsymbol{w} = 0$. Since $\boldsymbol{w}^{\mathsf{T}}\Lambda_{\boldsymbol{p}}\Lambda_{\boldsymbol{p}}\boldsymbol{w} = -c_\ell = 0$, it follows from the positive semi-definiteness of $\Lambda_{\boldsymbol{p}}$ that $\Lambda_{\boldsymbol{p}}\boldsymbol{w} = \boldsymbol{0}_{\tilde{n}}$, and thus $\mathsf{H}\underline{L}_\ell(\tilde{\boldsymbol{p}})\boldsymbol{w} = -\Lambda_{\boldsymbol{p}}\Lambda_{\boldsymbol{p}}\boldsymbol{w} = \boldsymbol{0}_{\tilde{n}}$. This implies that $\lambda_*^\ell = 0$, which is not possible due to (83). Therefore, $\mathsf{H}\underline{L}_\ell(\tilde{\boldsymbol{p}})\boldsymbol{w} \ne \boldsymbol{0}_{\tilde{n}}$. Furthermore, the negative semi-definiteness of $\mathsf{H}\underline{L}_\ell(\tilde{\boldsymbol{p}})$ implies that

$$c_\ell = \boldsymbol{w}^{\mathsf{T}}\mathsf{H}\underline{L}_\ell(\tilde{\boldsymbol{p}})\boldsymbol{w} < 0. \tag{84}$$

Let $\boldsymbol{v} \in \mathbb{R}^{k-1}$ be the unit norm eigenvector of $K_{\boldsymbol{q}}(\mathrm{diag}\,(\tilde{\boldsymbol{q}}) - \tilde{\boldsymbol{q}}\tilde{\boldsymbol{q}}^{\mathsf{T}})K_{\boldsymbol{q}}$ associated with $\lambda_*^\Phi := \lambda_{\min}(K_{\boldsymbol{q}}(\mathrm{diag}\,(\tilde{\boldsymbol{q}}) - \tilde{\boldsymbol{q}}\tilde{\boldsymbol{q}}^{\mathsf{T}})K_{\boldsymbol{q}}) = \lambda_{\min}(\mathsf{H}\tilde{\Phi}(\tilde{\boldsymbol{q}})(\mathrm{diag}\,(\tilde{\boldsymbol{q}}) - \tilde{\boldsymbol{q}}\tilde{\boldsymbol{q}}^{\mathsf{T}}))$, where the equality is due to Lemma 2. Let $\hat{\boldsymbol{v}} := K_{\boldsymbol{q}}\boldsymbol{v}$.

We will show that for $V = \boldsymbol{w}\hat{\boldsymbol{v}}^{\mathsf{T}}$, the parametrized curve $\beta$ defined in Lemma 23 satisfies $\frac{d}{dt}\langle \boldsymbol{p}, \dot{\beta}(t) - \dot{\gamma}(t) \rangle\big|_{t=0} < 0$, where $\gamma(t) = \underline{\ell}(\tilde{\boldsymbol{p}} + tV\tilde{\boldsymbol{q}})$. According to Lemma 24 this would imply that there exists $P \in \mathrm{ri}\,\Delta_n^k$, such that $[\mathrm{Mix}_\Phi(\underline{\ell}_x(P), \boldsymbol{q})]_{x \in [n]}^{\mathsf{T}}$ lies outside $\mathscr{S}_\ell$. From Theorem 5, we know that there exists $A_* \in \mathcal{A}^k$, such that $\ell_x(A_*) = \underline{\ell}_x(P), \forall x \in [n]$. Therefore, $[\mathrm{Mix}_\Phi(\ell_x(A_*), \boldsymbol{q})]_{x \in [n]}^{\mathsf{T}} = [\mathrm{Mix}_\Phi(\underline{\ell}_x(P), \boldsymbol{q})]_{x \in [n]}^{\mathsf{T}} \notin \mathscr{S}_\ell$, and thus $\ell$ is not $\Phi$-mixable.

From Lemma 23 (Equation 79) and the fact that $V_{\cdot,j} = \hat{v}_j\boldsymbol{w}$, for $j \in [\tilde{k}]$, we can write

$$\frac{d}{dt}\left\langle \boldsymbol{p}, \dot{\beta}(t) \right\rangle\bigg|_{t=0} = -\sum_{j=1}^{k-1} q_j\hat{v}_j^2\boldsymbol{w}^{\mathsf{T}}\mathsf{H}\underline{L}_\ell(\tilde{\boldsymbol{p}})\boldsymbol{w} - \mathrm{tr}(\mathrm{diag}\,(\boldsymbol{p})\,\mathsf{D}\tilde{\ell}(\tilde{\boldsymbol{p}})V(\mathsf{H}\tilde{\Phi}(\tilde{\boldsymbol{q}}))^{-1}(\mathsf{D}\tilde{\ell}(\tilde{\boldsymbol{p}})V)^{\mathsf{T}}),$$

$$= -\langle \tilde{\boldsymbol{q}}, \hat{\boldsymbol{v}} \odot \hat{\boldsymbol{v}} \rangle\,\boldsymbol{w}^{\mathsf{T}}\mathsf{H}\underline{L}_\ell(\tilde{\boldsymbol{p}})\boldsymbol{w} - (\hat{\boldsymbol{v}}^{\mathsf{T}}(\mathsf{H}\tilde{\Phi}(\boldsymbol{q}))^{-1}\hat{\boldsymbol{v}})\langle \boldsymbol{p}, [\mathsf{D}\tilde{\ell}(\tilde{\boldsymbol{p}})\boldsymbol{w}] \odot [(\mathsf{D}\tilde{\ell}(\tilde{\boldsymbol{p}})\boldsymbol{w}]\rangle,$$

where the second equality is obtained by noting that 1) $(\hat{v}^\mathsf{T}(\mathsf{H}\tilde{\Phi}(q))^{-1}\hat{v})$ is a scalar quantity and can be factorized out; and 2) $\mathrm{tr}(\mathrm{diag}(p)\mathsf{D}\tilde{\ell}(\tilde{p})w(\mathsf{D}\tilde{\ell}(\tilde{p})w)^\mathsf{T}) = \langle p, (\mathsf{D}\tilde{\ell}(\tilde{p})w) \odot (\mathsf{D}\tilde{\ell}(\tilde{p})w)\rangle$.

On the other hand, from Lemma 9, $\frac{d}{dt}\langle p, \dot{\gamma}(t)\rangle\big|_{t=0} = -\langle \tilde{q}, \hat{v}\rangle^2 w^\mathsf{T}\mathsf{H}\tilde{\underline{L}}_\ell(\tilde{q})w$. Using (5) and the definition of $c_\ell$, we get

$$
\begin{aligned}
\frac{d}{dt}\left\langle p, \dot{\beta}(t) - \dot{\gamma}(t)\right\rangle\bigg|_{t=0} &= [-\langle \tilde{q}, \hat{v}\odot\hat{v}\rangle + \langle\tilde{q},\hat{v}\rangle^2]c_\ell + \\
&\qquad (\hat{v}^\mathsf{T}(\mathsf{H}\tilde{\Phi}(q))^{-1}\hat{v})(w^\mathsf{T}(\mathsf{H}\tilde{\underline{L}}_\ell(\tilde{p}))(\mathsf{H}\tilde{\underline{L}}_{\log}(\tilde{p}))^{-1}\mathsf{H}\tilde{\underline{L}}_\ell(p)w), \\
&= -c_\ell[\langle\tilde{q},\hat{v}\odot\hat{v}\rangle - \langle\tilde{q},\hat{v}\rangle^2 - \lambda_*^\ell(\hat{v}^\mathsf{T}(\mathsf{H}\tilde{\Phi}(q))^{-1}\hat{v})], \\
&= -c_\ell[\hat{v}^\mathsf{T}(\mathrm{diag}\,(\tilde{q}) - \tilde{q}\tilde{q}^\mathsf{T})\hat{v} - \lambda_*^\ell(\hat{v}^\mathsf{T}(\mathsf{H}\tilde{\Phi}(q))^{-1}\hat{v})], \\
&= -c_\ell[\hat{v}^\mathsf{T}(\mathrm{diag}\,(\tilde{q}) - \tilde{q}\tilde{q}^\mathsf{T})\hat{v} - \lambda_*^\ell(v^\mathsf{T}K_q(K_qK_q)^{-1}K_qv)], \\
&= -c_\ell[v^\mathsf{T}K_q(\mathrm{diag}\,(\tilde{q}) - \tilde{q}\tilde{q}^\mathsf{T})K_qv - \lambda_*^\ell], \qquad (85)\\
&= -c_\ell[\lambda_*^\Phi - \lambda_*^\ell], \\
&= -c_\ell[\lambda_{\min}(\mathsf{H}\tilde{\Phi}(q)(\mathrm{diag}\,(\tilde{q}) - \tilde{q}\tilde{q}^\mathsf{T})) - \lambda_{\max}(\mathsf{H}\tilde{\underline{L}}_\ell(\tilde{p})(\mathsf{H}\tilde{\underline{L}}_{\log}(\tilde{p}))^{-1})],
\end{aligned}
$$

where in (85) we used the fact that $v^\mathsf{T}v = 1$. The last equality combined with (83) and (84) shows that $\frac{d}{dt}\langle p, \dot{\beta}(t) - \dot{\gamma}(t)\rangle\big|_{t=0} < 0$, which completes the proof. $\qquad\square$

## C.8 Proof of Lemma 15

**Lemma 15** *Let $\ell\colon \mathcal{A} \to [0, +\infty]^n$ be a loss. If $\mathrm{dom}\,\ell = \mathcal{A}$, then either $\mathfrak{H}_\ell = \varnothing$ or $\eta_\ell \in \mathfrak{H}_\ell$.*

*Proof.* Suppose $\mathfrak{H}_\ell \neq \varnothing$. Let $q \in \Delta_k$, $A := a_{1:k} \in \mathcal{A}^k$. By definition of $\eta_\ell$ there exists $(\eta_m) \subset [0, +\infty[$ such that $\ell$ is $\eta_m$-mixable and $\eta_m \overset{m\to\infty}{\to} \eta_\ell$. Therefore, $\forall m \in \mathbb{N}$, $\exists a_m \in \mathcal{A}$ such that

$$
\forall x \in [n],\ \ell_x(a_m) \le -\eta_m^{-1}\log\langle q, \exp(-\eta_m(\ell_x(A)))\rangle < +\infty, \qquad (86)
$$

where the right-most inequality follows from the fact $\mathrm{dom}\,\ell = \mathcal{A}$. Therefore, the sequence $(\ell(a_m)) \subset [0, +\infty[^n$ is bounded, and thus admits a convergent subsequence. If we let $s$ be the limit of this subsequence, then from (86) it follows that

$$
\forall x \in [n],\ s \le -\eta_\ell^{-1}\log\langle q, \exp(-\eta_\ell(\ell_x(A)))\rangle, \qquad (87)
$$

On the other hand, since $\ell$ is closed (by Assumption 1), it follows that there exists $a_* \in \mathcal{A}$ such that $\ell(a_*) = s$. Combining this with (87) implies that $\ell$ is $\eta_\ell$-mixable, and thus $\eta_\ell \in \mathfrak{H}_\ell$. $\qquad\square$

## C.9 Proof of Theorem 17

**Theorem 17** *Let $\ell$ and $\Phi$ be as in Theorem 16. Then*

$$
\eta_\ell^\Phi = \underline{\eta_\ell}\inf_{\tilde{q}\in\mathrm{int}\,\tilde{\Delta}_k}\lambda_{\min}(\mathsf{H}\tilde{\Phi}(\tilde{q})(\mathsf{H}\tilde{S}(\tilde{q}))^{-1}),
$$

*Proof.* From Theorem 16, $\ell$ is $\Phi_\eta$-mixable if and only if $\underline{\eta_\ell}\Phi_\eta - S = \eta^{-1}\underline{\eta_\ell}\Phi - S$ is convex on $\Delta_k$. When this is the case, Lemma 3 implies that

$$
1 \le \eta^{-1}\underline{\eta_\ell}(\inf_{\tilde{q}\in\mathrm{int}\,\tilde{\Delta}_k}\lambda_{\min}[\mathsf{H}\tilde{\Phi}(\tilde{q})[\mathsf{H}\tilde{S}(\tilde{q})]^{-1}]), \qquad (88)
$$

where we used the facts that $\mathsf{H}(\eta^{-1}\underline{\eta_\ell}\tilde{\Phi}) = \eta^{-1}\underline{\eta_\ell}\mathsf{H}\tilde{\Phi}$, $\lambda_{\min}(\cdot)$ is linear, and $\eta^{-1}\underline{\eta_\ell}$ is independent of $\tilde{q} \in \mathrm{int}\,\tilde{\Delta}_k$. Inequality 88 shows that the largest $\eta$ such that $\ell$ is $\Phi_\eta$-mixable is given by $\eta_\ell^\Phi$ in (11). $\qquad\square$

## C.10 Proof of Theorem 18

**Theorem 18** *Let* $S, \Phi \colon \mathbb{R}^k \to \mathbb{R} \cup \{+\infty\}$*, where* $S$ *is the Shannon entropy and* $\Phi$ *is an entropy such that* $\tilde{\Phi} := \Phi \circ \mathrm{II}_k$ *is twice differentiable on* $\mathrm{int}\, \tilde{\Delta}_k$. *A loss* $\ell \colon \mathcal{A} \to [0, +\infty[^n$, *with* $\underline{L}_\ell$ *twice differentiable on* $]0, +\infty[^n$, *is* $\Phi$*-mixable only if* $R_\ell^S \leq R_\ell^\Phi$.

*Proof.* Suppose $\ell$ is $\Phi$-mixable. Then from Theorem 16, $\underline{\eta_\ell}\Phi - S$ is convex on $\Delta_k$, and thus $\underline{\eta_\ell} = \eta_\ell^S > 0$ (Corollary 17). Furthermore, $\underline{\eta_\ell}\tilde{\Phi} - \tilde{S} = [\underline{\eta_\ell}\Phi - S] \circ \mathrm{II}_k$ is convex on $\mathrm{int}\, \tilde{\Delta}_k$, since $\mathrm{II}_k$ is an affine function. It follows from Lemma 3 and Corollary 17 that

$$\eta_\ell^\Phi = \underline{\eta_\ell} \inf_{\tilde{q} \in \mathrm{int}\, \tilde{\Delta}_k} \lambda_{\min}(\mathsf{H}\tilde{\Phi}(\tilde{q})(\mathsf{H}\tilde{S}(\tilde{q}))^{-1}) \geq 1 > 0.$$

Let $\boldsymbol{\mu} \in \mathrm{ri}\, \Delta_k$ and $\theta_* := \mathrm{argmax}_\theta D_S(\boldsymbol{e}_\theta, \boldsymbol{\mu})$. By definition of an entropy and the fact that the directional derivatives $\Phi'(\boldsymbol{\mu}; \cdot)$ and $S'(\boldsymbol{\mu}; \cdot)$ are finite on $\Delta_k$ [7, Prop. D.1.1.2], it holds that $D_\Phi(\boldsymbol{e}_{\theta_*}, \boldsymbol{\mu}), D_S(\boldsymbol{e}_{\theta_*}, \boldsymbol{\mu}) \in ]0, +\infty[$. Therefore, there exists $\alpha > 0$ such that $\alpha^{-1} D_\Phi(\boldsymbol{e}_{\theta_*}, \boldsymbol{\mu}) = D_S(\boldsymbol{e}_{\theta_*}, \boldsymbol{\mu})$. If we let $\Psi := \alpha^{-1}\Phi$, we get

$$D_\Psi(\boldsymbol{e}_{\theta_*}, \boldsymbol{\mu}) = D_S(\boldsymbol{e}_{\theta_*}, \boldsymbol{\mu}). \tag{89}$$

Let $d_\Psi(\tilde{q}) := \tilde{\Psi}(\tilde{q}) - \tilde{\Psi}(\tilde{\boldsymbol{\mu}}) - \langle \tilde{q} - \tilde{\boldsymbol{\mu}}, \nabla\tilde{\Psi}(\tilde{\boldsymbol{\mu}}) \rangle$. Observe that

$$d_\Psi(\tilde{q}) = \Psi(q) - \Psi(\boldsymbol{\mu}) - \langle q - \boldsymbol{\mu}, \nabla\Psi(\boldsymbol{\mu}) \rangle = D_\Psi(q, \boldsymbol{\mu}).$$

We define $d_S$ similarly. Suppose that $\eta_\ell^\Psi > \eta_\ell^S = \underline{\eta_\ell}$. Then, from Corollary 17, $\forall \tilde{q} \in \mathrm{int}\, \tilde{\Delta}_k$, $\lambda_{\min}(\mathsf{H}\tilde{\Psi}(\tilde{q})(\mathsf{H}\tilde{S}(\tilde{q}))^{-1}) > 1$. This implies that $\forall \tilde{q} \in \mathrm{int}\, \tilde{\Delta}_k$, $\lambda_{\min}(\mathsf{H}d_\Psi(\tilde{q})(\mathsf{H}d_S(\tilde{q}))^{-1}) > 1$, and from Lemma 3, $d_\Psi - d_S$ must be strictly convex on $\mathrm{int}\, \tilde{\Delta}_k$. We also have $\nabla d_\Psi(\tilde{\boldsymbol{\mu}}) - \nabla d_S(\tilde{\boldsymbol{\mu}}) = 0$ and $d_\Psi(\tilde{\boldsymbol{\mu}}) - d_S(\tilde{\boldsymbol{\mu}}) = 0$. Therefore, $d_\Psi - d_S$ attains a strict minimum at $\tilde{\boldsymbol{\mu}}$ (ibid., Thm. D.2.2.1); that is, $d_\Psi(\tilde{q}) > d_S(\tilde{q})$, $\forall \tilde{q} \in \tilde{\Delta}_k \setminus \{\tilde{\boldsymbol{\mu}}\}$. In particular, for $\tilde{q} = \Pi_k(\boldsymbol{e}_{\theta_*})$, we get $D_\Psi(\boldsymbol{e}_{\theta_*}, \boldsymbol{\mu}) = d_\Psi(\tilde{q}) > d_S(\tilde{q}) = D_S(\boldsymbol{e}_{\theta_*}, \boldsymbol{\mu})$, which contradicts (89). Therefore, $\eta_\ell^\Psi \leq \eta_\ell^S$, and thus

$$\begin{aligned} R_\ell^S(\boldsymbol{\mu}) = \mathrm{max}_\theta D_S(\boldsymbol{e}_\theta, \boldsymbol{\mu})/\eta_\ell^S &= D_S(\boldsymbol{e}_{\theta_*}, \boldsymbol{\mu})/\eta_\ell^S, \\ &\leq D_\Psi(\boldsymbol{e}_{\theta_*}, \boldsymbol{\mu})/\eta_\ell^\Psi, \qquad\qquad (90) \\ &\leq \mathrm{max}_\theta D_\Psi(\boldsymbol{e}_\theta, \boldsymbol{\mu})/\eta_\ell^\Psi, \\ &= R_\ell^\Psi(\boldsymbol{\mu}), \qquad\qquad\qquad\qquad (91) \end{aligned}$$

where (90) is due to $D_\Psi(\boldsymbol{e}_{\theta_*}, \boldsymbol{\mu}) = D_S(\boldsymbol{e}_{\theta_*}, \boldsymbol{\mu})$ and $\eta_\ell^\Psi \leq \eta_\ell^S$. Equation 91, implies that $R_\ell^S(\boldsymbol{\mu}) \leq R_\ell^\Phi(\boldsymbol{\mu})$, since $R_\ell^\Psi(\boldsymbol{\mu}) = R_\ell^{\alpha\Phi}(\boldsymbol{\mu}) = R_\ell^\Phi(\boldsymbol{\mu})$ [9]. Therefore,

$$\forall \boldsymbol{\mu} \in \mathrm{ri}\, \Delta_k, \ R_\ell^S(\boldsymbol{\mu}) \leq R_\ell^\Phi(\boldsymbol{\mu}). \tag{92}$$

It remains to consider the case where $\boldsymbol{\mu}$ is in the relative boundary of $\Delta_k$. Let $\boldsymbol{\mu} \in \mathrm{rbd}\, \Delta_k$. There exists $\mathfrak{l}_0 \subsetneq [k]$ such that $\boldsymbol{\mu} \in \Delta_{\mathfrak{l}_0}$. Let $\theta^* \in [k] \setminus \mathfrak{l}_0$ and $\mathfrak{l} := \mathfrak{l}_0 \cup \{\theta^*\}$. It holds that $\boldsymbol{\mu} \in \mathrm{rbd}\, \Delta_\mathfrak{l}$ and $\boldsymbol{\mu} + 2^{-1}(\boldsymbol{e}_{\theta^*} - \boldsymbol{\mu}) \in \mathrm{ri}\, \Delta_\mathfrak{l}$. Since $\ell$ is $\Phi$-mixable, it follows from Proposition 10 and the 1-homogeneity of $\Phi'(\boldsymbol{\mu}; \cdot)$ [7, Prop. D.1.1.2] that

$$\Phi'(\boldsymbol{\mu}; \boldsymbol{e}_{\theta^*} - \boldsymbol{\mu}) = 2\Phi'(\boldsymbol{\mu}; [\boldsymbol{\mu} + 2^{-1}(\boldsymbol{e}_{\theta^*} - \boldsymbol{\mu})] - \boldsymbol{\mu}) = -\infty.$$

Hence,

$$\begin{aligned} R_\ell^\Phi(\boldsymbol{\mu}) &= \mathrm{max}_{\theta \in [k]} D_\Phi(\boldsymbol{e}_\theta, \boldsymbol{\mu}), \\ &\geq D_\Phi(\boldsymbol{e}_{\theta^*}, \boldsymbol{\mu}) = \Phi(\boldsymbol{e}_{\theta^*}) - \Phi(\boldsymbol{\mu}) - \Phi'(\boldsymbol{\mu}; \boldsymbol{e}_{\theta^*} - \boldsymbol{\mu}) = +\infty. \qquad (93) \end{aligned}$$

Inequality 93 also applies to $S$, since $\ell$ is $(\underline{\eta_\ell}^{-1} S)$-mixable. From (93) and (92), we conclude that $\forall \boldsymbol{\mu} \in \Delta_k, \ R_\ell^S(\boldsymbol{\mu}) \leq R_\ell^\Phi(\boldsymbol{\mu})$. $\qquad\qquad\qquad\qquad\qquad\qquad\qquad\qquad \square$

## C.11 Proof of Theorem 19

**Theorem 19** *Let* $\Phi : \mathbb{R}^k \to \mathbb{R} \cup \{+\infty\}$ *be a $\Delta$-differentiable entropy. Let $\ell \colon \mathcal{A} \to [0, +\infty]^n$ be a loss such that $\underline{L}_\ell$ is twice differentiable on $]0, +\infty[^n$. Let $\boldsymbol{\beta}^t = -\eta \sum_{s=1}^{t-1} (\ell_{x^s}(A^s) + \boldsymbol{v}^s)$, where $\boldsymbol{v}^s \in \mathbb{R}^k$ and $A^s := \boldsymbol{a}_{1:k}^s \in \mathcal{A}^k$. If $\ell$ is $(\eta, \Phi)$-mixable then for initial distribution $\boldsymbol{q}^0 = \operatorname{argmin}_{\boldsymbol{q} \in \Delta_k} \max_{\theta \in [k]} D_\Phi(\boldsymbol{e}_\theta, \boldsymbol{q})$ and any sequence $(x^t, \boldsymbol{a}_{1:k}^t)_{t=1}^T$, the AGAA achieves the regret*

$$\forall \theta \in [k], \quad \operatorname{Loss}_{AGAA}^\ell(T) - \operatorname{Loss}_\theta^\ell(T) \leq R_\ell^\Phi + \sum_{t=1}^{T-1} (v_\theta^t - \langle \boldsymbol{v}^t, \boldsymbol{q}^t \rangle).$$

*Proof.* Recall that $\Phi_t(\boldsymbol{w}) := \Phi(\boldsymbol{w}) - \langle \boldsymbol{w}, \boldsymbol{\beta}^t - \boldsymbol{\theta}^t \rangle$, where $\boldsymbol{\theta}^t = -\eta \sum_{s=1}^{t-1} \ell_{x^s}(A^s)$. From Theorem 16 and since $\Phi_t$ is equal to $\Phi$ plus an affine function, it is clear that if $\ell$ is $(\eta, \Phi)$-mixable then $\ell$ is $(\eta, \Phi_t)$-mixable. Thus, for all $(A^t, \boldsymbol{q}^{t-1}) \in \mathcal{A}^k \times \Delta_k$, there exists $\boldsymbol{a}_*^t \in \mathcal{A}$ such that for any outcome $x^t \in [n]$

$$\ell_{x^t}(\boldsymbol{a}_*^t) \leq \eta^{-1}[\Phi_t^\star(\nabla\Phi_t(\boldsymbol{q}^{t-1})) - \Phi_t^\star(\nabla\Phi_t(\boldsymbol{q}^{t-1}) - \eta\ell_{x^t}(A^t))].$$

Summing over $t$ from 1 to $T$, we get

$$\sum_{t=1}^T \ell_{x^t}(\boldsymbol{a}_*^t) \leq \eta^{-1}[\Phi_1^\star(\nabla\Phi_1(\boldsymbol{q}^0)) - \Phi_T^\star(\nabla\Phi_T(\boldsymbol{q}^{T-1}) - \eta\ell_{x^T}(A^T))] \tag{94}$$

$$+ \eta^{-1} \sum_{t=1}^{T-1} \left[ \Phi_{t+1}^\star(\nabla\Phi_{t+1}(\boldsymbol{q}^t)) - \Phi_t^\star(\nabla\Phi_t(\boldsymbol{q}^{t-1}) - \eta\ell_{x^t}(A^t)) \right].$$

ODue to the properties of the entropic dual [9] and the definition of $\Phi_t$, the following holds for all $t \in [T]$ and $\boldsymbol{z}$ in $\mathbb{R}^k$,

$$\nabla\Phi_t(\boldsymbol{q}^{t-1}) = -\eta \sum_{s=1}^{t-1} \ell_{x^s}(A^s), \tag{95}$$

$$\Phi_t^\star(\boldsymbol{z}) = \Phi^\star(\boldsymbol{z} + \nabla\Phi(\boldsymbol{q}^{t-1}) + \eta \sum_{s=1}^{t-1} \ell_{x^s}(A^s)), \tag{96}$$

$$\nabla\Phi(\boldsymbol{q}^t) = \nabla\Phi(\boldsymbol{q}^{t-1}) - \eta\ell_{x^t}(A^t) - \eta\boldsymbol{v}^t. \tag{97}$$

Using (95)-(96), we get for all $0 \leq t < T$, $\Phi_{t+1}^\star(\nabla\Phi_{t+1}(\boldsymbol{q}^t)) = \Phi^\star(\nabla\Phi(\boldsymbol{q}^t))$, and in particular $\Phi_1^\star(\nabla\Phi_1(\boldsymbol{q}^0)) = \Phi^\star(\nabla\Phi(\boldsymbol{q}^0))$. Similarly, using (95)-(97), gives $\Phi_t^\star(\nabla\Phi_t(\boldsymbol{q}^{t-1}) - \eta\ell_{x^t}(A^t)) = \Phi^\star(\nabla\Phi(\boldsymbol{q}^t) + \eta\boldsymbol{v}^t)$ for all $1 \leq t \leq T$. Substituting back into (94) yields

$$\sum_{t=1}^T \ell_{x^t}(\boldsymbol{a}_*^t) \leq \eta^{-1}[\Phi^\star(\nabla\Phi(\boldsymbol{q}^0)) - \Phi^\star(\nabla\Phi(\boldsymbol{q}^T) + \eta\boldsymbol{v}^T)]$$

$$+ \eta^{-1} \sum_{t=1}^{T-1} \left[ \Phi^\star(\nabla\Phi(\boldsymbol{q}^t)) - \Phi^\star(\nabla\Phi(\boldsymbol{q}^t) + \eta\boldsymbol{v}^t) \right], \tag{98}$$

To conclude the proof, we note that since $\Phi$ is convex it holds that

$$\Phi^\star(\nabla\Phi(\boldsymbol{q}^t)) - \Phi^\star(\nabla\Phi(\boldsymbol{q}^t) + \eta\boldsymbol{v}^t) \leq -\eta\langle \boldsymbol{v}^t, \nabla\Phi^\star(\nabla\Phi(\boldsymbol{q}^t)) \rangle = -\eta\langle \boldsymbol{v}^t, \boldsymbol{q}^t \rangle, \tag{99}$$

which allows us to bound the sum on the right hand side of 98. To bound the rest of the terms, we use the fact that $\nabla\Phi(\boldsymbol{q}^T) = \nabla\Phi(\boldsymbol{q}^0) - \eta \sum_{t=1}^T (\ell_{x^t}(A^t) + \boldsymbol{v}^t)$, and thus by letting $\Phi_\eta := \eta^{-1}\Phi$,

$$\eta^{-1}[\Phi^\star(\nabla\Phi(\boldsymbol{q}^0)) - \Phi^\star(\nabla\Phi(\boldsymbol{q}^T) + \eta\boldsymbol{v}^T)] = \Phi_\eta^\star(\nabla\Phi_\eta(\boldsymbol{q}^0))$$

$$- \Phi_\eta^\star \left( \nabla\Phi_\eta(\boldsymbol{q}^0) - \sum_{t=1}^T \ell_{x^t}(A^t) - \sum_{t=1}^{T-1} \boldsymbol{v}^t \right),$$

$$= \inf_{\boldsymbol{q} \in \Delta_k} \left\langle \boldsymbol{q}, \sum_{t=1}^T \ell_{x^t}(A^t) + \sum_{t=1}^{T-1} \boldsymbol{v}^t \right\rangle + \frac{D_\Phi(\boldsymbol{q}, \boldsymbol{q}^0)}{\eta},$$

$$\leq \sum_{t=1}^T \ell_{x^t}(\boldsymbol{a}_\theta^t) + \sum_{t=1}^{T-1} v_\theta^t + \frac{D_\Phi(\boldsymbol{e}_\theta, \boldsymbol{q}^0)}{\eta}, \forall \theta \in [k].$$

Substituting this last inequality and (99) back into (98) yields the desired bound. $\qquad\square$

# D Defining the Bayes Risk Using the Superprediction Set

In this section, we argue that when a loss $\ell\colon \mathcal{A} \to [0,+\infty]^n$ is mixable, in the classical or generalized sense, it does not matter whether we define the Bayes risk $\underline{L}_\ell$ using the full superprediction set $\mathscr{S}_\ell^\infty$ or its finite part $\mathscr{S}_\ell$. Recall the definition of the Bayes risk;

**Definition 2** *Let $\ell\colon \mathcal{A} \to [0,+\infty]^n$ be a loss such that $\operatorname{dom}\ell \neq \varnothing$. The* Bayes risk $\underline{L}_\ell : \mathbb{R}^n \to \mathbb{R} \cup \{-\infty\}$ *is defined by*

$$\forall \boldsymbol{u} \in \mathbb{R}^n, \quad \underline{L}_\ell(\boldsymbol{u}) := \inf_{\boldsymbol{z} \in \mathscr{S}_\ell} \langle \boldsymbol{u}, \boldsymbol{z} \rangle. \tag{100}$$

Note that the right hand side of (100) does not change if we substitute $\mathscr{S}_\ell$ for its closure — $\overline{\mathscr{S}_\ell}$ — with respect to $[0,+\infty]^n$. Thus, it suffices to show that $\mathscr{S}_\ell^\infty \subseteq \overline{\mathscr{S}_\ell}$ when the loss $\ell$ is mixable. We show this in Theorem 26, but first we give a characterization of the (finite part) of the superprediction set for a proper loss.

**Proposition 25.** *Let $\ell\colon \Delta_n \to [0,+\infty]^n$ be a proper loss. If $\underline{L}_\ell$ is differentiable on $]0,+\infty[^n$, then*

$$\overline{\mathscr{S}_\ell} \supseteq \mathcal{C}_\ell := \{\boldsymbol{u} \in [0,+\infty]^n : \forall \boldsymbol{p} \in \Delta_n, \underline{L}_\ell(\boldsymbol{p}) \leq \langle \boldsymbol{p}, \boldsymbol{u} \rangle\}. \tag{101}$$

*Proof.* Let $\boldsymbol{v} \in \mathcal{C}_\ell \cap [0,+\infty[^n$. Let $f\colon \operatorname{ri}\Delta_n \times [n] \to \mathbb{R}$ be defined by $f(\boldsymbol{p}, x) := \ell_x(\boldsymbol{p}) - v_x$. By the choice of $\boldsymbol{v}$, we have $\mathbb{E}_{x \sim \boldsymbol{p}} f(\boldsymbol{p}, x) = \langle \boldsymbol{p}, \ell(\boldsymbol{p}) \rangle - \langle \boldsymbol{p}, \boldsymbol{v} \rangle \leq 0$ for all $\boldsymbol{p} \in \Delta_n$. Since $\underline{L}_\ell$ is differentiable on $]0,+\infty[^n$, by assumption, $\ell$ is continuous on $\operatorname{ri}\Delta_n$, and thus $f$ is continuous in the first argument. Since $\boldsymbol{v}$ has finite components, the map $f$ satisfies all the conditions of Lemma 5. Therefore, there exists $(\boldsymbol{p}_m) \subset \operatorname{ri}\Delta_n$ such that

$$\forall m \in \mathbb{N}, \forall x \in [n], \ \ell_x(\boldsymbol{p}_m) \leq v_x + \frac{1}{m}. \tag{102}$$

Without loss of generality, we can assume by extracting a subsequence if necessary that $\ell(\boldsymbol{p}_m)$ converges to $\boldsymbol{s} \in [0,+\infty]^n$. By definition, we have $\boldsymbol{s} \in \overline{\mathscr{S}_\ell}$, and from (102) it follows that $\boldsymbol{s} \leq \boldsymbol{v}$ coordinate-wise. Thus, $\boldsymbol{v}$ is in $\overline{\mathscr{S}_\ell}$.

The above argument shows that $\mathcal{C}_\ell \cap [0,+\infty[^n \subseteq \overline{\mathscr{S}_\ell}$, and since $\overline{\mathscr{S}_\ell}$ is closed in $[0,+\infty]^n$ we have $\overline{\mathcal{C}} \subseteq \overline{\mathscr{S}_\ell}$, where $\overline{\mathcal{C}}$ is the closure of $\mathcal{C}_\ell \cap [0,+\infty[^n$ in $[0,+\infty]^n$. Now it suffice to show that $\mathcal{C}_\ell \subseteq \overline{\mathcal{C}}$ to complete the proof.

Let $\boldsymbol{u} \in \mathcal{C}_\ell$ and $\mathfrak{l} := \{x \in [n] : u_x < +\infty\}$. Define $(\boldsymbol{u}_m) \subset [0,+\infty[^n$ by $u_{m,x} = u_x$ if $x \in \mathfrak{l}$; and $m$ otherwise. Let $\boldsymbol{p} \in \Delta_n$. It follows that

$$\begin{aligned} \langle \boldsymbol{p}, \boldsymbol{u}_m \rangle &= \sum_{x' \in \mathfrak{l}} p_{x'} u_{m,x'} + \sum_{x \notin \mathfrak{l}} p_x u_{m,x}, \\ &= \sum_{x' \in \mathfrak{l}} p_{x'} u_{m,x'} + \sum_{x \notin \mathfrak{l}} p_x u_{m,x}, . \end{aligned} \tag{103}$$

**Claim 2.** $\forall \epsilon > 0, \exists m_\epsilon \geq 1, \forall \boldsymbol{p} \in \Delta_k, \underline{L}_\ell(\boldsymbol{p}) \leq \langle \boldsymbol{p}, \boldsymbol{u}_{m_\epsilon} \rangle - \epsilon$.

Suppose that Claim 2 is false. This means that there exists $\delta > 0$ such that

$$\forall \boldsymbol{m} \geq 1, \exists \boldsymbol{p}_m \in \Delta_n, \langle \boldsymbol{p}_m, \boldsymbol{u}_m \rangle - \delta < \underline{L}_\ell(\boldsymbol{p}_m). \tag{104}$$

We may assume, by extracting a subsequence if necessary ($\Delta_n$ is compact), that $(\boldsymbol{p}_m)$ converges to $\boldsymbol{p}_* \in \Delta_n$. Taking the limit $m \to \infty$ in (104) would lead to the contradiction '$\langle \boldsymbol{p}_*, \boldsymbol{u} \rangle < \underline{L}_\ell(\boldsymbol{p}_*)$', since from (103) we have $\lim_{m \to \infty} \langle \boldsymbol{p}_m, \boldsymbol{u}_m \rangle = \langle \boldsymbol{p}_*, \boldsymbol{u} \rangle$. Therefore, Claim 2 is true. For $\epsilon = \frac{1}{k}$ let $m_k := m_\epsilon$ be as in Claim (2). The claim then implies that $\liminf_{k \to \infty} \langle \boldsymbol{p}, \boldsymbol{u}_{m_k} \rangle \geq \underline{L}_\ell(\boldsymbol{p})$ uniformly for $\boldsymbol{p} \in \Delta_k$. By the claim we also have that $\boldsymbol{u}_{m_k} \in \mathcal{C}_\ell \cap [0,+\infty[^n$ for all $k \in \mathbb{N}$, and by construction of $\boldsymbol{v}_m$, we have $\lim_{k \to \infty} \boldsymbol{u}_{m_k} = \boldsymbol{u}$. This shows that $\mathcal{C}_\ell \subseteq \overline{\mathcal{C}}$, which completes the proof.

$\square$

**Theorem 26.** *Let $\ell : \mathcal{A} \to [0,+\infty]^n$ be a loss. If $\mathscr{S}_\ell^\infty \nsubseteq \overline{\mathscr{S}_\ell}$, then $\ell$ is not mixable.*

*Proof.* Suppose that $\ell$ is mixable and let $\underline{\ell}$ be a proper support loss of $\ell$. From Proposition 12, $\underline{L}_\ell$ is differentiable on $]0, +\infty[^n$, and thus Theorem 5 implies that $\overline{\mathscr{S}_\ell} = \overline{\mathscr{S}_{\underline{\ell}}}$. Therefore, Lemma 25 implies that $\overline{\mathscr{S}_\ell} \supseteq \{\boldsymbol{u} \in [0, +\infty]^n : \forall \boldsymbol{p} \in \Delta_n, \underline{L}_\ell(\boldsymbol{p}) \leq \langle \boldsymbol{p}, \boldsymbol{u} \rangle\}$. Thus, if $\mathscr{S}_\ell^\infty \not\subseteq \overline{\mathscr{S}_\ell}$, there exists $\epsilon > 0$, $\boldsymbol{p}_\epsilon \in \Delta_k$, and $\boldsymbol{s} \in \mathscr{S}_\ell^\infty \setminus \overline{\mathscr{S}_\ell}$ such that

$$\langle \boldsymbol{p}_\epsilon, \boldsymbol{s} \rangle < \underline{L}_\ell(\boldsymbol{p}_\epsilon) - 2\epsilon. \tag{105}$$

Note that $\boldsymbol{p}_\epsilon$ cannot be in $\operatorname{ri} \Delta_n$; otherwise, (105) would imply that $\boldsymbol{s}$ has all finite components, and thus would be included in $\overline{\mathscr{S}_\ell}$, which is a contradiction. Assume from now on that $\boldsymbol{p}_\epsilon \in \operatorname{rbd} \Delta_n$. From the definition of the support loss, there exists a sequence $(\boldsymbol{p}_m) \subseteq \operatorname{ri} \Delta_n$ such that $\boldsymbol{p}_m \overset{m\to\infty}{\to} \boldsymbol{p}_\epsilon$ and $\underline{\ell}(\boldsymbol{p}_m) \overset{m\to\infty}{\to} \underline{\ell}(\boldsymbol{p}_\epsilon)$. Therefore, Theorem 5 implies that there exists $\boldsymbol{a}_\epsilon \in \mathcal{A}$ such that

$$\langle \boldsymbol{p}_\epsilon, \ell(\boldsymbol{a}_\epsilon) \rangle < \langle \boldsymbol{p}_\epsilon, \underline{\ell}(\boldsymbol{p}_\epsilon) \rangle + \epsilon. \tag{106}$$

To see this, note that since $(\boldsymbol{p}_m) \subset \operatorname{ri} \Delta_n \subseteq \operatorname{dom} \underline{\ell}$, Theorem 5 guarantees the existence of a sequence $(\boldsymbol{a}_m) \subset \mathcal{A}$ such that $\ell(\boldsymbol{a}_m) = \underline{\ell}(\boldsymbol{p}_m)$. On the other hand, for any $x \in [n]$ such that $\ell_x(\boldsymbol{p}_\epsilon) = +\infty$, we have $p_{\epsilon,x} = 0$ — otherwise, $\underline{L}_\ell(\boldsymbol{p}_\epsilon)$ would be infinite. It follows, by continuity of the inner product that $\langle \boldsymbol{p}_\epsilon, \ell(\boldsymbol{a}_m) \rangle \overset{m\to\infty}{\to} \langle \boldsymbol{p}_\epsilon, \underline{\ell}(\boldsymbol{p}_m) \rangle$, and thus it suffices to pick $\boldsymbol{a}_\epsilon$ equal to $\boldsymbol{a}_m$ for $m$ large enough.

Now since $\ell$ is $\eta$-mixable, there exists $\eta > 0$ and $\boldsymbol{a}_* \in \mathcal{A}$ such that

$$\ell(\boldsymbol{a}_*) \leq -\eta^{-1} \log\left(\frac{1}{2} e^{-\eta \boldsymbol{s}} + \frac{1}{2} e^{-\eta \ell(\boldsymbol{a}_\epsilon)}\right),$$

and due to the convexity of $-\log$,

$$\leq \frac{1}{2}\boldsymbol{s} + \frac{1}{2}\ell(\boldsymbol{a}_\epsilon).$$

Using (105) and (106) yields

$$\langle \boldsymbol{p}_\epsilon, \ell(\boldsymbol{a}_*) \rangle \leq \underline{L}_\ell(\boldsymbol{p}_\epsilon) - \epsilon/2. \tag{107}$$

On the other hand, by definition of a proper support loss, $\langle \boldsymbol{p}_\epsilon, \underline{\ell}(\boldsymbol{p}_\epsilon) \rangle \leq \langle \boldsymbol{p}_\epsilon, \ell(\boldsymbol{a}_*) \rangle$. This combined with (107), lead to the contradiction $\langle \boldsymbol{p}_\epsilon, \underline{\ell}(\boldsymbol{p}_\epsilon) \rangle < \underline{L}_\ell(\boldsymbol{p}_\epsilon)$. $\qquad\square$

# E   The Update Step of the GAA and the Mirror Descent Algorithm

In this section, we demonstrate that the update steps of the GAA and the Mirror Descent Algorithm are essentially the same (at least for finite losses) according to the definition of the MDA given by Beck and Teboulle [2];

Let $\ell\colon \mathcal{A} \to [0, +\infty[^n$ be a loss and $\Phi\colon \mathbb{R}^k \to \mathbb{R} \cup \{+\infty\}$ an entropy such that $\tilde{\Phi}$ is differentiable on $\operatorname{int} \tilde{\Delta}_k$. Let $\boldsymbol{q}^t$ be the update distribution of the GAA at round $t$ and $\tilde{\boldsymbol{q}}^t = \Pi_k(\boldsymbol{q}^t)$. It follows from the definition of $\boldsymbol{q}^t$ (see Algorithm 2) that

$$\tilde{\boldsymbol{q}}^t = \underset{\tilde{\boldsymbol{q}} \in \tilde{\Delta}_k}{\operatorname{argmin}} \left\langle \Pi_k(\tilde{\boldsymbol{q}}), \ell_{x^t}(A^t) \right\rangle + \eta^{-1} D_{\tilde{\Phi}}(\tilde{\boldsymbol{q}}, \tilde{\boldsymbol{q}}^{t-1}),$$

$$= \underset{\tilde{\boldsymbol{q}} \in \tilde{\Delta}_k}{\operatorname{argmin}} \left\langle \tilde{\boldsymbol{q}}, J_k^\mathsf{T} \ell_{x^t}(A^t) \right\rangle + \eta^{-1} D_{\tilde{\Phi}}(\tilde{\boldsymbol{q}}, \tilde{\boldsymbol{q}}^{t-1}),$$

$$= \underset{\tilde{\boldsymbol{q}} \in \tilde{\Delta}_k}{\operatorname{argmin}} \left\langle \tilde{\boldsymbol{q}}, \nabla l_t(\tilde{\boldsymbol{q}}^{t-1}) \right\rangle + \eta^{-1} D_{\tilde{\Phi}}(\tilde{\boldsymbol{q}}, \tilde{\boldsymbol{q}}^{t-1}), \tag{108}$$

where $l_t(\tilde{\boldsymbol{\mu}}) \coloneqq \langle \Pi_k(\tilde{\boldsymbol{\mu}}), \ell_{x^t}(A^t) \rangle = \langle \boldsymbol{\mu}, \ell_{x^t}(A^t) \rangle$. Update (108) is, by definition [2], the MDA with the sequence of losses $l_t$ on $\operatorname{int} \tilde{\Delta}_k$, 'distance' function $D_{\tilde{\Phi}}(\cdot, \cdot)$, and learning rate $\eta$. Therefore, the MDA is exactly the update step of the GAA.

# F   The Generalized Aggregating Algorithm Using the Shannon Entropy S

The purpose of this appendix is to show that the GAA reduces to the AA when the former uses the Shannon entropy. In this case, generalized and classical mixability are equivalent. In what follows, we make use of the following proposition which is proved in C.5.

**Proposition 20** *For the Shannon entropy* S, *it holds that* $\tilde{S}^*(\boldsymbol{v}) = \log(\langle\exp(\boldsymbol{v}), \mathbf{1}_{\tilde{k}}\rangle + 1), \forall \boldsymbol{v} \in \mathbb{R}^{k-1}$, *and* $S^\star(\boldsymbol{z}) = \log\langle\exp(\boldsymbol{z}), \mathbf{1}_k\rangle, \forall \boldsymbol{z} \in \mathbb{R}^k$.

Let $\ell\colon \mathcal{A} \to [0, +\infty[^n$ be a loss and $\Phi$ be as in Proposition 14 and suppose that $\Phi$ and $\tilde{\Phi}^*$ are differentiable on $\operatorname{ri}\Delta_k$ and $\mathbb{R}^{k-1}$, respectively. It was shown in [9] that

$$\nabla\Phi^\star(\nabla\Phi(\boldsymbol{q}) - \ell_x(A)) = \underset{\boldsymbol{\mu}\in\Delta_k}{\operatorname{argmin}}\langle\boldsymbol{\mu}, \ell_x(A)\rangle + D_\Phi(\boldsymbol{\mu}, \boldsymbol{q}), \tag{109}$$

$$\operatorname{Mix}_\Phi(\ell_x(A), \boldsymbol{q}) = \Phi^\star(\nabla\Phi(\boldsymbol{q})) - \Phi^\star(\nabla\Phi(\boldsymbol{q}) - \ell_x(A)). \tag{110}$$

Let $\boldsymbol{q} \in \operatorname{ri}\Delta_k$. By definition of S, $\nabla S(\boldsymbol{q}) = \log\boldsymbol{q} + \mathbf{1}_k$, and due to Proposition 20, $S^\star(\boldsymbol{z}) = \log\langle\exp\boldsymbol{z}, \mathbf{1}_k\rangle, \boldsymbol{z} \in \mathbb{R}^k$. Therefore, $\nabla S(\boldsymbol{q}) - \eta\ell_x(A) = \log(\exp(-\eta\ell_x(A)) \odot \boldsymbol{q}) + \mathbf{1}_k$ and $\nabla S^\star(\boldsymbol{z}) = \frac{\exp\boldsymbol{z}}{\langle\exp\boldsymbol{z}, \mathbf{1}_k\rangle}, \forall(x, A) \in [n] \times (\operatorname{dom}\ell)^k$. Thus,

$$\nabla S^\star(\nabla S(\boldsymbol{q}) - \eta\ell_x(A)) = \frac{\exp(-\eta\ell_x(A)) \odot \boldsymbol{q}}{\langle\exp(-\eta\ell_x(A)), \boldsymbol{q}\rangle}. \tag{111}$$

Let $S_\eta := \eta^{-1} S$. Then $\nabla S = \eta\nabla S_\eta$ and $\forall\boldsymbol{z} \in \mathbb{R}^k, \nabla S_\eta^\star(\boldsymbol{z}) = \nabla S^\star(\eta\boldsymbol{z})$ [9].[2] Then the left hand side of (111) can be written as $\nabla S_\eta^\star(\nabla S_\eta(\boldsymbol{q}) - \ell_x(A))$. Using this fact, (109) and (111) show that the update distribution $\boldsymbol{q}^t$ of the GAA (Algorithm 2) coincides with that of the AA after substituting $\boldsymbol{q}, x$, and $A$ by $\boldsymbol{q}^{t-1}, x^t$, and $A^t := [\boldsymbol{a}_\theta]_{\theta\in[k]}$, respectively.

Now using the fact that $\operatorname{Mix}_S^\eta(\ell_x(A), \boldsymbol{q}) = \eta^{-1}\operatorname{Mix}_S(\eta\ell_x(A), \boldsymbol{q})$ [9] and (110), we get

$$\operatorname{Mix}_S^\eta(\ell_x(A), \boldsymbol{q}) = \eta^{-1}[S^\star(\nabla S(\boldsymbol{q})) - S^\star(\nabla S(\boldsymbol{q}) - \eta\ell_x(A))],$$

$$= -\eta^{-1}\log\langle\exp(-\eta\ell_x(A)), \boldsymbol{q}\rangle. \tag{112}$$

Equation 112 shows that the $\eta$-mixability condition is equivalent to the $(\eta, S)$-mixability condition for a finite loss. This remains true for losses taking infinite values — see the proof of Theorem 11 in Appendix C.5.

## G  Legendre $\Phi$, but no $\Phi$-mixable $\ell$

In this appendix, we construct a *Legendre type* entropy [11] for which there are no $\Phi$-mixable losses satisfying a weak condition (see below).

Let $\ell : \mathcal{A} \to [0, +\infty]^n$ be a loss satisfying condition 1. According to Alexandrov's Theorem, a concave function is twice differentiable almost everywhere (see e.g. [4, Thm. 6.7]). Now we give a version of Theorem 14 which does not assume the twice differentiability of the Bayes risk. The proof is almost identical to that of Theorem 14 with only minor modifications.

**Theorem 27.** *Let* $\Phi\colon \mathbb{R}^k \to \mathbb{R} \cup \{+\infty\}$ *be an entropy such that* $\tilde{\Phi}$ *is twice differentiable on* $\operatorname{int}\tilde{\Delta}_k$, *and* $\ell\colon \mathcal{A} \to [0, +\infty]^n$ *a loss satisfying Condition 1 and such that* $\exists(\tilde{\boldsymbol{p}}, \boldsymbol{v}) \in \mathcal{D}\times\mathbb{R}^{\tilde{n}}, \mathsf{H}\underline{\tilde{L}}_\ell(\tilde{\boldsymbol{p}})\boldsymbol{v} \neq \mathbf{0}_{\tilde{n}}$, *where* $\mathcal{D} \subset \operatorname{int}\tilde{\Delta}_n$ *is a set of Lebesgue measure 1 where* $\underline{\tilde{L}}_\ell$ *is twice differentiable, and define*

$$\underline{\eta_\ell}^* := \inf_{\tilde{\boldsymbol{p}}\in\mathcal{D}}(\lambda_{\max}([\mathsf{H}\underline{\tilde{L}}_{\log}(\tilde{\boldsymbol{p}})]^{-1}\mathsf{H}\underline{\tilde{L}}_\ell(\tilde{\boldsymbol{p}})))^{-1}. \tag{113}$$

*Then* $\ell$ *is* $\Phi$*-mixable only if* $\underline{\eta_\ell}^*\Phi - S$ *is convex on* $\Delta_k$.

The new condition on the Bayes risk is much weaker than requiring $\underline{L}_\ell$ to be twice differentiable on $]0, +\infty[^n$. In the next example, we will show that there exists a Legendre type entropy for which there are no $\Phi$-mixable losses satisfying the condition of Theorem 27.

**Example 1.** *Let* $\Phi : \mathbb{R}^2 \to \mathbb{R} \cup \{+\infty\}$ *be an entropy such that*

$$\forall q \in]0, 1[,\ \Phi(q, 1-q) = \tilde{\Phi}(q) = \int_{1/2}^q \log\left(\frac{\log(1-t)}{\log t}\right)dt.$$

$\tilde{\Phi}$ *is differentiable and strictly convex on the open set* $(0, 1)$. *Furthermore, it satisfies* (10) *which makes it a function of Legendre type [11, Lem. 26.2]. In fact,* (10) *is satisfied due to*

$$\left| \frac{d}{dq} \tilde{\Phi}(q) \right| = \left| \log \left( \frac{\log(1-q)}{\log q} \right) \right| \overset{q \to b}{\to} +\infty, \text{ where } b \in \{0, 1\},$$

$$\frac{d^2}{dq^2} \tilde{\Phi}(q) = \frac{-1}{q \log q} + \frac{-1}{(1-q)\log(1-q)} > 0, \ \forall q \in ]0, 1[.$$

*The Shannon entropy on* $\Delta_2$ *is defined by* $\mathrm{S}(q, 1-q) = \tilde{\mathrm{S}}(q) = q \log q + (1-q)\log(1-q)$, *for* $q \in ]0, 1[$. *Thus,* $\frac{d^2}{dq^2} \tilde{\mathrm{S}}(q) = \frac{1}{q(1-q)}$.

*Suppose now that there exists a* $\Phi$-*mixable loss* $\ell \colon \mathcal{A} \to [0, +\infty]^n$ *satisfying condition 1 and such that* $\exists (\tilde{\boldsymbol{p}}, \boldsymbol{v}) \in \mathcal{D} \times \mathbb{R}^{\tilde{n}}, \mathsf{H} \underline{\tilde{L}}_\ell(\tilde{\boldsymbol{p}}) \boldsymbol{v} \neq \boldsymbol{0}_{\tilde{n}}$. *Let* $\underline{\eta_\ell}^*$ *be as in* (113). *By definition, we have* $\underline{\eta_\ell}^* < +\infty$, *and thus*

$$\underline{\eta_\ell}^* \left[ \frac{d^2}{dq^2} \tilde{\Phi}(q) \right] \left[ \frac{d^2}{dq^2} \tilde{\mathrm{S}}(q) \right]^{-1} = \underline{\eta_\ell}^* \left( \frac{q-1}{\log q} + \frac{-q}{\log(1-q)} \right) \overset{q \to b}{\to} 0, \tag{114}$$

*where* $b \in \{0, 1\}$. *From Lemma 3,* (114) *implies that* $\underline{\eta_\ell}^* \Phi - \mathrm{S}$ *is not convex on* $\Delta_k$, *which is a contradiction according to Theorem 27.*

## H  Loss Surface and Superprediction Set

In this appendix, we derive an expression for the curvature of the image of a proper loss function. We will need the following lemma.

**Lemma 28.** *Let* $\sigma : [0, +\infty[^n \to \mathbb{R}$ *be a 1-homogeneous, twice differentiable function on* $]0, +\infty[^n$. *Then* $\sigma$ *is concave on* $]0, +\infty[^n$ *if and only if* $\tilde{\sigma} = \sigma \circ \amalg_n$ *is concave on* $\mathrm{int}\, \tilde{\Delta}_n$.

*Proof.* The forward implication is immediate; if $\sigma$ is concave on $]0, +\infty[^n$, then $\sigma \circ \amalg_k$ is concave on $\mathrm{int}\, \tilde{\Delta}_k$, since $\amalg_k$ is an affine function.

Now assume that $\tilde{\sigma}$ is concave on $\mathrm{int}\, \tilde{\Delta}_k$. Let $\lambda \in [0, 1]$ and $(\boldsymbol{p}, \boldsymbol{q}) \in [0, +\infty[^n \times [0, +\infty[^n$. We need to show that

$$\lambda \sigma(\boldsymbol{p}) + (1-\lambda)\sigma(\boldsymbol{q}) \leq \sigma(\lambda \boldsymbol{p} + (1-\lambda)\boldsymbol{q}). \tag{115}$$

Note that if $\boldsymbol{p} = \boldsymbol{0}$ or $\boldsymbol{q} = \boldsymbol{0}$, (115) is trivially with equality due to the 1-homogeneity of $\sigma$. Now assume that $\boldsymbol{p}$ and $\boldsymbol{q}$ are non-zero and let $c := \lambda \|\boldsymbol{p}\|_1 + (1-\lambda)\|\boldsymbol{q}\|_1$. For convenience, we also denote $\boldsymbol{p}_1 = \boldsymbol{p}/\|\boldsymbol{p}\|_1$ and $\boldsymbol{q}_1 = \boldsymbol{q}/\|\boldsymbol{q}\|_1$ which are both in $\Delta_n$. It follows that

$$\lambda \sigma(\boldsymbol{p}) + (1-\lambda)\sigma(\boldsymbol{q}) = cM \left( \lambda \frac{\|\boldsymbol{p}\|_1}{c} \sigma(\boldsymbol{p}_1) + (1-\lambda)\frac{\|\boldsymbol{q}\|_1}{c} \sigma(\boldsymbol{q}_1) \right),$$

$$= c \left( \lambda \frac{\|\boldsymbol{p}\|_1}{c} \tilde{\sigma}(\tilde{\boldsymbol{p}}_1) + (1-\lambda)\frac{\|\boldsymbol{q}\|_1}{c} \tilde{\sigma}(\tilde{\boldsymbol{q}}_1) \right),$$

$$\leq c\tilde{\sigma} \left( \lambda \frac{\|\boldsymbol{p}\|_1}{c} \tilde{\boldsymbol{p}}_1 + (1-\lambda)\frac{\|\boldsymbol{q}\|_1}{c} \tilde{\boldsymbol{q}}_1 \right),$$

$$= c\sigma \left( \lambda \frac{\|\boldsymbol{p}\|_1}{c} \boldsymbol{p}_1 + (1-\lambda)\frac{\|\boldsymbol{q}\|_1}{c} \boldsymbol{q}_1 \right),$$

$$= \sigma(\lambda \boldsymbol{p} + (1-\lambda)\boldsymbol{q}),$$

where the first and last equalities are due the 1-homogeneity of $\sigma$ and the inequality is due to $\tilde{\sigma}$ being concave on the $\mathrm{int}\, \tilde{\Delta}_n$. □

## H.1 Convexity of the Superprediction Set

In the literature, many theoretical results involving loss functions relied on the fact that the superprediction set of a proper loss is convex [16, 6]. An earlier proof of this result by [16] was incomplete[3]. In the next theorem we restate this result.

**Theorem 29.** *If $\ell \colon \Delta_n \to [0, +\infty[^n$ is a continuous proper loss, then $\mathscr{S}_\ell = \bigcap_{\boldsymbol{p} \in \Delta_n} \mathcal{H}_{-\boldsymbol{p}, -\underline{L}_\ell(\boldsymbol{p})}$. In particular, $\mathscr{S}_\ell$ is convex.*

$\mathscr{S}_\ell \subseteq \bigcap_{\boldsymbol{p} \in \Delta_n} \mathcal{H}_{-\boldsymbol{p}, -\underline{L}_\ell(\boldsymbol{p})}.$ : Let $\boldsymbol{v} \in \mathscr{S}_\ell$, $\boldsymbol{u} \in [0, +\infty[^n$, and $\boldsymbol{q} \in \Delta_n$ such that $\boldsymbol{v} = \ell(\boldsymbol{q}) + \boldsymbol{u}$. Since $\ell$ is proper then $\forall \boldsymbol{p} \in \Delta_n, \underline{L}_\ell(\boldsymbol{p}) = \langle \boldsymbol{p}, \ell(\boldsymbol{p}) \rangle \leq \langle \boldsymbol{p}, \ell(\boldsymbol{q}) \rangle \leq \langle \boldsymbol{p}, \ell(\boldsymbol{q}) + \boldsymbol{u} \rangle = \langle \boldsymbol{p}, \boldsymbol{v} \rangle$. Therefore, $\boldsymbol{v} \in \bigcap_{\boldsymbol{p} \in \Delta_n} \mathcal{H}_{-\boldsymbol{p}, -\underline{L}_\ell(\boldsymbol{p})}.$

$[\bigcap_{\boldsymbol{p} \in \Delta_n} \mathcal{H}_{-\boldsymbol{p}, -\underline{L}_\ell(\boldsymbol{p})} \subseteq \mathscr{S}_\ell]$: Let $\boldsymbol{v} \in \bigcap_{\boldsymbol{p} \in \Delta_n} \mathcal{H}_{-\boldsymbol{p}, -\underline{L}_\ell(\boldsymbol{p})}$. Let $\Omega = [n]$, $\Delta(\Omega) = \Delta_n$, and $Q(\boldsymbol{p}, x) = \ell_x(\boldsymbol{p}) - v_x$ for all $(\boldsymbol{p}, x) \in \Delta_n \times [n]$. Since $\boldsymbol{v} \in \bigcap_{\boldsymbol{p} \in \Delta_n} \mathcal{H}_{-\boldsymbol{p}, -\underline{L}_\ell(\boldsymbol{p})}$, $\mathbb{E}_{x \sim \boldsymbol{p}} Q(\boldsymbol{p}, x) = \langle \boldsymbol{p}, \ell(\boldsymbol{p}) \rangle - \langle \boldsymbol{p}, \boldsymbol{v} \rangle \leq 0$ for all $\boldsymbol{p} \in \Delta_n$. Lemma 4, implies that there exists $\boldsymbol{p}_* \in \Delta_n$ such that $Q(\boldsymbol{p}_*, x) = \ell_x(\boldsymbol{p}_*) - v_x \leq 0$, for all $x \in [n]$. This shows that $\boldsymbol{v} \in \mathscr{S}_\ell$. $\qquad\square$

## H.2 Curvature of the Loss Surface

The *normal curvature* of a $\tilde{n}$-manifold $\mathcal{S}$ [13] at a point $\boldsymbol{r} \in \mathcal{S}$ in the direction of $\boldsymbol{w} \in T_{\boldsymbol{r}}\mathcal{S}$, where $T_{\boldsymbol{r}}\mathcal{S}$ is the *tangent space* of $\mathcal{S}$ at $\boldsymbol{r} \in \mathcal{S}$, is defined by

$$\kappa(\boldsymbol{r}, \boldsymbol{w}) = \frac{\langle \boldsymbol{w}, \mathsf{DN}^{\mathcal{S}}(\boldsymbol{r})\boldsymbol{w} \rangle}{\langle \boldsymbol{w}, \boldsymbol{w} \rangle}, \tag{116}$$

where $\mathsf{N}^{\mathcal{S}}(\boldsymbol{r})$ is the normal vector to the surface at $\boldsymbol{r}$. The *minimum principal curvature* of $\mathcal{S}$ at $\boldsymbol{r}$ is expressed as $\underline{\kappa}(\boldsymbol{r}) := \inf\{\kappa(\boldsymbol{r}, \boldsymbol{w}) : \boldsymbol{w} \in T_{\boldsymbol{r}}\mathcal{S} \cap \mathcal{B}(\boldsymbol{r}, 1)\}$.

In the next theorem, we establish a direct link between the curvature of a loss surface and the Hessian of the loss' Bayes risk.

**Theorem 30.** *Let $\ell \colon \mathrm{ri}\,\Delta_n \to [0, +\infty[^n$ be a loss whose Bayes risk is twice differentiable and strictly concave on $]0, +\infty[^n$. Let $\boldsymbol{p} \in \mathrm{ri}\,\Delta_n$, $X_{\boldsymbol{p}} := I_{\tilde{n}} - \tilde{\boldsymbol{p}}\mathbf{1}_{\tilde{n}}^{\mathsf{T}}$, and $\boldsymbol{w} \in T_{\tilde{\ell}(\tilde{\boldsymbol{p}})}\mathcal{S}_\ell$. Then*

1. *$\exists \boldsymbol{v} \in \mathbb{R}^{n-1}$ such that $\mathsf{D}\tilde{\ell}(\tilde{\boldsymbol{p}})\boldsymbol{v} = \boldsymbol{w}$.*

2. *$\mathcal{S}_\ell$ is a $\tilde{n}$-manifold.*

3. *The normal curvature of $\mathcal{S}_\ell$ at $\ell(\boldsymbol{p}) = \tilde{\ell}(\tilde{\boldsymbol{p}})$ in the direction $\boldsymbol{w}$ is given by*

$$\kappa_\ell(\ell(\boldsymbol{p}), \boldsymbol{w}) = \left\| \begin{bmatrix} X_{\boldsymbol{p}} \\ -\tilde{\boldsymbol{p}}^{\mathsf{T}} \end{bmatrix} (-\mathsf{H}\underline{L}_\ell(\tilde{\boldsymbol{p}}))^{\frac{1}{2}} \boldsymbol{u} \right\|^{-1}, \tag{117}$$

*where $\boldsymbol{u} = (-\mathsf{H}\underline{L}_\ell(\tilde{\boldsymbol{p}}))^{\frac{1}{2}}\boldsymbol{v} / \|(-\mathsf{H}\underline{L}_\ell(\tilde{\boldsymbol{p}}))^{\frac{1}{2}}\boldsymbol{v}\|.*

It becomes clear from (117) that smaller eigenvalues of $-\mathsf{H}\underline{L}_\ell(\tilde{\boldsymbol{p}})$ will tend to make the loss surface more curved at $\ell(\boldsymbol{p})$, and vice versa.

Before proving Theorem 30, we first define parameterizations on manifolds.

**Definition 31** (Local and Global Parameterization). *Let $\mathcal{S} \subseteq \mathbb{R}^n$ be a $\tilde{n}$-manifold and $\mathcal{U}$ an open set in $\mathbb{R}^{\tilde{n}}$. The map $\varphi \colon \mathcal{U} \to \mathcal{S}$ is called a* local parameterization *of $\mathcal{S}$ if $\mathsf{D}\varphi(\boldsymbol{u}) \colon \mathbb{R}^{\tilde{n}} \to T_{\varphi(\boldsymbol{u})}\mathcal{S}$ is injective for all $\boldsymbol{u} \in \mathcal{U}$, where $T_{\varphi(\boldsymbol{u})}\mathcal{S}$ is the tangent space of $\mathcal{S}$ at $\varphi(\boldsymbol{u}) \in \mathcal{S}$. $\varphi$ is called a* global parameterization *of $\mathcal{S}$ if it is, additionally, onto.*

Let $\varphi$ be a global parameterization of $\mathcal{S}$ and $\mathsf{N}^{\varphi} := \mathsf{N}^{\mathcal{S}} \circ \varphi$. By a direct application of the chain rule, (116) can be written as

$$\kappa(\varphi(\boldsymbol{u}), \boldsymbol{w}) = \frac{\langle \boldsymbol{w}, \mathsf{DN}^{\varphi}(\boldsymbol{u})\boldsymbol{v} \rangle}{\langle \boldsymbol{w}, \boldsymbol{w} \rangle}, \tag{118}$$

where $\boldsymbol{v}$ is such that $D\varphi(\boldsymbol{u})\boldsymbol{v} = \boldsymbol{w}$. The existence of such a $\boldsymbol{v}$ is guaranteed by the fact that $D\varphi$ is injective and $\dim \mathbb{R}^{\tilde{n}} = \dim T_{\varphi(\boldsymbol{u})}\mathcal{S} = \tilde{n}$.

***Theorem 30.*** First we show that $\mathcal{S}_\ell$ is a $\tilde{n}$-manifold. Consider the map $\tilde{\ell} : \operatorname{int} \tilde{\Delta}_n \to \mathcal{S}_\ell$ and note that $\operatorname{int} \tilde{\Delta}_n$ is trivially a $\tilde{n}$-manifold. Due to the strict concavity of the Bayes risk, $\tilde{\ell}$ is injective [14] and from Lemmas 8 and 28, $D\tilde{\ell}(\tilde{\boldsymbol{p}}) : \mathbb{R}^{\tilde{n}} \to T_{\tilde{\ell}(\tilde{\boldsymbol{p}})}\mathcal{S}_\ell$ is also injective. Therefore, $\tilde{\ell}$ is an *immersion* [10]. $\tilde{\ell}$ is also *proper* in the sense that the preimage of every compact subset of $\mathcal{S}_\ell$ is compact. Therefore, $\tilde{\ell}$ is a proper injective immersion, and thus it is an embedding from the $\tilde{n}$-manifold $\operatorname{int} \tilde{\Delta}_n$ to $\mathcal{S}_\ell$ (ibid.). Hence, $\mathcal{S}_\ell$ is a manifold.

Now we prove (117). The map $\tilde{\ell}$ is a global parameterization of $\mathcal{S}_\ell$. In fact, from Lemma 8, $D\tilde{\ell}(\tilde{\boldsymbol{p}})$ has rank $\tilde{n}$, for all $\tilde{\boldsymbol{p}} \in \operatorname{int} \tilde{\Delta}_n$, which implies that $D\tilde{\ell}(\tilde{\boldsymbol{p}})$ is onto from $\mathbb{R}^{\tilde{n}}$ to $T_{\tilde{\ell}(\tilde{\boldsymbol{p}})}\mathcal{S}_\ell$. Therefore, given $\boldsymbol{w} \in T_{\tilde{\ell}(\tilde{\boldsymbol{p}})}\mathcal{S}_\ell$, there exists $\boldsymbol{v} \in \mathbb{R}^{\tilde{n}}$ such that $\boldsymbol{w} = D\tilde{\ell}(\tilde{\boldsymbol{p}})\boldsymbol{v}$. Furthermore, Lemma 8 implies that $\mathsf{N}^{\tilde{\ell}}(\tilde{\boldsymbol{p}}) = \boldsymbol{p}$, since $\langle \boldsymbol{p}, D\tilde{\ell}(\tilde{\boldsymbol{p}}) \rangle = \boldsymbol{0}_{\tilde{n}}^{\mathsf{T}}$. Substituting $\mathsf{N}^{\tilde{\ell}}$ into (118) yields

$$
\begin{aligned}
\kappa_\ell(\tilde{\ell}(\tilde{\boldsymbol{p}}), \boldsymbol{w}) &= \frac{\boldsymbol{v}^{\mathsf{T}}(D\tilde{\ell}(\tilde{\boldsymbol{p}}))^{\mathsf{T}} \begin{bmatrix} I_{\tilde{n}}, \\ \boldsymbol{1}_{\tilde{n}} \end{bmatrix} \boldsymbol{v}}{\left\langle D\tilde{\ell}(\tilde{\boldsymbol{p}})\boldsymbol{v}, D\tilde{\ell}(\tilde{\boldsymbol{p}})\boldsymbol{v} \right\rangle}, \\
&= \frac{\boldsymbol{v}^{\mathsf{T}} \mathsf{H}\underline{\tilde{L}}_\ell(\tilde{\boldsymbol{p}}) \begin{bmatrix} X_{\boldsymbol{p}}^{\mathsf{T}}, & -\tilde{\boldsymbol{p}} \end{bmatrix} \begin{bmatrix} I_{\tilde{n}} \\ \boldsymbol{1}_{\tilde{n}} \end{bmatrix} \boldsymbol{v}}{\left\langle D\tilde{\ell}(\tilde{\boldsymbol{p}})\boldsymbol{v}, D\tilde{\ell}(\tilde{\boldsymbol{p}})\boldsymbol{v} \right\rangle}, \\
&= \frac{\boldsymbol{v}^{\mathsf{T}} \mathsf{H}\underline{\tilde{L}}_\ell(\tilde{\boldsymbol{p}})\boldsymbol{v}}{\boldsymbol{v}^{\mathsf{T}} \mathsf{H}\underline{\tilde{L}}_\ell(\tilde{\boldsymbol{p}}) \begin{bmatrix} X_{\boldsymbol{p}}^{\mathsf{T}}, & -\tilde{\boldsymbol{p}} \end{bmatrix} \begin{bmatrix} X_{\boldsymbol{p}} \\ -\tilde{\boldsymbol{p}}^{\mathsf{T}} \end{bmatrix} \mathsf{H}\underline{\tilde{L}}_\ell(\tilde{\boldsymbol{p}})\boldsymbol{v}}.
\end{aligned}
\tag{119}
$$

Setting $\boldsymbol{u} = (-\mathsf{H}\underline{\tilde{L}}_\ell(\tilde{\boldsymbol{p}}))^{\frac{1}{2}}\boldsymbol{v}/\|(-\mathsf{H}\underline{\tilde{L}}_\ell(\tilde{\boldsymbol{p}}))^{\frac{1}{2}}\boldsymbol{v}\|$ in (119) gives the desired result. $\qquad\square$

# I  Classical Mixability Revisited

In this appendix, we provide a more concise proof of the necessary and sufficient conditions for the convexity of the superprediction set [14].

**Theorem 32.** *Let $\ell : \Delta_n \to [0, +\infty[^n$ be a strictly proper loss whose Bayes risk is twice differentiable on $]0, +\infty[^n$. The following points are equivalent;*

(i) $\forall \tilde{\boldsymbol{p}} \in \operatorname{int} \tilde{\Delta}_n, \ \eta \mathsf{H}\underline{\tilde{L}}_\ell(\tilde{\boldsymbol{p}}) \succeq \mathsf{H}\underline{\tilde{L}}_{\log}(\tilde{\boldsymbol{p}})$.

(ii) $e^{-\eta \mathscr{S}_\ell} = \bigcap_{\boldsymbol{p} \in \Delta_n} \mathcal{H}_{\tau(\boldsymbol{p}),1} \cap [0, +\infty[^n, \text{ where } \tau(\boldsymbol{p}) := \boldsymbol{p} \odot e^{\eta \ell(\boldsymbol{p})}$.

(iii) $e^{-\eta \mathscr{S}_\ell}$ *is convex.*

*Proof.* We already showed (i) $\implies$ (ii) $\implies$ (iii) in the proof of Theorem 7.

We now show (iii) $\implies$ (i). Since $e^{-\eta \mathscr{S}_\ell}$ is convex, any point $\boldsymbol{s} \in \operatorname{bd} e^{-\eta \mathscr{S}_\ell}$ is supported by a hyperplane [7, Lem. A.4.2.1]. Since $\boldsymbol{u} \to e^{-\eta \boldsymbol{u}}$ is a homeomorphism, it maps boundaries to boundaries. From this and Lemma 17, $\operatorname{bd} e^{-\eta \mathscr{S}_\ell} = e^{-\eta \mathcal{S}_\ell}$. Thus, for $\boldsymbol{p} \in \operatorname{ri} \Delta_n$, there exists a unit-norm vector $\boldsymbol{u} \in \mathbb{R}^n$ such that for all $\boldsymbol{s} \in \mathscr{S}_\ell$ it either holds that $\langle \boldsymbol{u}, e^{-\eta \ell(\boldsymbol{p})} \rangle \leq \langle \boldsymbol{u}, e^{-\eta \boldsymbol{s}} \rangle$; or $\langle \boldsymbol{u}, e^{-\eta \ell(\boldsymbol{p})} \rangle \geq \langle \boldsymbol{u}, e^{-\eta \boldsymbol{s}} \rangle$. It is easy to see that it is the latter case that holds, since we can choose $\boldsymbol{s} = \ell(\boldsymbol{r}) + c\boldsymbol{1} \in \mathscr{S}_\ell$, for $\boldsymbol{r} \in \Delta_n$, and make $\langle \boldsymbol{u}, e^{-\eta \boldsymbol{s}} \rangle$ arbitrarily small by making $c \in \mathbb{R}$ large. Therefore, $\forall \boldsymbol{r} \in \operatorname{ri} \Delta_n, \langle \boldsymbol{u}, e^{-\eta \tilde{\ell}(\tilde{\boldsymbol{p}})} \rangle = \langle \boldsymbol{u}, e^{-\eta \ell(\boldsymbol{p})} \rangle \geq \langle \boldsymbol{u}, e^{-\eta \ell(\boldsymbol{r})} \rangle = \langle \boldsymbol{u}, e^{-\eta \tilde{\ell}(\tilde{\boldsymbol{r}})} \rangle$ and $\tilde{\boldsymbol{p}}$ is a critical point of the function $f(\tilde{\boldsymbol{r}}) := \langle \boldsymbol{u}, e^{-\eta \tilde{\ell}(\tilde{\boldsymbol{r}})} \rangle$ on $\operatorname{int} \tilde{\Delta}_n$. This implies that $\nabla f(\tilde{\boldsymbol{p}}) = \boldsymbol{0}_{\tilde{n}}$; that is, $-\eta \langle \boldsymbol{u}, \operatorname{diag}(e^{-\eta \tilde{\ell}(\tilde{\boldsymbol{p}})})D\tilde{\ell}(\tilde{\boldsymbol{p}}) \rangle = -\eta \langle \operatorname{diag}(e^{-\eta \tilde{\ell}(\tilde{\boldsymbol{p}})})\boldsymbol{u}, D\tilde{\ell}(\tilde{\boldsymbol{p}}) \rangle = \boldsymbol{0}_{\tilde{n}}^{\mathsf{T}}$. From Lemma 8, there exists $\lambda \in \mathbb{R}$ such that $\operatorname{diag}(e^{-\eta \tilde{\ell}(\tilde{\boldsymbol{p}})})\boldsymbol{u} = \lambda \boldsymbol{p}$. Therefore, $\boldsymbol{u} = \lambda \boldsymbol{p} \odot e^{\eta \tilde{\ell}(\tilde{\boldsymbol{p}})}$, where $\lambda = \|\boldsymbol{p} \odot e^{\eta \tilde{\ell}(\tilde{\boldsymbol{p}})}\|^{-1}$,

since $\|\boldsymbol{u}\| = 1$. For $\boldsymbol{v} \in \mathbb{R}^{n-1}$, let $\tilde{\boldsymbol{\alpha}}^t := \tilde{\boldsymbol{p}} + t\boldsymbol{v}$, where $t \in \{s : \tilde{\boldsymbol{p}} + s\boldsymbol{v} \in \operatorname{int} \tilde{\Delta}_n\}$. Since $f$ is twice differentiable and attains a maximum at $\tilde{\boldsymbol{p}}$,

$$
\begin{aligned}
0 \geq \frac{1}{\lambda\eta} \left. \frac{d^2}{dt^2} f \circ \tilde{\boldsymbol{\alpha}}^t \right|_{t=0} &= \frac{1}{\lambda} \left. \frac{d}{dt} \left\langle \boldsymbol{u}, \operatorname{diag} e^{-\eta\tilde{\ell}(\tilde{\boldsymbol{\alpha}}^t)} \mathsf{D}\tilde{\ell}(\tilde{\boldsymbol{\alpha}}^t)\boldsymbol{v} \right\rangle \right|_{t=0}, \\
&= \left. \frac{d}{dt} \left\langle \boldsymbol{p} \odot e^{\eta\tilde{\ell}(\tilde{\boldsymbol{p}})}, \operatorname{diag} e^{-\eta\tilde{\ell}(\tilde{\boldsymbol{\alpha}}^t)} \mathsf{D}\tilde{\ell}(\tilde{\boldsymbol{p}})\boldsymbol{v} \right\rangle \right|_{t=0} + \left. \frac{d}{dt} \left\langle \boldsymbol{p}, \mathsf{D}\tilde{\ell}(\tilde{\boldsymbol{\alpha}}^t)\boldsymbol{v} \right\rangle \right|_{t=0}, \\
&= \eta\boldsymbol{v}^{\mathsf{T}} \mathsf{H}\underline{\tilde{L}}_\ell(\tilde{\boldsymbol{p}})(\mathsf{H}\underline{\tilde{L}}_{\log}(\tilde{\boldsymbol{p}}))^{-1} \mathsf{H}\underline{\tilde{L}}_\ell(\tilde{\boldsymbol{p}})\boldsymbol{v} - \boldsymbol{v}^{\mathsf{T}} \mathsf{H}\underline{\tilde{L}}_\ell(\tilde{\boldsymbol{p}})\boldsymbol{v}, \quad (120)
\end{aligned}
$$

where in the second equality we substituted $\boldsymbol{u}$ by $\lambda\boldsymbol{p} \odot e^{\eta\tilde{\ell}(\tilde{\boldsymbol{p}})}$ and in (120) we used (5) and (6) from Lemma 9. Note that by the assumptions on $\ell$ it follows that the Bayes risk $\underline{\tilde{L}}_\ell$ is strictly concave [14, Lemma 6] and $-\mathsf{H}\underline{\tilde{L}}_\ell(\tilde{\boldsymbol{p}})$ is symmetric negative-definite. In particular, $\mathsf{H}\underline{\tilde{L}}_\ell(\tilde{\boldsymbol{p}})$ is invertible. Setting $\hat{\boldsymbol{v}} := \mathsf{H}\underline{\tilde{L}}_\ell(\tilde{\boldsymbol{p}})\boldsymbol{v}$ in (120) yields

$$
0 \geq \eta\hat{\boldsymbol{v}}(\mathsf{H}\underline{\tilde{L}}_{\log}(\tilde{\boldsymbol{p}}))^{-1}\hat{\boldsymbol{v}} - \hat{\boldsymbol{v}}(\mathsf{H}\underline{\tilde{L}}_\ell(\tilde{\boldsymbol{p}}))^{-1}\hat{\boldsymbol{v}}.
$$

Since $\boldsymbol{v} \in \mathbb{R}^{n-1}$ was chosen arbitrarily, $(\mathsf{H}\underline{\tilde{L}}_\ell(\tilde{\boldsymbol{p}}))^{-1} \succeq \eta(\mathsf{H}\underline{\tilde{L}}_{\log}(\tilde{\boldsymbol{p}}))^{-1}, \forall \tilde{\boldsymbol{p}} \in \operatorname{int} \tilde{\Delta}_n$. This is equivalent to the condition $\forall \tilde{\boldsymbol{p}} \in \operatorname{int} \tilde{\Delta}_n$, $\eta\mathsf{H}\underline{\tilde{L}}_\ell(\tilde{\boldsymbol{p}}) \succeq \mathsf{H}\underline{\tilde{L}}_{\log}(\tilde{\boldsymbol{p}})$. $\qquad\square$

## J  An Experiment on Football Prediction Dataset

Figure 1: The figure corresponds to the 2005/2006, 2006/2007, 2007/2008, and 2008/2009 seasons. The solid lines represent, at each round $t$, the difference between the cumulative losses of the experts and that of the learner who uses either the AA (left) or the AGAA (right); that is, $\operatorname{Loss}_\theta^{\ell_{\mathrm{Brier}}}(t) - \operatorname{Loss}_{\mathfrak{M}}^{\ell_{\mathrm{Brier}}}(t)$, for $\mathfrak{M} \in \{\mathrm{AA}, \mathrm{AGAA}\}$. The red dashed lines represent the negative of the regret bound in (12) with respect to the best expert $\theta^*$; that is, $-R_{\ell_{\mathrm{Brier}}}^{\mathrm{S}} - \Delta R_{\theta^*}(t)$ at each round $t$.

### J.1  Testing the AGAA

To test the AGAA empirically, we used prediction data[4] from the British football leagues, including the Premier Leagues, Championships, Leagues 1-2, and Conferences. The first dataset contains predictions for the 2005/2006, 2006/2007, 2007/2008, and 2008/2009 seasons, matching the dataset used in [15]. The second dataset contains predictions for the 2009/2010, 2010/2011, 2011/2012, and 2012/2013 seasons. For this set, we considered predictions from 9 bookmakers; Bet365, Bet&Win, Blue Square, Gamebookers, Interwetten, Ladbrokes, Stan James, VC Bet, and William Hill.

On each dataset, we compared the performance of the AGAA with that of the AA using the Brier score (the Brier loss is 1-mixable). For the AGAA, we chose $\boldsymbol{\beta}^t$ according to Theorem 19 with $\boldsymbol{v}^t := -\frac{1}{2t} \sum_{s=1}^t \ell_{x^s}(A^s)$ and we set $\Phi = \mathrm{S}$, i.e. the Shannon entropy. The results in Figure 1 [resp. Figure 2] correspond to the seasons from 2005 to 2009 [resp. 2009 to 2013]. For fair comparison

**Figure 2:** The figure corresponds to the 2009/2010, 2010/2011, 2011/2012, and 2012/2013 seasons The solid lines represent, at each round $t$, the difference between the cumulative losses of the experts and that of the learner who uses either the AA (left) or the AGAA (right); that is, $\text{Loss}_\theta^{\ell_{\text{Brier}}}(t) - \text{Loss}_{\mathfrak{M}}^{\ell_{\text{Brier}}}(t)$, for $\mathfrak{M} \in \{\text{AA}, \text{AGAA}\}$. The red dashed lines represent the negative of the regret bound in (12) with respect to the best expert $\theta^*$; that is, $-R_{\ell_{\text{Brier}}}^{\text{S}} - \Delta R_{\theta^*}(t)$ at each round $t$.

**Figure 3:** The figure on the left [resp. right] hand side corresponds to the football seasons from 2005 to 2009 [resp. 2009 to 2013]. The solid lines represent, at each round $t$, the difference between the cumulative losses of the experts and that of the learner using the AA-AGAA meta algorithm (refer to text); that is, $\text{Loss}_\theta^{\ell_{\text{Brier}}}(t) - \text{Loss}_{\text{AA-AGAA}}^{\ell_{\text{Brier}}}(t)$. The red dashed lines represent the negative of the regret bound in (12) with respect to the best expert $\theta^*$; that is, $-R_{\ell_{\text{Brier}}}^{\text{S}} - \Delta R_{\theta^*}(t)$ at each round $t$.

with the results of Vovk [15], we 1) used the same substitution function as [15]; 2) used the same method for converting odds to probabilities; and 3) sorted the data first by date then by league and then by name of the host team (For more detail see [15]).

In all figures the solid lines represent, at each round $t$, the difference between the cumulative losses of the experts and that of the learners; that is, $\text{Loss}_\theta^{\ell_{\text{Brier}}}(t) - \text{Loss}_{\mathfrak{M}}^{\ell_{\text{Brier}}}(t)$, for $\mathfrak{M} = \text{AA}, \text{AGAA}$. The red dashed lines represent the negative of the regret bound in (12) with respect to the best expert $\theta^*$; that is, $-R_{\ell_{\text{Brier}}}^{\text{S}} - \Delta R_{\theta^*}(t) = -R_{\ell_{\text{Brier}}}^{\text{S}} - \sum_{s=1}^{t-1}(v_\theta^s - \langle \boldsymbol{v}^s, \boldsymbol{q}^s \rangle)$ at each round $t$, where $(\boldsymbol{q}^s)$ are the distributions over experts.

From Figures 1 and 2 it can be seen that the learners using the AGAA perform better than the best expert (and better than the AA) at the end of the games.

## J.2 Testing a AA-AGAA Meta-Learner

Consider the algorithm (referred to as AA-AGAA) that takes the outputs of the AGAA and the AA as in the previous section and aggregates them using the AA to yield a *meta prediction*. The worst case

regret of this algorithm is guaranteed not to exceed that of the original AA and AGAA by more than $\eta^{-1} \log 2$ for an $\eta$-mixable loss. Figure 3 shows the results for this algorithm for the same datasets as the previous section. The AA-AGAA still achieves a negative regret at the end of the game.

## Footnotes

[1]The Shannon entropy is usually defined with a minus sign. However, it will be more convenient for us to work without it.

[2]Reid et al. [9] showed the equality $\nabla\Phi_\eta^\star(\boldsymbol{u}) = \nabla\Phi^\star(\eta\boldsymbol{u}), \forall\boldsymbol{u} \in \operatorname{dom}\Phi^\star$, for any entropy differentiable on $\Delta_k$ - not just for the Shannon Entropy.

[3]It was claimed that if $\mathscr{S}_\ell$ is non-convex, there exists a point $\boldsymbol{s}_0$ on the loss surface $\mathcal{S}_\ell$ such that no hyperplane supports $\mathscr{S}_\ell$ at $\boldsymbol{s}_0$. The non-convexity of a set by itself is not sufficient to make such a claim; the continuity of the loss $\ell$ is required.

[4]The data was collected from http://www.football-data.co.uk/.