[Reviews · NeurIPS 2018]

Reviewer 1



Updated score from rebuttal. With improved presentation and a clear message, this could be a good paper. This paper further generalizes the useful notion of mixability for prediction with experts advice, building off a long history of results from the regret minimization community and solving an open problem about characterizing \Phi mixibility. This is mostly accomplished by defining the "support loss", which allows one to upper bound the given loss but possesses nicer properties and the authors to characterize Phi mixability completely, solving an open problem. A new algorithm (AGAA) is proposed and analyzed. The regret bound is intricate, making comparisons difficult. Some simulations are provided in the appendix, but I feel that this algorithm is under explored. The technical contributions seem solid, and a long line of work on Phi-mixability is essentially closed by this paper. However, I do have concerns with the presentation and impact. First, the paper is very densely written and would be better suited as a COLT or jmrl paper. Yes, there is a lot of background and definitions to go through, but many of the definitions are unnecessary or unnecessarily technical (e.g. line 87-90). While the technicality of the definitions are necessary for the most general case, the authors should consider presenting a less general case in the main body for venues like NIPS. The second worry is the interest to the NIPS community. Mixability and aggregating algorithms are not even mainstream in communities like COLT, and I'm afraid this paper will not find the right audience an NIPS and not have the impact it should. A few more comments: The typesetting and symbols used are sort of out of control, e.g. the symbol on line 169. I don't think S was explicitly defined to be the Shannon entropy.

Reviewer 2



The authors investigate the problem of achieving constant regret in the problem of learning with expert advice. Classical results of Vovk show that the latter is the case if the loss function satisfies a mixability criterion. Moreover, in that case the standard exponential weights aggregating algorithm achieves the constant regret. Inspired by the work of Reid et al. the authors define and analyze the notion of Phi-mixability, where Phi is some convex function, interpreted as a generic entropy function. Classic mixability is equivalent to Phi mixability for Phi being the Shannon entropy. Phi mixability basically corresponds to: for any mixed strategy there always exist a deterministic action that achieves loss smaller than the loss of a regularized loss, regularized by the divergence of Phi regularized loss (i.e. a Mirror descent step). Based on Phi mixability they define a generalization of the exponential aggregating algorithm where instead of an exponential updates they perform a mirror descent step. Their main result is a characterization of Phi mixability: a loss if Phi mixable if and only if it is mixable based on the Shannon entropy! The latter is a suprising result and contrary to prior work conjectures that for each loss there is an appropriate entropy function that is in some sense "optimal" for performing aggregation. The authors essentially show that it is without worst-case loss to only consider the Shannon entropy. Finally the propose a modification to their algorithm that can achieve better regret for some instances of the problem, even though it has the same worst-case performance as the exponential aggregation algorithm. I found the papers main finding interesting, but a bit of technical in nature and very specialized in scope. The paper is also very dense and hard to read in terms of notation (side note: I believe the Hessian H is never defined as notation). For this reason it is not clear to me that it will have a broad appeal at NIPS. The broadest appeal should be the potential practical performance improvement of their final adaptive algorithm. However the authors do not support their improved performance with any experimental performance of this algorithm. However, I do find the main technical result a deep and surprising one, hence I vote for weak acceptance.

Reviewer 3



This paper studies a generalization of the notion of mixability, called \Phi-mixability, and developed an adaptive generalized aggregating algorithm (AGAA) for it. To me, the interesting part of this paper is that it reveals the fundamental nature of the Shannon entropy in defining mixability: For a loss to be \Phi-mixable, it is necessary that it is mixable w.r.t. the Shannon entropy. Indeed, (9) gives an if and only if characterization of \Phi-mixability using mixability w.r.t. the Shannon entropy. Theorem 17 shows that exploiting the general \Phi-mixability does not help reduce the regret, though this result is only specific to the so-called generalized AA. The AGAA is not very exciting to me. This paper provides a scenario where the AGAA can even achieve a negative regret, but the scenario does not seem very general. A more satisfactory argument, in my opinion, should be similar to the results for, say, the optimistic mirror descent; there is a clear characterization for the data to be "easy", and it is clear that smaller regret can be achieved with easy data. Moreover, as mentioned in the paper, the AGAA can also yield worse performance. The main weakness of this paper, in my view, is that the main message is not very clear. The abstract emphasized both the novel characterization of \Phi-mixability and the AGAA. The characterization of \Phi-mixability, however, seems to suggest that generalizing the notion of mixability (at least in the direction of this paper) is not really helpful. The argument for the superiority of the AGAA is not strong enough, as I said in the preceding paragraph. I did not have the time to check the very long proofs. Other comments: 1. The presentation can be improved. This paper uses many terminologies and symbols without defining them first. For example, "Vovk's exponential mixing" in the first paragraph can be confusing to non-experts of online learning, and this confusion can be easily avoided by rewriting the sentence. The loss \ell is not defined before the appearance of R_\ell. The definition of H in (5) is missing. 2. The if and only if condition (9) looks similar to the condition proposed in the following two papers. a. "A descent lemma beyond Lipschitz gradient continuity: First-order methods revisited and applications" by Bauschke et al. b. "Relatively smooth convex optimization by first-order methods, and applications" by Lu et al. There is perhaps be some relationship. 3. ln 96, p. 3: The claim that "we make no topological assumption on A" is unnecessarily too strong. 4. ln 247, p. 6: The similarity with Vovk's work on the fundamental nature of the logarithmic loss is mentioned. However, this similarity is clear to me, as Vovk's result is about the choice of the loss function, instead of the definition of mixability. Please elaborate.